# Tumour acidosis remodels the glycocalyx to control lipid scavenging and ferroptosis

Anna Bång-Rudenstam [1], Myriam Cerezo-Magaña [1], Marton Horvath [1], Hugo Talbot [1], Emma Gustafsson [1], Stevanus Jonathan [2], Chaitali Chakraborty [3,4], Itzel Nissen [3,4], Kelin Gonçalves de Oliveira [1], Axel Boukredine [1], Sarah Beyer[1], Julio Enriquez Perez[2], Maria C. Johansson [1], Lena Kjellén [5], Emil Tykesson [6], Anders Malmström[6], Toin H. van Kuppevelt[7], Karin Forsberg-Nilsson [8,9], Jeffrey D. Esko [10,11], Silvia Remeseiro [3,4], Johan Bengzon[2,12], Valeria Governa [1] & Mattias Belting [1,13] ✉

Aggressive tumours are defined by microenvironmental stress adaptation and metabolic reprogramming. Within this niche, lipid droplet accumulation has emerged as a key strategy to buffer toxic lipids and suppress ferroptosis. Lipid droplet formation can occur via de novo lipogenesis or extracellular lipid-scavenging. However, how tumour cells coordinate these processes remains poorly understood. Here we identify a chondroitin sulfate (CS)-enriched glycocalyx as a hallmark of the acidic microenvironment in glioblastoma and central nervous system metastases. This CS-rich glycocalyx encapsulates tumour cells, limits lipid particle uptake and protects against lipid-induced ferroptosis. Mechanistically, we demonstrate that converging hypoxia-inducible factor and transforming growth factor beta signalling induces a glycan switch on syndecan-1—replacing heparan sulfate with CS—thereby impairing its lipid-scavenging function. Dual inhibition of CS biosynthesis and diacylglycerol O-acyltransferase-1, a critical enzyme in lipid droplet formation, triggers catastrophic lipid peroxidation and ferroptotic cell death. These findings define glycan remodelling as a core determinant of metabolic plasticity, positioning the dynamic glycocalyx as a master regulator of nutrient access, ferroptotic sensitivity and therapeutic vulnerability in cancer.

Aggressive tumours are defined by their ability to adapt to microenvironmental stress[1,2]. Within the tumour microenvironment (TME), cancer cells encounter intersecting pressures, including hypoxia, nutrient limitation, oxidative imbalance and extracellular acidosis, that reprogram cellular metabolism and promote therapy resistance[3,4]. A consistent feature of this adaptation is the accumulation of lipid droplets (LDs), which buffer toxic lipids, modulate the immune cell compartment[5,6], and promote survival under hostile conditions[7–9]. LD formation may result from de novo lipogenesis or from scavenging extracellular lipid sources such as free fatty acids (FAs), lipoproteins and extracellular vesicles (EVs)[10]. Although individual mechanisms of lipid uptake and storage have been described, how these processes are coordinated under chronic metabolic stress remains incompletely understood.

Glycosylation plays critical roles in cell–cell communication, immune modulation and nutrient scavenging[11–13]. Accumulating evidence supports an important role of heparan sulfate proteoglycans (HSPGs) in cancer cell uptake of lipoproteins and EV lipid particles[14–17], yet little is known about how glycan reorganization integrates with metabolic pathways under stress conditions. Glycans are synthesized through the orchestrated activity of glycosyltransferases and

**Fig. 1 | CS-enriched glycocalyx defines the lipid-rich, stressed tumour niche.** **a**, Fluorescence imaging of LDs stained by LipidTox in GBM tumour sections (left; representative of $n > 5$ patients), 3D cultures (middle; representative of $n > 10$ spheroids) and 2D cultures (right; representative of $n = 4$ cultures). Scale bars: left, 500 and 100 μm (zoomed); middle, 100 and 50 μm (zoomed); right, 10 μm. **b**, GSEA shows significant enrichment of pathways related to glycocalyx remodelling and lipid storage in LD$^+$ versus LD$^-$ GBM tumour areas captured by LCM ($n = 5$ patients). ECM, extracellular matrix. **c**, Volcano plots of enriched pathways (NES, normalized enrichment score; FDR < 0.1) in GBM 3D (LD$^+$) versus 2D (LD$^-$) primary cultures (U3054MG, U3047MG and U3017MG; $n = 3$ biological replicates per sample). Pathways from **b** are highlighted. **d**, Heatmap of genes selected based on their consistent upregulation ($\geq 0.5 \log_2$FC) in LD$^+$ versus

LD$^-$ GBM tumour areas ($n = 5$ patients) as well as in 3D versus 2D cultures from at least two out of three patients (U3054MG, U3047MG and U3017MG; $n = 3$ biological replicates per sample). **e**, Quantification of LD$^+$/CS$^+$ gene signature expression in the indicated GBM regions from IvyGAP ($n = 122$). Comparison of group means versus 'pseudopalisading cells' was performed. Boxplots represent the interquartile range with the median (centre line); the upper and lower quartiles are represented by whiskers, and outliers are represented as individual dots. Squares indicate zoomed area (**a**). N, necrosis. GSEA used the Hallmark, Reactome, KEGG and GOBP pathway databases (**b** and **c**). Significance was determined by Benjamini–Hochberg (BH)-adjusted nominal value (**b**) or by one-sample Wilcoxon signed-rank test (**e**). *FDR < 0.1; **FDR < 0.05 and ***FDR < 0.01 (**b**).

sulfotransferases, enabling rapid and context-dependent structural diversity[18], suggesting glycosylation as a sensitive mediator of environmental adaptation.

In this Article we aim to elucidate the molecular underpinnings of metabolic adaptation in the stressed TME. LD accumulation has been well documented in glioblastoma (GBM), a prototypical high-grade brain malignancy characterized by severe metabolic stress[5,19]. Unexpectedly, we observed prominent glycocalyx modification in the LD-rich niche of patient tumours, and explored how glycan remodelling intersects with lipid metabolism during tumour stress adaptation. Our results highlight an acidosis-induced glycan program with potential as a metabolic vulnerability, offering alternative therapeutic avenues for targeting the lipid-stressed TME.

## Results

### CS-enriched glycocalyx defines the lipid-rich, stressed tumour niche

We initially assessed the LD phenotype that was found in perinecrotic/pseudopalisading regions of patient GBM sections (Fig. 1a, left) and three-dimensional (3D) spheroid cultures (Fig. 1a, middle), but was largely absent in primary, patient-derived 2D cultures (Fig. 1a, right). Laser capture microdissection of tumour sections (excluding vasculature and CD68$^+$ immune cells) and transcriptome profiling (Extended Data Fig. 1a) revealed a striking enrichment of pathways related to glycocalyx within the LD$^+$ niche, particularly those involving CS and dermatan sulfate (DS) glycosaminoglycans (GAGs) and proteoglycan (PG) remodelling (Fig. 1b). Consistent with the LD$^+$ phenotype,

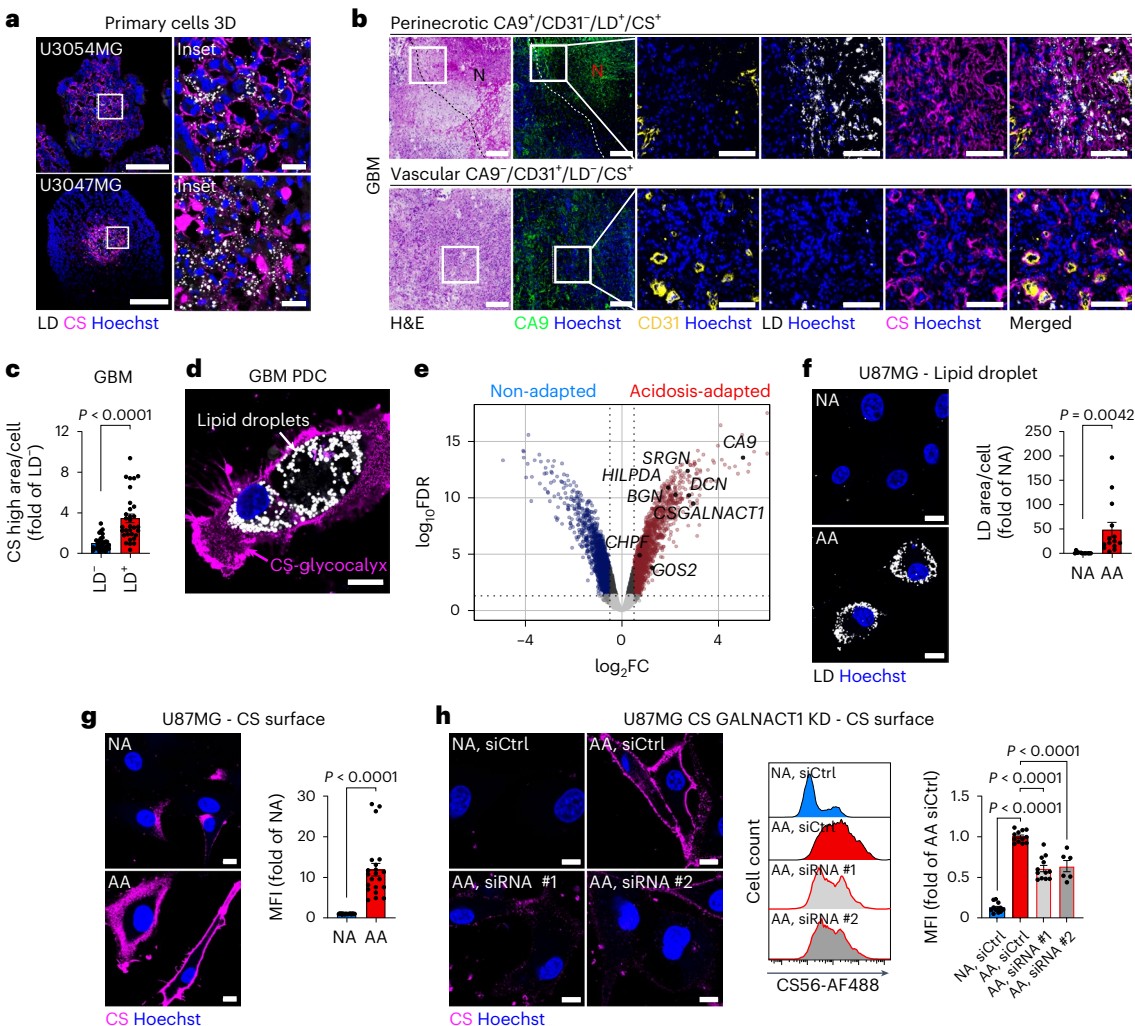

**Fig. 2 | CS-glycocalyx encapsulation is an adaptive response to tumour acidosis. a**, Fluorescence imaging of LDs and CS in patient-derived U3054MG and U3047MG 3D cultures (representative of $n$ = 10 spheroids per patient). Scale bars: 200 and 20 μm (zoomed). **b**, H&E and matching fluorescence images (indicated by dashed lines) of GBM tumour sections, highlighting the perinecrotic region (top row; CA9⁺/CD31⁻/LD⁺/CS⁺) and vascular region (bottom row; CA9⁻/CD31⁺/LD⁻/CS⁺) (representative of $n$ > 5 patients). Scale bars: 200 and 100 μm (zoomed). **c**, Quantification of CS high area in LD⁺ versus LD⁻ tumour regions from GBM sections (mean fold of LD⁻ ± s.e.m., $n$ = 32, four patients). **d**, Confocal imaging of LDs and CS surface signal in freshly resected GBM PDC (representative of $n$ = 4 patients). Scale bar: 10 μm. **e**, Volcano plot of upregulated (red) or downregulated (blue) genes from an mRNA array ($\log_2$FC > 0.5, adjusted $P$ value (adj$P_v$) < 0.05) in acidosis-adapted (pH 6.4) versus non-adapted (pH 7.4) U87MG GBM cells (mean fold of non-adapted ± s.e.m., $n$ = 3 biological replicates). **f**, Confocal imaging (left) of LDs in U87MG acidosis-adapted (AA) and non-adapted (NA) cells (representative of ≥3 independent experiments), and corresponding quantification (right; mean fold of NA ± s.e.m., $n$ = 14 images per condition, three independent experiments). Scale bars: 10 μm. **g**, Confocal imaging (left; representative of ≥3 independent experiments) and flow cytometry quantification (right) of the CS surface signal in AA and NA cells (mean fold of NA ± s.e.m., $n$ = 21, seven independent experiments). Scale bars: 10 μm. MFI, mean fluorescence intensity. **h**, Confocal imaging (left; representative of ≥2 independent experiments) and flow cytometry analysis (middle, representative histogram; right, quantification) of the CS surface signal in AA and NA cells after siRNA-mediated knockdown (KD) of CSGALNACT1 (by siRNA#1 and #2) or control siRNA (siCtrl) (mean fold of AA siCtrl ± s.e.m., $n$ = 6 (AA siRNA #2) and $n$ = 12 (all other groups), two or four independent experiments, respectively). Scale bars: 10 μm. Squares indicate zoomed area (**a**,**b**). CS was visualized with the CS-56 antibody (**a**, U3054MG; **g**,**h**) or via scFv clone GD3G7 (**a**, U3047MG; **b**,**d**) and quantified via CS-56-AF488 (**g**,**h**). Significance was determined by two-sided $t$-test (**c**,**f**,**g**) or by one-way analysis of variance (ANOVA) (**h**).

we also observed an enrichment of genes involved in lipid storage and LD biogenesis in LD⁺ versus LD⁻ regions (Fig. 1b). This transcriptional signature was recapitulated in 3D (LD⁺) compared to 2D (LD⁻) cultures (Fig. 1c). Glycans, unlike nucleic acids or proteins, are synthesized without a template, relying on a complex enzymatic 'sugar machinery'[20] (Extended Data Fig. 1b), and we sought to further explore the functional relevance of this signal. Based on consistent overexpression across both LD⁺ tumour regions and spheroids, we identified a 21-gene signature comprising markers of metabolic stress (for example, *CA9*, *CA12*, *VEGFA*), CS biosynthesis (*CHPF*, *CSGALNACT1*, *CHSY1*, *UST*), CSPG core proteins (*BGN*, *CSPG4*, *NCAN*, *VCAN*) and lipid metabolism and LD

formation (*FASN*, *HILPDA*, *PPARD*, *PPARGC1A*, *VLDLR*) (Fig. 1d). Spatial transcriptomics from the Ivy Glioblastoma Atlas Project (IvyGAP)[21] confirmed that this LD⁺/CS⁺ signature was enriched in pseudopalisading regions (Fig. 1e).

We could validate the CS signature at the phenotypic level, as immunostaining showed a prominent CS-rich glycocalyx in LD⁺ spheroid cores (Fig. 2a and Extended Data Fig. 1c), where it co-localized with CA9, a canonical marker of hypoxia and acidosis (Extended Data Fig. 1d)[22]. The pH (low) insertion peptide (pHLIP)[23] was employed to directly assess the pH distribution (Extended Data Fig. 1e,f), showing preferential pHLIP signal in the central, CA9-positive regions of 3D spheroids

(Extended Data Fig. 1g–i). In GBM tissue, we also observed CS enrichment in CA9$^+$/LD$^+$ versus CA9$^-$/LD$^-$ regions (Fig. 2b,c), and we consistently identified a subpopulation exhibiting both LDs and a robust CS-glycocalyx in freshly isolated, patient-derived cultures (PDCs; Fig. 2d and Extended Data Fig. 1j). In contrast, low-grade gliomas lacked this phenotype (Extended Data Fig. 1k), suggesting an association with high-grade malignancy. Moreover, the LD$^+$/CS$^+$ phenotype was preserved in a patient-derived GBM xenograft (Extended Data Fig. 1l). Also, central nervous system (CNS) metastases from kidney, melanoma and lung primaries harboured CS-rich cells in perinecrotic (CA9$^+$/CD31$^-$/LD$^+$/CS$^+$) and perivascular (CA9$^-$/CD31$^+$/LD$^-$/CS$^+$) compartments (Extended Data Fig. 2), showing that the LD$^+$/CS$^+$ phenotype was not restricted to primary brain tumours. These findings highlight CS-glycocalyx accumulation as a hallmark of metabolically challenged regions in aggressive tumours.

## CS-glycocalyx encapsulation as an adaptive response to tumour acidosis

Acidosis and hypoxia are central stressors of the TME, driving aggressive phenotypes and therapy resistance[1,24]. To model acidosis adaptation, we cultured glioma cells at pH 6.4 for 10 weeks, generating acidosis-adapted (AA) lines. Compared to non-adapted (NA, pH 7.4) controls, AA cells showed induction of genes and pathways involved in CS biosynthesis and PG remodelling (Fig. 2e and Extended Data Fig. 3a,b). We found strong upregulation of the CS-initiating enzyme *CSGALNACT1* (~10-fold) and CSPG core proteins, such as *SRGN*, *BGN* and *DCN* (Fig. 2e and Extended Data Fig. 3a). AA cells also displayed elevated expression of LD-related genes (*HILPDA*, *GOS2*; Fig. 2e) and increased LD accumulation (Fig. 2f). Interestingly, confocal imaging and flow cytometry confirmed a pronounced CS-glycocalyx in AA cells (~10-fold increase compared to NA cells), corroborated by antibodies recognizing distinct CS epitopes (Fig. 2g and Extended Data Fig. 3c,d), as well as by biochemical CS disaccharide analysis (Extended Data Fig. 3e–g). This response may be conserved, as acidosis-adapted PANC1 pancreatic cancer cells were similarly enriched for PG-related pathways, PG-related genes and cell-surface CS (Extended Data Fig. 3h–j). To isolate the specific contribution of hypoxia, we next employed short-term (48 h) stress conditions, as long-term hypoxia triggers acidosis and metabolic rewiring[25]. Short-term acidosis was sufficient to activate lipid and PG-related pathways, induce expression of *CSGALNACT1* and other CSPG biosynthetic genes, and to increase cell-surface CS levels (Extended Data Fig. 4a–d), although this was less pronounced than in AA cells (compare with Fig. 2g). In contrast, hypoxia did not upregulate PG-related pathways or CS biosynthetic genes, and failed to induce

cell-surface CS (Extended Data Fig. 4e–g). *CSGALNACT1* was consistently upregulated in LD$^+$ tumour regions and spheroids, as well as in AA cells and short-term acidosis, but not in hypoxia. Notably, CSGALNACT1 operates at a critical decision point in PG biosynthesis by catalysing the first committed step toward CS polymer elongation on a common tetrasaccharide linker (Xyl–Gal–Gal–GlcA) shared by CSPGs and HSPGs (Extended Data Fig. 1b)[26,27]. We performed siRNA-mediated knockdown of CSGALNACT1 in AA cells (Extended Data Fig. 4h), resulting in a marked reduction of cell-surface CS (Fig. 2h). Together, these data reveal acidosis adaptation and CSGALNACT1 as important drivers of the CS-glycocalyx phenotype.

## Cooperative TGF-β and HIF signalling induces CS-glycocalyx remodelling during acidosis adaptation

Transforming growth factor beta (TGF-β) is a known mediator of CSPG remodelling in fibrosis[28,29], and regulates adaptation to tumour acidosis[30]. We found significant enrichment of TGF-β signalling in acidosis as well as in LD$^+$ tumour regions and spheroids, and AA cells showed increased levels of active TGF-β, SMAD2 phosphorylation and SNAIL (Fig. 3a,b and Extended Data Fig. 5a–c). Conditioned media from AA cells, but not NA cells, as well as recombinant TGF-β1 and TGF-β2, induced surface CS in parental GBM cells (Fig. 3c,d and Extended Data Fig. 5d,e). Moreover, inhibition of TGF-β receptors limited acidosis-driven CS-glycocalyx formation (Fig. 3e).

We also found an enrichment of hypoxia-inducible factor (HIF)-associated gene signatures in acidosis as well as in LD$^+$ tumour regions and spheroids (Fig. 3f and Extended Data Fig. 5f). Although, HIFs are central mediators of the hypoxic-acidic TME and cooperate with TGF-β in TME remodelling[31], their direct role in CS-glycocalyx formation remains unexplored. AA cells showed increased HIF-1α and HIF-2α protein expression (Fig. 3g), and cell-surface CS expression was induced by pharmacologic HIF stabilization with dimethyloxalylglycine (DMOG; Fig. 3h,i and Extended Data Fig. 5g,h). Co-stimulation with DMOG and TGF-β further amplified CS levels, comparable to those observed in AA cells (Extended Data Fig. 5i). Moreover, CUT & RUN analysis revealed a genome-wide increase in HIF-1α binding sites in AA versus NA cells, comparable to DMOG treatment (Extended Data Fig. 5j and Supplementary Table 1). Notably, HIF-1α binding sites were primarily gained at promoter regions (<5 kb from the transcription start site, TSS; Extended Data Fig. 5k). Importantly, both acidic adaptation (Fig. 3j) and DMOG treatment (Extended Data Fig. 5l) redirected HIF-1α binding toward promoters of genes related to CS, PG and GAG pathways. This included key genes in CS biosynthesis, where HIF-1α also occupied distal promoter regions (<10 kb from TSS) and other regulatory regions (Fig. 3k,l and Extended Data Fig. 5m,n). Together,

---

**Fig. 3 | Cooperative TGF-β and HIF signalling induces CS-glycocalyx remodelling during acidosis adaptation. a**, Enrichment of 'TGF-β signalling pathway' genes in LD$^+$ versus LD$^-$ GBM tumour areas and U3054MG 3D versus 2D cultures (top), or in U87MG AA versus NA and short-term (48 h) pH 6.4 versus pH 7.4 conditions (bottom) (*n* = 3 biological replicates). **b**, Immunoblotting for active TGF-β, phosphorylated (Ser465/467)/total SMAD2, and SNAIL in U87MG AA and NA cells with (10% FBS) or without (serum-free, SF) exogenous lipids (representative of one or two independent experiments). α-tubulin was used as a loading control. **c**, Confocal imaging of the CS surface signal in U87MG and U3054MG cells treated with/without TGF-β1 (4 ng ml$^{-1}$, 48 h, pH 7.4) (representative of ≥2 independent experiments). Scale bars: 10 μm. **d**, Flow cytometry quantification of the CS surface signal in U87MG treated as in **c** (mean fold of Ctrl ± s.e.m., *n* = 9, three independent experiments). **e**, Confocal imaging of CS surface signal in U87MG cells following short-term acidosis treatment with/without TGFβRi (15 μM, 48 h, pH 6.4) (representative of three independent experiments). Scale bars: 10 μm. **f**, Enrichment of 'hypoxia hallmark' genes in LD$^+$ versus LD$^-$ GBM tumour areas and U3054MG 3D versus 2D cultures (top), or U87MG AA versus NA, and short-term (48 h) pH 6.4 versus pH 7.4 conditions (bottom) (*n* = 3 biological replicates). **g**, Immunoblotting of HIF-1α

and HIF-2α expression in U87MG AA and NA cells (representative of one or two independent experiments). β-actin was used as a loading control. **h**, Confocal imaging of the CS surface signal in U87MG and U3054MG cells treated with/without DMOG (0.5 or 1 mM respectively, 72 h, pH 7.4) (representative of ≥2 independent experiments). Scale bars: 10 μm. **i**, Flow cytometry quantification of the CS surface signal in U87MG and U3054MG cells treated as in **h** (mean fold of Ctrl ± s.e.m., *n* = 9, three independent experiments). **j**, Number of genes related to glycocalyx remodelling with HIF-1α peaks at promoter regions (<5-kb from the transcription start site, TSS) in the indicated subsets (NA-unique, AA-unique, common). **k**, Number of HIF-1α binding sites in the proximity of genes of interest (<5 kb, <10 kb and <100 kb from TSS). **l**, HIF-1α binding sites at the loci of *CSGALNACT1*, in U87MG AA and NA cells. Yellow-shaded regions indicate promoters annotated by the European Promoter Database or regulatory elements defined by ENCODE. Differential peaks: gained (red) or lost (blue) in AA versus NA cells, and invariable (grey). CS surface signal was visualized via CS-56 antibody (**c,e,h**) and quantified via CS-56-AF488 (**d,i**). Significance was determined by BH-adjusted nominal *P* value (**a,f**) or by two-sided *t*-test (**d,i**). *FDR < 0.1; **FDR< 0.05 and ****FDR< 0.001 (**a,f**).

---

these data position CS-glycocalyx remodelling as a key feature of acidic stress adaptation, mediated by cooperative TGF-β and HIF signalling.

## CS-glycocalyx limits lipid scavenging via SDC1 glycan remodelling under acidosis

LD formation is increasingly recognized as a protective sink against toxic lipids in the stressed TME[7], but how lipid influx is modulated to balance de novo lipogenesis and lipid availability to prevent overload remains poorly understood. FA synthase (*FASN*) expression was increased in LD+ tumour regions and spheroids (Fig. 1d). However, FASN inhibitor (FASNi) treatment had no effect on acidosis-driven LD accumulation (Extended Data Fig. 6a). Importantly, supplementation with

serum, low-density lipoprotein (LDL) or EV lipid particles was essential to sustain LD formation under acidosis (Extended Data Fig. 6b–e). Similarly, CS-glycocalyx induction under acidosis depended on extracellular lipid availability (Extended Data Fig. 6d,f–h), indicating that the CS+/LD+ phenotype is independent of FASN and instead relies on extracellular lipids. To further dissect how lipid storage and CS-glycocalyx induction may be linked functionally, we blocked LD formation using the DGAT1 inhibitor A922500 (DGAT1i; Fig. 4a and Extended Data Fig. 6i). LD disruption resulted in a further, compensatory increase in CS-glycocalyx expression in both acidic 2D cultures (Fig. 4b and Extended Data Fig. 6i) and spheroids (Fig. 4c,d). This suggested that CS-glycocalyx may represent an adaptive response

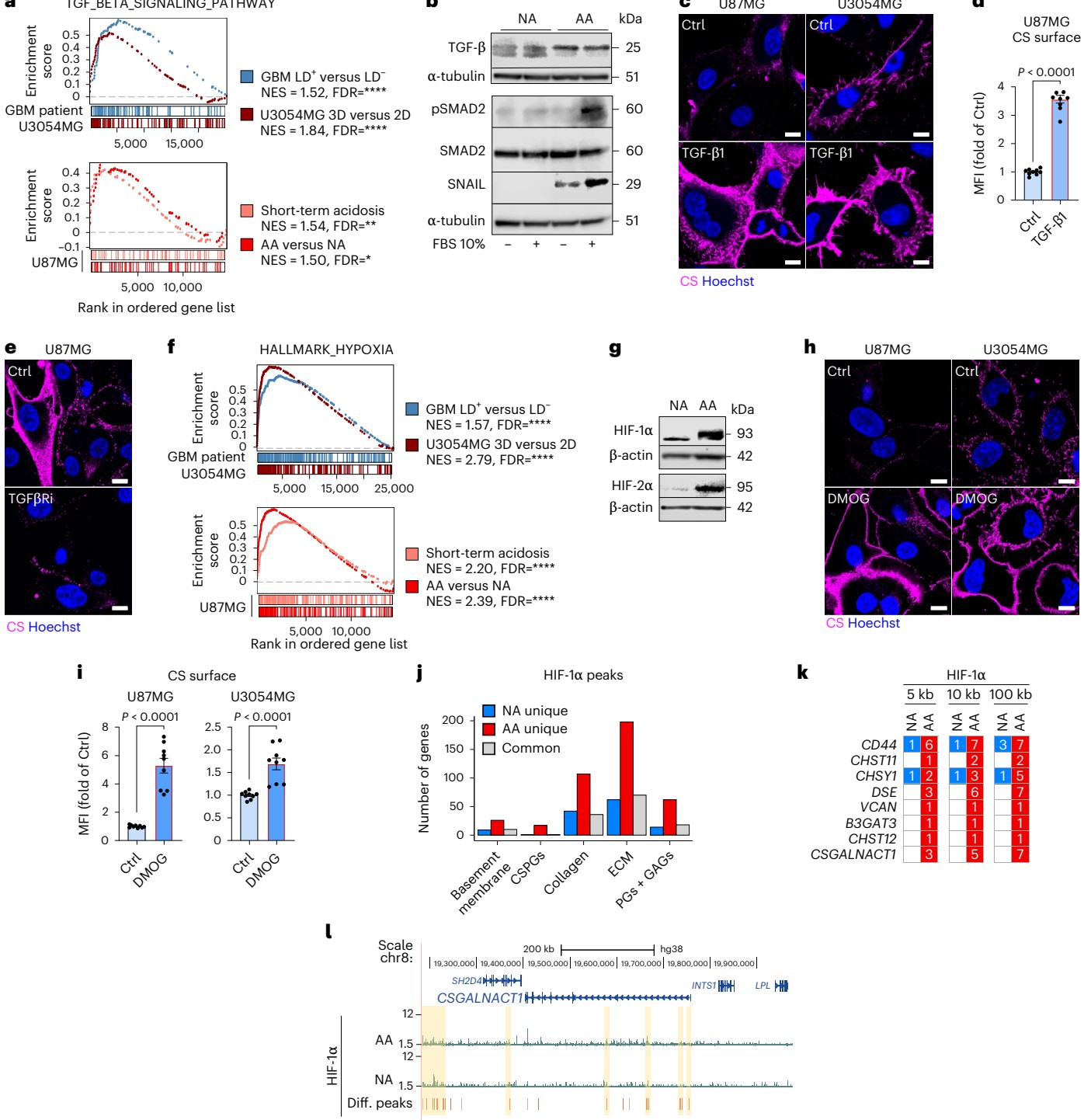

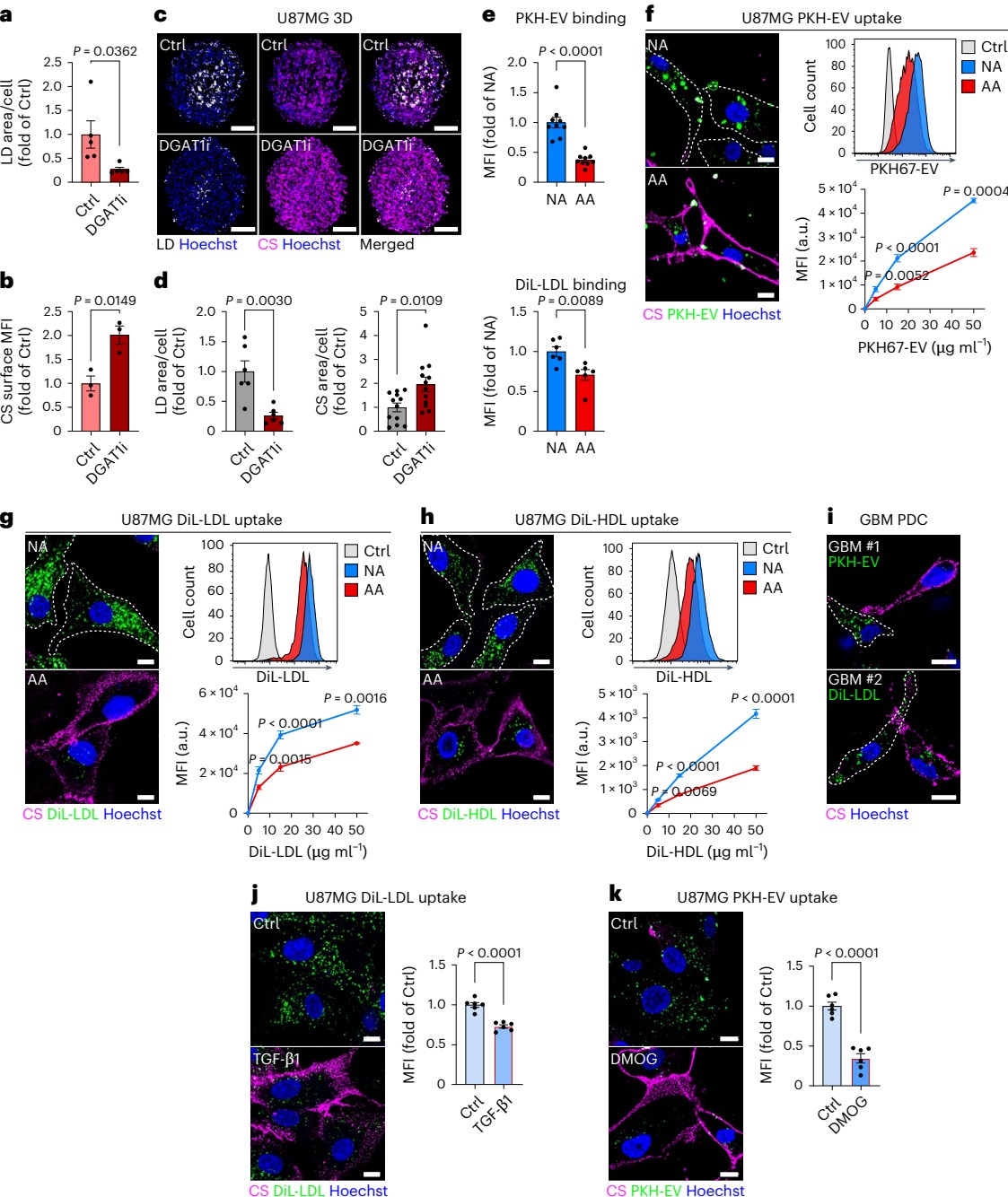

**Fig. 4 | CS-glycocalyx is induced in response to exogenous lipid particles and restricts their uptake under acidic conditions. a**, Quantification of LDs in U87MG cells following treatment with LDL (50 μg ml⁻¹) with/without DGAT1i (10 μM, 48 h, at pH 6.4) (mean fold of Ctrl ± s.e.m., n = 5 images per condition). **b**, Flow cytometry quantification of the CS surface signal in U87MG cells treated as in **a** (mean fold of Ctrl ± s.e.m., n = 3 biological replicates). **c**, Imaging of LDs and CS in U87MG 3D cultures treated with/without DGAT1i (40 μM, seven days) (representative of n ≥ 6 spheroids per condition). Scale bars: 200 μm. **d**, Quantification of LDs (left) and CS (right) from **c** (mean fold of Ctrl ± s.e.m., n = 6 (LDs) and n = 12 (CS) spheroids per condition). **e**, Flow cytometry quantification of cell-surface binding of PKH67-EV (top) or DiL-LDL (bottom) (both 15 μg ml⁻¹) in AA and NA cells (mean fold of NA ± s.e.m., n = 9 (EVs) and n = 6 (LDL), three and two independent experiments, respectively). **f**–**h**, Confocal imaging (left) of CS surface signal and uptake of PKH67-EV (**f**), DiL-LDL (**g**) or DiL-HDL (**h**) (20 μg ml⁻¹, 1 h) in AA and NA cells (representative of ≥3 independent experiments), and corresponding flow cytometry analyses showing representative histograms (15 μg ml⁻¹, 1 h) and dose-dependent quantification of lipid particle uptake (right; mean fold of NA ± s.e.m., n = 9 (EV/LDL 5 and 15 μg ml⁻¹), n = 3 (EV/LDL 50 μg ml⁻¹)

and n = 6 (HDL), representative of ≥2 independent experiments). Dashed lines delineate NA cell borders. Scale bars: 10 μm. **i**, Confocal imaging of CS surface signal and lipid particle uptake (PKH67-EV or DiL-LDL, 50 μg ml⁻¹, 2 h) in freshly resected GBM PDCs (representative of n = 2 individual patients for each lipid source). Dashed lines delineate borders of CS-low cells with high lipid uptake. Scale bars: 10 μm. **j**, Confocal imaging of CS surface signal and DiL-LDL uptake (40 μg ml⁻¹, 1 h) (left; representative of two independent experiments), and corresponding flow cytometry quantification of DiL-LDL uptake (15 μg ml⁻¹, 1 h) (right), in U87MG cells pre-treated with/without exogenous TGF-β1 (4 ng ml⁻¹, 48 h, pH 7.4) (mean fold of Ctrl ± s.e.m., n = 6, two independent experiments). Scale bars: 10 μm. **k**, Confocal imaging of CS surface signal and PKH67-EV uptake (50 μg ml⁻¹, 1 h) (left; representative of two independent experiments), and corresponding flow cytometry quantification of PKH67-EV uptake (15 μg ml⁻¹, 1 h) (right), in U87MG cells pre-treated with/without DMOG (0.5 mM, 72 h, pH 7.4) (mean fold of Ctrl ± s.e.m., n = 6, two independent experiments). Scale bars: 10 μm. CS surface signal was quantified via CS-56-AF488 (**b**) and visualized via CS-56 antibody (**c**,**f**–**i**, GBM #1; **j**,**k**) or scFv clone GD3G7 (**i**, GBM #2). Significance was determined by two-sided t-test (**a**,**b**,**d**–**h**,**j**,**k**).

to excess or unmetabolized lipids in the acidic microenvironment. In support of this, AA cells displayed reduced binding (Fig. 4e) and uptake of EVs and LDL (Fig. 4f,g), as well as apoE-containing high-density lipoprotein (HDL; Fig. 4h and Extended Data Fig. 6j). This phenotype was also observed after short-term acidosis (Extended Data Fig. 6k,l), and patient tumour samples showed an inverse correlation between CS-glycocalyx levels and lipid uptake (Fig. 4i). Notably, overall endocytic activity was increased in AA versus NA cells (Extended Data Fig. 6m,n), and overall expression and sulfation of HSPGs, widely recognized as key mediators of lipoprotein and EV scavenging[15,32–35], remained intact in AA cells (Extended Data Fig. 6o,p). Finally, inducing the CS-glycocalyx with TGF-β or DMOG in cells cultured at pH 7.4 mimicked the lipid uptake defect observed under acidosis (Fig. 4j,k and Extended Data Fig. 6q–s). These data support a model in which CS-glycocalyx encapsulation restricts access to extracellular lipids during metabolic stress.

To directly assess the role of the CS-glycocalyx as a barrier to lipid scavenging, we first treated AA and NA cells with sodium chlorate, which inhibits the HS and CS sulfation essential for ligand binding (Extended Data Fig. 7a). Consistent with compromised HSPG function in NA cells, sodium chlorate treatment diminished EV binding and uptake to levels observed in AA cells (Fig. 5a). However, in AA cells, sodium chlorate had no impact (Fig. 5a), indicating that residual lipid particle uptake proceeds via HSPG-independent mechanisms. We next employed enzymatic, genetic and pharmacological strategies to specifically dismantle the CS-glycocalyx. Surface CS chains were effectively removed either by exogenous application of chondroitinase ABC/AC1 lyases (CS'ase) (Fig. 5b, left, and Extended Data Fig. 7b) or by U87MG cells stably expressing chondroitinase ABC (ChABC) (Fig. 5c, left, and Extended Data Fig. 7c). Both approaches restored EV binding in acidic cells (Fig. 5b,c, middle panels); intriguingly, this did not translate into similarly enhanced EV uptake (Fig. 5b,c, right panels). These findings suggest that, although the CS-glycocalyx imposes a barrier to lipid particle binding, specific HSPG-mediated scavenging functions are not reinstated upon CS-glycocalyx removal alone. Strikingly, inhibition of CS biosynthesis by CSGALNACT1 knockdown restored lipid scavenging in AA cells (Fig. 5d,e and Extended Data Fig. 7d). Similarly, treatment with the CSPG inhibitor 4-nitrophenyl β-D-xylopyranoside (CSi), which competes with CS substitution onto core proteins[36], fully restored lipid uptake, matching the levels observed in NA cells (Fig. 5f,g).

These findings prompted us to focus on syndecan-1 (SDC1), a key cell-surface HSPG implicated in lipid particle scavenging[33,37,38]. High-resolution imaging revealed robust co-internalization of SDC1 with lipid particles into endocytic vesicles in NA cells (Extended Data Fig. 7e). Moreover, consistent with SDC1-dependent scavenging[14,39], EV uptake by NA cells mainly followed membrane raft-mediated endocytosis (Extended Data Fig. 7f). Conversely, residual EV uptake in AA cells was predominantly routed through macropinocytosis (Extended Data Fig. 7f). Notably, SDC1 is a hybrid PG that can variably carry CS chains, particularly under TGF-β signalling[40], raising the possibility of perturbed HS substitution of SDC1 in AA cells. Indeed, despite comparable total SDC1 levels between NA and AA cells (Extended Data Fig. 7g), HS-substituted SDC1 was nearly absent in AA cells (Fig. 5h), which was associated with decreased SDC1 surface presentation and internalization (Fig. 5i,j and Extended Data Fig. 7h,i). Additionally, SDC1 localization shifted from vesicular compartments in NA cells to a diffuse distribution in AA cells (Extended Data Fig. 7j). Notably, CSi treatment both restored SDC1 internalization (Fig. 5j) and reinstated its vesicular localization in AA cells (Extended Data Fig. 7j). Collectively, these data delineate a dual mechanism by which CS induction impairs lipid scavenging under acidic stress: (1) by establishing a barrier to lipid particle binding and (2) by disrupting the SDC1-HS scavenging function (Extended Data Fig. 7k).

## CS-glycocalyx functions as a protective shield preventing lipid overload and cytotoxicity during acidosis adaptation

We hypothesized that the CS-glycocalyx, by restricting lipid scavenging, serves to maintain lipid homeostasis and prevent lipotoxicity in acidosis. To test this, we initially challenged U87MG and primary GBM cultures to high concentrations of lipid particles simultaneously with the introduction of acidosis, that is, prior to a fully established CS-glycocalyx. This led to a progressive cytotoxic response over time (Fig. 6a,b), as well as growth arrest (Extended Data Fig. 8a,b). Notably, these effects were specific to the combination of acidosis and high-dose lipids, as neither acidosis alone nor lipids at pH 7.4 induced comparable cytotoxic effects (Fig. 6a,b). Inhibition of CS-glycocalyx formation using the CS biosynthesis competitor CSi further sensitized cells to the early antiproliferative effects of lipid particles (Extended Data Fig. 8c). Moreover, CSi enhanced lipid-induced cytotoxicity at acidic pH, with lower lipid doses being sufficient to trigger cell death (Fig. 6c and Extended Data Fig. 8d). Again, these effects were not observed at pH 7.4, underscoring a context-dependent protective role of CS-glycocalyx. Supporting this, CSGALNACT1 knockdown resulted in enhanced lipid-induced cytotoxicity in AA cells, a response absent in NA cells (Fig. 6d and Extended Data Fig. 8e), and dependent on extracellular lipids (Fig. 6e and Extended Data Fig. 8f). To further investigate the role of CS-glycocalyx in a model where acidosis progressively develops, we examined the effects of CSi treatment in 3D cultures. We first could confirm a striking reduction in CS-glycocalyx in the acidic spheroid core with CSi treatment (Extended Data Fig. 8g,h). Interestingly, in parallel, we found a significant LD induction in the spheroid core (Fig. 6f,g and Extended Data Fig. 8i,j). This compensatory upregulation of LDs led us to speculate that the CS-glycocalyx shield and the LD intracellular sink cooperatively mediate lipid homeostasis during acidosis adaptation, preventing lipotoxicity. Consistent with this, CSi treatment led to dose-dependent inhibition of spheroid growth (Fig. 6h and Extended Data Fig. 8k), although the response was predominantly cytostatic.

We next aimed to understand whether acidosis-induced CS-glycocalyx was associated with a more aggressive phenotype, and whether this could be targeted in vivo. AA compared to NA spheroids exhibited enhanced invasiveness (Extended Data Fig. 8l), and AA cells displayed accelerated growth and reduced survival relative to NA cells in a mouse xenograft model (Extended Data Fig. 8m,n). Similarly to patient GBM, AA-derived tumours displayed prominent CS-glycocalyx enrichment that overlapped with CA9 and LDs (Extended Data Fig. 8o). Given its physicochemical properties and high polarity, CSi is unlikely to cross the blood–brain barrier (BBB). To enable local delivery, we employed osmotic pumps for continuous intracerebral administration over seven days (Fig. 6i). Notably, this treatment was sufficient to prolong survival in mice bearing AA xenografts (Fig. 6j). Together, these findings reveal that the CS-glycocalyx functions in concert with LDs to prevent lipid overload and associated cytotoxicity during acidosis adaptation.

## Dual targeting of CS-glycocalyx and LD formation synergistically disrupts lipid homeostasis and compromises survival of acidic cancer cells

We next explored whether combined targeting of the CS-glycocalyx and LD formation could provide a strategy to effectively destabilize the acidic tumour niche (Extended Data Fig. 9a). DGAT1i treatment alone induced some cytotoxicity under acidosis, which was markedly potentiated by concomitant CSi treatment (Fig. 7a,b and Extended Data Fig. 9b). This synergistic effect was strictly dependent on acidic conditions and the presence of extracellular lipids (Extended Data Fig. 9c). Supporting these findings, CSGALNACT1 knockdown similarly enhanced DGAT1i-induced cytotoxicity in AA cells (Fig. 7c and Extended Data Fig. 9d, left), while sparing NA cells (Extended Data Fig. 9d, right). siRNA treatment can lead to

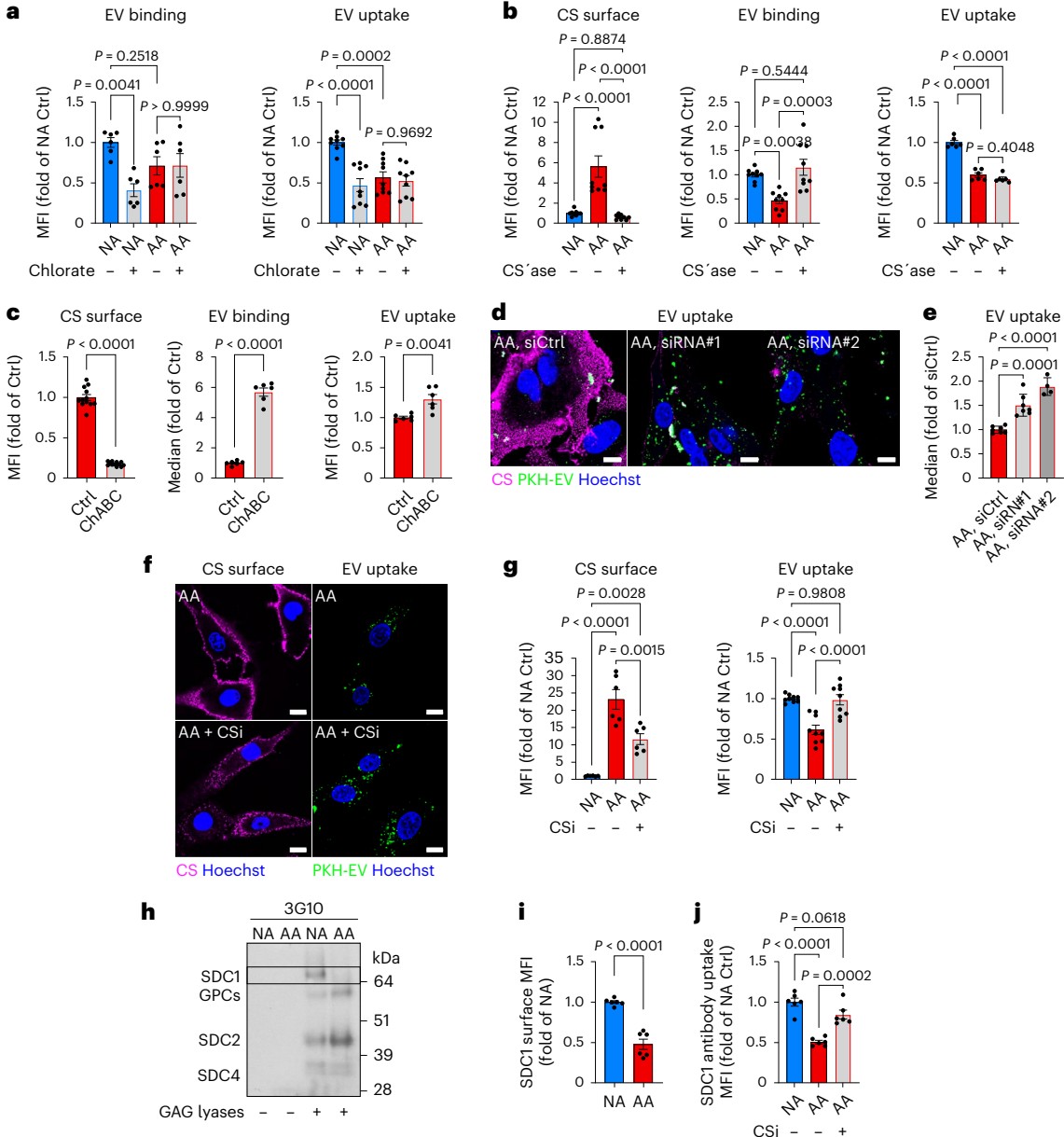

**Fig. 5 | Acidosis-induced CS-glycocalyx restricts lipid uptake through encapsulation and SDC1 glycan remodelling. a**, Flow cytometry quantification of PKH67-EV cell-surface binding (60 µg ml⁻¹; left) and uptake (15 µg ml⁻¹, 1 h; right), in U87MG AA and NA cells after sodium chlorate pre-treatment (chlorate, 25 mM, 24 h) (mean fold of NA ± s.e.m., $n = 6$ (EV binding) and $n = 9$ (EV uptake), two and three independent experiments, respectively). **b**, Flow cytometry quantification of CS surface signal (left), PKH67-EV cell-surface binding (15 µg ml⁻¹; middle) and PKH67-EV uptake (15 µg ml⁻¹, 1 h; right), in U87MG AA cells after ChABC/AC1 lyases digestion (CS'ase, 6 h) (mean fold of AA or NA ± s.e.m., $n = 9$ (CS surface and EV binding) and $n = 6$ (EV uptake), three and two independent experiments, respectively). **c**, Flow cytometry quantification of CS surface signal (left), PKH67-EV cell-surface binding (30 µg ml⁻¹; middle) and PKH67-EV uptake (30 µg ml⁻¹, 1 h; right), in ChABC-expressing U87MG cells under acidic conditions (48 h, pH 6.4) (mean fold of Ctrl ± s.e.m., $n = 12$ (CS surface) and $n = 6$ (EV binding and uptake), four and two independent experiments, respectively). **d,e**, Confocal imaging (**d**) of CS surface signal and PKH67-EV uptake (40 µg ml⁻¹, 1 h) (representative of ≥2 independent experiments), and corresponding flow cytometry quantification (**e**) of PKH67-EV uptake (20 µg ml⁻¹, 1 h), in U87MG AA cells pre-treated with control siRNA (siCtrl) or two different siRNAs targeting CSGALNACT1 (siRNA#1 and #2) (mean fold of siCtrl ± s.e.m.,

$n = 4$ (siRNA#2) and $n = 7$ (all other groups), two independent experiments). Scale bars: 10 µm. **f,g**, Confocal imaging (**f**) of the CS surface signal and PKH67-EV uptake (40 µg ml⁻¹, 1 h) (representative of two independent experiments), and corresponding flow cytometry quantification (**g**) of CS surface signal (left) and PKH67-EV uptake (15 µg ml⁻¹, 1 h; right), in U87MG NA and AA cells pre-treated or not with CSi (2.5 mM, 48 h) (mean fold of NA Ctrl ± s.e.m., $n = 6$ (CS surface) and $n = 9$ (EV uptake), two and three independent experiments, respectively). Scale bars: 10 µm. **h**, Total PGs isolated from U87MG AA and NA cells were treated (+) or not (−) with GAG lyases (HS III and ABC lyase). Core proteins were then separated by SDS–PAGE and HSPGs visualized by immunoblotting with 3G10 anti-HS stub antibody. The band corresponding to SDC1 was absent in AA cells (signal highlighted within the black lines). Non-digested PGs (lanes 1 and 2) showed no signal, confirming 3G10 specificity (representative of two independent experiments). **i,j**, Flow cytometry quantification of cell-surface SDC1 (**i**) (mean fold of NA ± s.e.m., $n = 6$, two independent experiments), and anti-SDC1 antibody uptake (**j**) (mean fold of NA ± s.e.m., $n = 6$, two independent experiments), in U87MG AA and NA cells treated as in **f**. CS surface signal was quantified via CS-56-AF488 (**b,c,g**) and visualized via CS-56 antibody (**d,f**). Significance was determined by one-way ANOVA (**a,b,e,g,j**) or two-sided *t*-test (**c,i**).

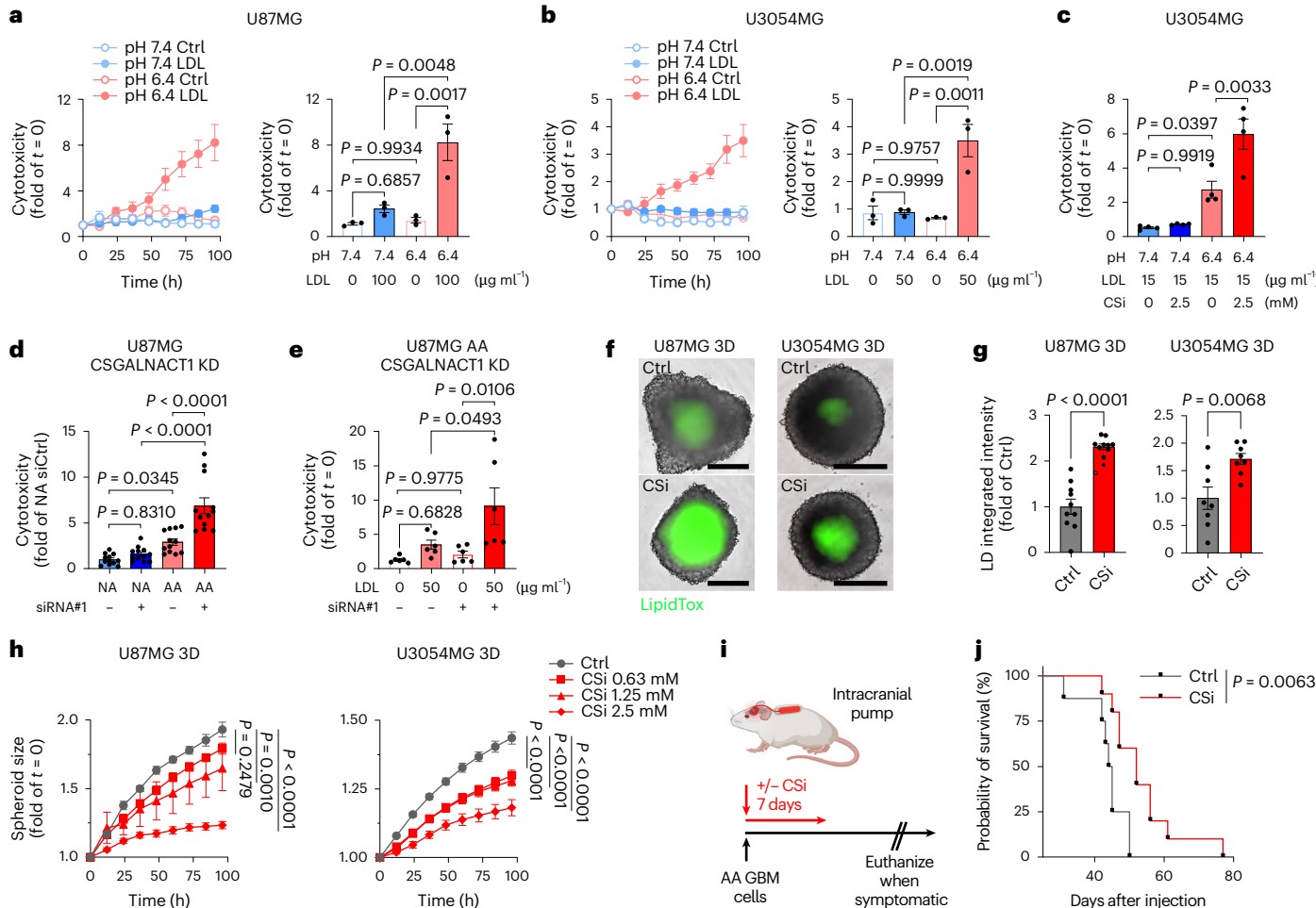

**Fig. 6 | CS-glycocalyx functions as a protective shield preventing lipid overload and cytotoxicity during acidosis adaptation. a,b**, Cytotoxicity over time (left), and corresponding quantification at 96 h (right), in U87MG (**a**) or U3054MG (**b**) cells challenged with/without high-dose LDL (pH 6.4 or 7.4), as indicated (mean fold of $t = 0 \pm$ s.e.m., $n = 3$ biological replicates). **c**, Cytotoxicity quantification at 96 h in U3054MG cells treated with/without CSi and low-dose LDL (pH 6.4 or 7.4), as indicated (mean fold of $t = 0 \pm$ s.e.m., $n = 4$ biological replicates). **d**, Cytotoxicity quantification at 72 h in U87MG AA and NA cells (10% FBS) after siRNA-mediated CSGALNACT1 KD (mean fold of NA siCtrl $\pm$ s.e.m., $n = 12$, two independent experiments). **e**, Cytotoxicity quantification at 72 h in U87MG AA cells treated with low-dose LDL after siRNA-mediated CSGALNACT1 KD (mean fold of $t = 0 \pm$ s.e.m., $n = 6$, two independent experiments). **f**, IncuCyte images of LipidTox accumulation in U87MG and U3054MG spheroids after treatment with/without CSi (1.25 mM, 72 h) (representative of $n \geq 8$ spheroids/condition). Scale

bars: 400 µm. **g**, Quantification of **f** (mean fold of Ctrl $\pm$ s.e.m., $n = 10$ (U87MG) and $n = 8$ (U3054MG) spheroids/condition, two or one independent experiments, respectively). **h**, Spheroid size over time in U87MG and U3054MG 3D cultures treated with/without CSi, as indicated (mean fold of $t = 0 \pm$ s.e.m., $n = 8$ (U87MG Ctrl and CSi 2.5 mM; U3054MG) and $n = 4$ (U87MG CSi 0.63 and 1.25 mM) spheroids/condition, two or one independent experiments, respectively). **i**, Experimental design for local CNS delivery of CSi via osmotic pumps over seven days. **j**, Kaplan–Meier survival curves from an orthotopic U87MG AA xenograft model, either treated with sham pump (Ctrl, $n = 8$ mice) or treated with CSi (2.5 mM, $n = 10$ mice). Data in **a**–**h** were acquired by IncuCyte live-cell imaging. Significance was determined by one-way ANOVA (**a**–**e**), two-sided $t$-test (**g**), two-way ANOVA (**h** (at 96 h)) or log-rank (Mantel–Cox) test (**j**). Illustration **i** was created with BioRender.com.

GPX4 upregulation and sensitization to ferroptotis[41]. However, siRNA-mediated CSGALNACT1 knockdown had no apparent stimulatory effect on GPX4 expression (Extended Data Fig. 9e). The combinatorial vulnerability of CS-glycocalyx and LD inhibition extended to several spheroid models (Fig. 7d,e and Extended Data Fig. 9f), as well as AA cell-derived spheroid invasiveness (Fig. 7f). Together, these results suggest that simultaneous disruption of CS-glycocalyx and LD formation creates a metabolic vulnerability in acidic tumour cells by uncoupling lipid uptake control from lipid detoxification.

## Combined inhibition of CS-glycocalyx and LD formation triggers ferroptosis in acidosis

Ferroptosis is characterized by excessive lipid peroxidation[42]. We hypothesized that the CS-glycocalyx acts as a critical protective barrier against ferroptosis in the acidic TME. To test this, we employed C11-BODIPY[581/591], a fluorescent lipid peroxidation sensor, and observed significantly increased lipid peroxidation upon combined CSi and DGAT1i treatment in acidic conditions (Fig. 8a and Extended Data Fig. 9g). This was accompanied by pronounced oxidative lipid damage, which was effectively suppressed by alpha-tocopherol (vitamin E), a lipophilic antioxidant (Extended Data Fig. 9h), and associated cytotoxicity in 2D cultures and spheroids (Extended Data Fig. 9i,j). The combined cytotoxicity of CSi and DGAT1i was abrogated by ferrostatin-1 or liproxstatin-1[43,44] (Fig. 8b,c and Extended Data Fig. 10a), confirming ferroptosis as the underlying mechanism. To corroborate these findings, we included inhibitors of apoptosis (QVD), autophagy (3-MA) and necroptosis (Nec-1s)[45], showing that only QVD reduced cytotoxicity, whereas none of the inhibitors restored cell density (Extended Data Fig. 10b–d). The QVD effects align with recent evidence that caspases can modulate ferroptotic cytotoxicity downstream of

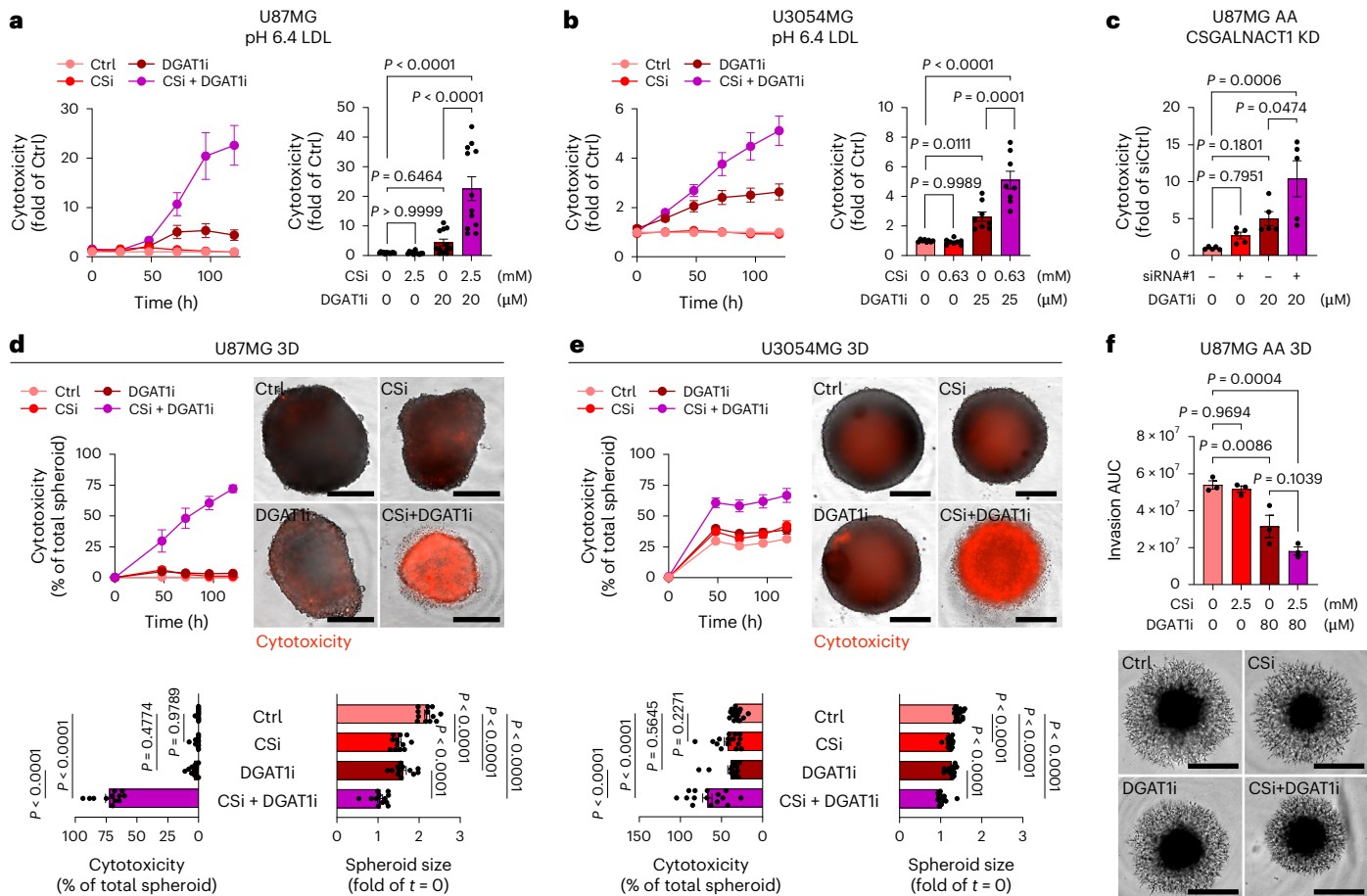

**Fig. 7 | Dual targeting of CS-glycocalyx and LD formation synergistically disrupts lipid homeostasis of acidic cancer cells. a**,**b**, Cytotoxicity over time (left), and corresponding quantification at 120 h (right), in U87MG (**a**) and U3054MG (**b**) cells treated with CSi and/or DGAT1i at pH 6.4 in the presence of low-dose LDL, as indicated (mean fold of LDL Ctrl ± s.e.m., $n$ = 12 (U87MG) and $n$ = 8 (U3054MG), three or two independent experiments, respectively). **c**, Cytotoxicity quantification at 120 h of combined effect of siRNA-mediated CSGALNACT1 KD and DGAT1i treatment in U87MG AA cells cultured with low-dose LDL (mean fold of siCtrl ± s.e.m., $n$ = 5, two independent experiments). **d**,**e**, Cytotoxic effect of CSi (2.5 mM) and/or DGAT1i (80 μM) treatment in U87MG

(**d**) and U3054MG (**e**) 3D cultures. Cytotoxicity over time (top left), IncuCyte images at 120 h (top right), and corresponding quantification of cytotoxicity and spheroid size at 120 h (bottom) (mean ± s.e.m., $n$ = 12 (U87MG) and $n$ = 15 (U3054MG) spheroids per condition, three and four independent experiments, respectively). Scale bars: 400 μm. **f**, Spheroid invasion area (AUC, 0–96 h) quantification (top), and IncuCyte images at 96 h (bottom), in U87MG AA 3D cultures treated with CSi (2.5 mM) and/or DGAT1i (80 μM) (AUC ± s.e.m., $n$ = 3 spheroids per condition, representative of three independent experiments). Scale bars: 800 μm. Data in **a**–**f** were acquired by IncuCyte live-cell imaging. Significance was determined by one-way ANOVA (**a**–**f**).

lipid peroxidation[46], and a crosstalk between apoptotic and ferroptotic pathways[47]. Moreover, CSi and DGAT1i combination treatment was associated with extensive mitochondrial fragmentation and oxidative stress, effects that were significantly reduced by ferroptosis blockade (Fig. 8d,e and Extended Data Fig. 10e). Notably, these effects required the presence of extracellular lipids (Extended Data Fig. 10f) and were strictly dependent on acidic conditions (Extended Data Fig. 10g), underscoring the specificity of this ferroptotic vulnerability to the acidic, lipid-rich TME. Finally, we assessed the combination therapy in the aggressive AA cell-derived xenograft model (Fig. 8f). Under these conditions, we examined whether the CSi dosage could be reduced when combined with DGAT1i. CSi monotherapy again had a survival effect, although the lower concentration did not reach statistical significance ($P$ = 0.1421), but DGAT1i alone showed no effect (Fig. 8f). However, the combination of CSi and DGAT1i significantly extended survival compared to controls (Fig. 8f). This was accompanied by increased tumour cell death (Fig. 8g), which overlapped with markers associated with ferroptosis, including malondialdehyde (MDA) and SLC7A11 (Fig. 8h and Extended Data Fig. 10h). Together, these data establish that CS-glycocalyx and LDs cooperatively function to limit ferroptosis in acidic cancer cells. Their combined inhibition unleashes a

ferroptotic vulnerability that may be therapeutically exploited to target the lipid-stressed tumour niche (Extended Data Fig. 10i).

## Discussion

We have identified a glycan-mediated response to tumour acidosis in which intracellular LD accumulation is coupled to the formation of a CS-enriched glycocalyx. Together, these features constitute a bipartite adaptation: LDs buffer toxic lipids internally, while the CS-rich glycocalyx forms an external barrier that restricts lipid particle uptake and limits ferroptosis.

CS restructuring may be a more general adaptive response in cancer, as recently supported by CS-glycocalyx-mediated resistance during androgen receptor pathway inhibition in prostate cancer[48]. Importantly, our findings, together with earlier studies, highlight the dynamic and context-dependent role of PGs in regulating lipid uptake. Under acute environmental stress (2–6 h) or perturbed GPX4-mediated antioxidant defences, HSPG-mediated lipid uptake supports cellular adaptation[14,16,17]. In contrast, we show that persistent stress triggers a glycan switch that drives the formation of a CS-rich glycocalyx, which acts as a barrier to extracellular lipid access and enables evasion of ferroptosis.

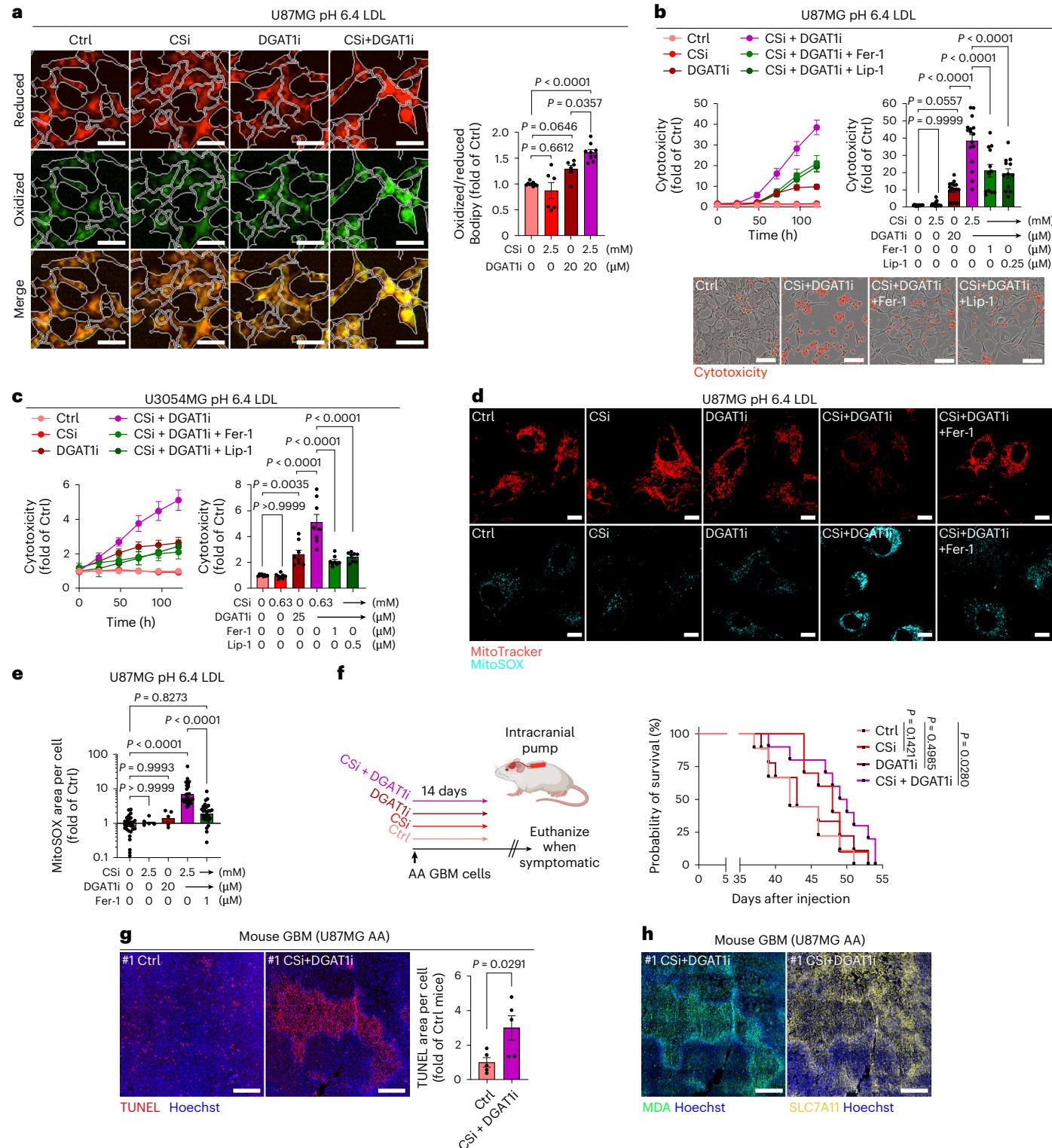

We demonstrate a specific role of CSGALNACT1 that dictates CS substitution on PGs. The choice between HS and CS attachment onto proteins reflects a regulated competition between the initiating enzymes[26]. The induction of CSGALNACT1 probably outcompetes the more sequence-restricted HS-initiating enzymes in hybrid PGs. Together, acidosis orchestrates a glycan switch in which SDC1, a hybrid CS/HSPG, is depleted of HS to restrict lipid particle influx. We employed EVs, which are physiologically relevant lipid carriers in the CNS[49], in parallel with LDL and HDL to probe the broader principle of lipid particle uptake in acidosis. Although neither LDL nor

HDL cross an intact BBB, increased permeability and abnormal transcytosis, particularly in hypoxic/acidic tumour areas, may be more permissive[50], as supported by the leakage of GBM-derived EVs into the circulation[51]. Our data demonstrate that CS-glycocalyx induction suppresses the uptake of multiple structurally distinct lipid particles that rely on SDC1–HSPG. Notably, SDC1–HSPG also mediates scavenging of apoE-containing lipoproteins[34,38,52], supporting the notion that HDL-like, apoE-containing lipid particles, which dominate in astrocytes and microglia[53], use the same uptake machinery. The potential contribution of circulating lipoproteins to the GBM ecosystem as well

**Fig. 8 | Combined inhibition of CS-glycocalyx and LD formation triggers lipid peroxidation and ferroptotic cell death in acidic cancer cells.**
**a**, IncuCyte images (left), and corresponding quantification (right) of cellular lipid peroxidation, measured as the ratio of oxidized to reduced Bodipy signal per cell, in U87MG cells treated with CSi and/or DGAT1i at pH 6.4 in the presence of low-dose LDL, as indicated (mean fold of Ctrl ± s.e.m., $n = 9$ (Ctrl and CSi + DGAT1i) and $n = 6$ (CSi and DGAT1i), from three and two independent experiments, respectively). Scale bars: 50 μm. **b,c**, Cytotoxicity over time (left), and corresponding quantification at 120 h (right), in U87MG (**b**) and U3054MG (**c**) cells treated as in **a** with/without the addition of ferrostatin-1 (Fer-1) or liproxstatin-1 (Lip-1), as indicated (mean fold of Ctrl ± s.e.m. $n = 12$ (U87MG, groups with Fer-1 and Lip-1), $n = 16$ (U87MG, all other groups) and $n = 8$ (U3054), from three, four or two independent experiments, respectively). In **b** (bottom), IncuCyte images at 120 h are shown. Scale bars: 100 μm. **d**, Confocal imaging of U87MG cells treated as in **b** visualizing mitochondria integrity by MitoTracker Red after 30 h of treatment (top), or peroxidized lipids by MitoSOX after 26 h of treatment (bottom) (representative of two independent experiments). Scale bars: 10 μm. **e**, Corresponding quantification of the MitoSOX signal from **d** (mean

fold of Ctrl ± s.e.m., $n = 28$ images per group for all groups except CSi and DGAT1i, where $n = 5$ per group, two and one independent experiments, respectively). **f**, Experimental design (left) of local CNS delivery of CSi and/or DGAT1i through osmotic pumps over 14 days and Kaplan−Meier survival curves (right) from the orthotopic U87MG AA xenograft model, either treated with control sham pump (Ctrl, $n = 9$) or treated with CSi (1.25 mM, $n = 10$), DGAT1i (80 μM, $n = 9$) or CSi + DGAT1i ($n = 10$). **g**, Fluorescence imaging of TUNEL staining in the orthotopic AA xenograft model treated with control sham pump (Ctrl) or the combination of CSi (1.25 mM) and DGAT1i (80 μM) (left; representative of $n = 5$ mice per group), and corresponding quantification (right; mean of Ctrl ± s.e.m., $n = 5$ mice per group with 10–23 separate areas per mouse covering at least 50% of the tumour area). Scale bars: 500 μm. **h**, Fluorescence imaging of ferroptosis-associated markers, MDA and SLC7A11, in consecutive sections of the same area of mouse #1 CSi + DGAT1i in **g** (representative of $n = 3$ mice). Scale bars: 500 μm. Data in **a**–**c** were acquired by IncuCyte live-cell imaging. Significance was determined by one-way ANOVA (**a**–**c**,**e**), by log-rank (Mantel−Cox) test (**f**) or by two-sided $t$-test (**g**). Illustration **f** was created with BioRender.com.

as astrocyte-derived HDL particles remains an important question for future studies. Moreover, the possibility that other HSPG-dependent ligands[54] are also hindered by CS-glycocalyx should be further explored. Notably, abnormal insulin and FA exposure of hepatocytes has previously been shown to induce the exchange of CS for HS on SDC1, resulting in decreased affinity for lipoprotein particles[55]. In this Article we provide a direct demonstration that site-specific glycosylation remodelling governs nutrient acquisition in cancer.

We find that CS-glycocalyx formation is driven by the coordinated action of HIF and TGF-β. Cooperative interactions between HIF and TGF-β signalling have previously been reported, driving extracellular matrix (ECM) reorganization and tumour progression[56,57]. Notably, renal cell carcinoma, which exhibits constitutive HIF activation and LD accumulation, also overexpresses TGF-β as well as CSPGs[58–60]. However, a direct role of HIFs in the regulation of CS-glycocalyx formation has not been described previously. Our data provide evidence that HIF-1α binds to the promoters of genes related to PG function and GAG biosynthesis in response to acidic adaptation. The precise mechanisms by which TGF-β and HIFs cooperate to remodel the stressed TME remain an important area for future investigation.

We also observed CS enrichment in CA9⁻/LD⁻/CD31⁺ regions, raising the possibility that CS remodelling contributes to the dysfunctional vasculature in GBM. Notably, recent work in mice revealed that the brain endothelial glycocalyx undergoes shifts in GAGs (including CS and HS) during ageing[61]. Such glycocalyx alterations may affect barrier leakiness, immune cell infiltration and the perivascular invasion routes of GBM cells. Future studies should determine whether CS accumulates in the endothelial glycocalyx or is associated with perivascular pericytes, potentially under the influence of TGF-β, and whether its abundance distinguishes GBM from healthy brain and low-grade glioma vasculature.

Feron and collaborators reported that LD accumulation can promote a mesenchymal-like invasive phenotype in acidic cancer cells[30]. Extending this concept, the same group revealed that exogenous polyunsaturated FAs (PUFAs) induce lipid peroxidation and ferroptosis[62]. Others have shown that LDs can mitigate lipid peroxidation and reactive oxygen species (ROS) accumulation in acidic osteosarcoma cells[63], and DGAT1 inhibition demonstrated promising effects in a subcutaneous GBM model[19]. Although previous studies have demonstrated that peroxidation of $n$−3 and $n$−6 PUFAs can promote ferroptosis in acidosis[62], we introduce the concept that cancer cells fine-tune their balance between environmental lipid supply and intracellular storage into LDs. DGAT1 targeting alone further amplified the insulating effect of the CS-glycocalyx, resulting in compensatory inhibition of extracellular lipid scavenging. Glycocalyx remodelling and lipid detoxification thus act in concert to regulate ferroptotic sensitivity. These insights

open alternative avenues for therapeutic strategies whereby concurrent disruption of LDs and the CS-glycocalyx could be particularly effective when combined with interventions that increase the dietary supply and peroxidation of PUFAs.

Extracranial tumour models do not recapitulate the BBB and tissue-specific properties of the brain, posing a general challenge for translational efforts in GBM. We employed orthotopic tumour cell injections but were limited by the technical constraints of achieving sustained, local drug delivery via osmotic pumps. The future development of BBB-permeable CS and DGAT inhibitors or strategies for transient BBB opening will be essential to advance this therapeutic concept in vivo. Nonetheless, the concordance between patient tumour data and human PDC-derived in vitro and primary 3D models provides strong support for the relevance of the CS-glycocalyx in human GBM.

In summary, we uncover a stress-induced glycosylation program that governs lipid uptake, storage and survival in acidic tumours. These findings define glycan remodelling as a core determinant of metabolic plasticity and highlight the glycocalyx as a targetable shield sustaining tumour fitness under hostile conditions.

## Online content

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

[1]Department of Clinical Sciences, Division of Oncology, Lund University, Lund, Sweden. [2]Department of Clinical Sciences, Division of Neurosurgery, Lund University, Lund, Sweden. [3]Department of Medical and Translational Biology, Section of Molecular Medicine, Umeå University, Umeå, Sweden. [4]Wallenberg Centre for Molecular Medicine (WCMM), Umeå University, Umeå, Sweden. [5]Department of Medical Biochemistry and Microbiology, Science for Life Laboratory, Uppsala University, Uppsala, Sweden. [6]Department of Experimental Medical Science, Lund University, Lund, Sweden. [7]Department of Biochemistry, Radboud Institute for Molecular Life Sciences, Radboud University Medical Centre, Nijmegen, The Netherlands. [8]Department of Immunology, Genetics and Pathology, Science for Life Laboratory, Uppsala University, Uppsala, Sweden. [9]Science for Life Laboratory, Uppsala University, Uppsala, Sweden. [10]Glycobiology Research and Training Center, University of California San Diego, La Jolla, CA, USA. [11]Department of Cellular and Molecular Medicine, University of California San Diego, La Jolla, CA, USA. [12]Department of Neurosurgery, Skåne University Hospital, Lund, Sweden. [13]Department of Hematology, Oncology and Radiophysics, Skåne University Hospital, Lund, Sweden. ✉e-mail: mattias.belting@med.lu.se

## Methods

### Ethical statement

All research involving human and animal materials in this study was conducted in accordance with relevant ethical regulations.

### Compounds and antibodies

The following compounds were used: sodium chlorate (044408) from Alfa Aesar; cholera toxin subunit B-Alexa Fluor 488 (C34775), HCS LipidTOX green neutral lipid stain (H34475), MitoSOX (M36008), MitoTracker FM Red (M22425), BODIPY 581/591 C11 (D3861), DiL-labelled LDL from human plasma (L3482), Transferrin-Alexa Fluor 488 (T13342), all from Invitrogen; human TGF-β 1 recombinant protein (100-21C) and human TGF-β2 recombinant protein (100-35B) from PeproTech; IncuCyte Cytotox green dye (4632), IncuCyte Cytotox red dye (4633) from Sartorius; liproxstatin-1 (S7699), quinoline-Val-Asp-difluorophenoxymethylketone (S7311) from Selleck; ferrostatin-1 (SML0583), 4-nitrophenyl β-D-xylopyranoside (2132), chondroitinase ABC (C2905), chondroitinase AC1 (C2780), dextran–FITC (46945), DGAT1 inhibitor A922500 (A1737), dimethyloxalylglycine (D3695), Fasnall benzenesulfonate salt/FASN inhibitor (SML1815), LDL human (LP2), alpha-tocopherol (T3634), albumin-FITC (A9771), heparinase I (H2519), heparinase III (H8891), all from Sigma-Aldrich; Hoechst 33342 (1399) from Thermo Fisher Scientific; TGF-β receptor inhibitor (SB431542, 1614) from Tocris Bioscence; DiL-labelled HDL from human plasma (770330), human LDL (770200) from Kalen Biomedical; necrostatin 1S (HY-14622A), 3-methyladenine (HY-19312) from MedChemExpress.

The acidic pH reporter pH-low insertion peptide variant 3 (pHLIP V3; NH2-ACDDQNPWRAYLDLLFPTDTLLLDLLW-COOH)[23] was prepared by solid-phase peptide synthesis and conjugated with tetramethylrhodamine (TAMRA) by Innovagen. The molecular weight of the peptide was confirmed by mass spectrometry analysis, and the purity was determined by analytical high-performance liquid chromatography (HPLC).

The following antibodies were used: α-tubulin (clone DM1A, ab7291, western blot (WB): 1:10,000), CD63 (clone MEM-259, ab8219, WB: 1:1,000), syndecan-1 (clone EPR6454, ab128936, IF/Flow Cyt: 1:500, WB: 1:3,000), EEA1 (ab2900, WB: 1:1,000), flotillin1 (ab41927, WB: 1:1,000), TSG101 (ab30871, WB: 1:1,000), β-actin (ab8227, WB: 1:10,000), CD9 (clone EPR2949, ab92726, WB: 1:1,000), GPX4 (clone EPNCIR144, ab125066, WB: 1:1,000); all from Abcam; mouse CD31 (clone MEC 13.3, 553371 IF 1:100) from BD Biosciences; CA9 (clone M75, AB1001, IF: 1:200) from Bioscience Slovakia; CD68 (clone D4B9C, 76437, IF 1:800), HIF-2α (clone D6T8V, 59973, WB: 1:1,000), SNAIL (clone C15D3, 3879, WB: 1:2,000), total-SMAD2 (clone D43B4, 5339, WB: 1:2,000), phospho-SMAD2 (Ser465/467) (clone 138D4, 3108, WB:1:2,000), TGF-β (3711, WB: 1:2,000), all from Cell Signaling; human CD31 (clone JC70A, M0823, IF: 1:50) from Dako; HIF-1α (GTX127309, WB: 1:1,000) from GeneTex; malondialdehyde (clone 6H6, MA5-27559, IF: 1:50), SLC7A11 (clone A7C6-R, MA5-44922, IF: 1:200), both from Invitrogen; chondroitinase ABC (ChABC) (clone 1E10, NBP1-96141, IF:100), apoE (clone WUE-4, NB110-60531, WB: 1:500), both from Novus Biologicals; CS (clone CS-56[64], C8035, IF/Flow Cyt: 1:200) from Sigma-Aldrich; single-chain fragment variable (scFv) HS (clone, AO4B08[65], IF/Flow Cyt: 1:50), CS (clone GD3G7[66], IF/Flow Cyt: 1:50), CS (clone IO3H10[67], IF/Flow Cyt:1:50) (kindly provided by Dr T. H. van Kuppevelt) and used together with mouse anti-VSV (clone P5D4, V5507, IF/Flow Cyt: 1:500) or rabbit anti-VSV (V4888, IF/Flow Cyt: 1:500), all from Sigma-Aldrich.

The following secondary antibodies were used: horseradish-peroxidase (HRP)-conjugated anti-rabbit (7074, WB: 1:10,000) from Cell Signaling or anti-mouse (a9044, WB: 1:10,000) from Sigma-Aldrich; goat anti-mouse Alexa Fluor 488 (A1100, 1:500), Alexa Fluor 546 (A11030, 1:500), Alexa Fluor 647 (A21235, 1:500) or goat anti-rabbit Alexa Fluor 488 (A11008, 1:500), Alexa Fluor 546 (A11010, 1:500), Alexa Fluor 647 (A21244, 1:500), streptavidin Alexa Fluor 488 (S32354, 1:500), streptavidin Alexa Fluor 546 (S11225, 1:500) or streptavidin Alexa Fluor 647 (S21374, 1:500), all from Invitrogen.

### Human brain tumour sample collection and processing

Clinical specimens were collected from patients referred to the Neurosurgery Department at Lund University Hospital, Sweden. The study was carried out according to the ICH/GCP guidelines and in agreement with the Helsinki declaration, and was approved by the local ethics committee, Lund University (Dnr. 454 2018/37). Inclusion criteria were age 18 years or above, WHO performance status 0–4, and ability to give written informed consent. No exclusion criteria related to sex and gender were present for the study. Participation was voluntary, and no financial or other incentives were provided. Patients were diagnosed by routine magnetic resonance imaging (MRI) of the brain, followed by standard surgical and pathological procedures, and received standard oncological treatment and appropriate follow-up according to national recommendations. Tumour specimens obtained from patients with glioma (WHO grade 2–4) or CNS metastasis were cryopreserved by snap-freezing in isopentane for further immunohistochemistry and immunofluorescence (IF) evaluation. Alternatively, fresh tumour tissue was minced with a dissecting scalpel, then dissociated with TrypLE Express (Gibco) and DNase I (Thermo Fisher Scientific) at 37 °C for 20 min on an orbital shaker. After filtration through 70- and 40-µm nylon cell filters, red blood cells were removed using red blood cell lysis buffer (BioLegend). PDCs were allowed to adhere before proceeding with further experiments and were fixed for IF analysis.

### Cell lines and patient-derived primary GBM cultures

Human GBM (U87MG, HBT-14) and pancreatic adenocarcinoma cell lines (PANC1, CRL-1469), both newly purchased from ATCC, were routinely cultured in high-glucose Dulbecco's modified Eagle medium (DMEM; Cytiva HyClone) supplemented with 10% fetal bovine serum (FBS; Sigma-Aldrich), 2 mM L-glutamine (L-Glut; Sigma-Aldrich), 100 U ml$^{-1}$ penicillin and 100 µg ml$^{-1}$ streptomycin (PEST; Sigma-Aldrich). Patient-derived primary GBM cell cultures from the Human Glioma Cell Culture Biobank (HGCC)[68], Uppsala U3054MG, U3047MG and U3017MG, were routinely cultured on surfaces precoated with 10 µg ml$^{-1}$ poly-L-ornithine (Sigma-Aldrich) and 10 µg ml$^{-1}$ laminin from Engelbreth–Holm–Swarm murine sarcoma basement membrane (Sigma-Aldrich), in primary cell medium composed of Neurobasal (Gibco) and DMEM/F12 medium (1:1, Gibco) supplemented with 10 ng ml$^{-1}$ epidermal growth factor (EGF) (Peprotech), 10 ng ml$^{-1}$ fibroblast growth factor 2 (FGF2) (Peprotech), stem cell supplements 1% N2 (Gibco) and 2% B27 (Gibco) and 1% penicillin/streptomycin (PEST). For 3D spheroid cultures, GBM cells were grown either in poly(2-hydroxyethyl methacrylate) (poly-HEMA; Merck)-coated dishes or in PrimeSurface 3D culture spheroid plates (S-Bio), then placed on an orbital shaker at 90 r.p.m. for 3–14 days.

**Acidosis-adapted (6.4/AA) and non-adapted (7.4/NA) culture cells.** To investigate the effects of acidosis, cells were cultured for the indicated timepoints in pH 6.4 medium supplemented with 20 mM HEPES (Merck), 20 mM 4-morpholineethanesulfonic acid sodium salt (MES; Sigma-Aldrich) and 20 mM 4-morpholinepropanesulfonic acid (MOPS; Sigma-Aldrich) to obtain stable acidic conditions. Medium pH was adjusted using 1 M HCl and/or 1 M NaOH, and sterile-filtered before use. AA cancer cells were established after 10 weeks treatment in pH 6.4. Control NA cells were grown under the same conditions but at physiological pH 7.4.

All cells were routinely cultured in a humidified atmosphere of 5% $CO_2$ at 37 °C. For hypoxia experiments, cells were incubated in a humidified Sci-tive NN hypoxia workstation (Ruskinn Technology) set at 5% $CO_2$, 94% $N_2$, 1% $O_2$ and 37 °C for the indicated timepoints. Cells were routinely tested for mycoplasma by Hoechst staining and high-resolution confocal microscopy.

## Laser microdissection

Human GBM tumour cryosections (10 µm) were mounted on nuclease DNase and RNase-free membranes (FrameSlidePET; Zeiss). The samples were rapidly stained for nuclei with cresyl violet (Sigma-Aldrich) and dehydrated in ice-cold ethanol. Adjacent sections were mounted on poly-lysine coated slides and stained for nuclei (Hoechst; Thermo Fisher Scientific), HCS LipidTOX (1:500) and the macrophage marker CD68. CD68 was used to identify and exclude LD-loaded macrophages, as described previously[5]. The tumour areas categorized as LD$^+$/CD68$^-$ and LD$^-$/CD68$^-$ from different membranes were isolated by laser microdissection (LCM) using the Zeiss PALM system employing a ×5 objective to identify the region of interest and a ×20 objective for precise cutting ($n = 5$ patients, with a total area of ~10 mm$^2$), pooled by group and then dissolved in 50 µl of lysis solution within specialized AdhesiveCaps. RNA extraction, quality control and mRNA expression analyses are described in the 'Sample preparation for gene expression analysis' section.

## Sample preparation for gene expression analysis

**For 3D versus 2D.** Primary GBM cells (U3054MG, U3047MG and U3017MG) were grown at pH 7.4 in routine culture medium as described above. Sub-confluent 2D cultures were lysed 72 h after seeding. For 3D spheroid cultures, cells were cultured in poly-HEMA coated dishes at $2 \times 10^5$ cells ml$^{-1}$ for 14 days before lysis, with medium exchanged every fourth day.

**For acidosis/hypoxia treatment.** U87MG cells were grown short term (48 h) in serum-free routine culture medium at pH 7.4 or 6.4, or at pH 7.4 in hypoxia, before lysis. U87MG and PANC1 NA and AA cells were grown in serum-free culture medium for 48 h, before lysis. For LCM-isolated GBM samples and primary GBM cell 3D/2D culture samples, RNA was isolated using an AllPrep DNA/RNA micro kit (Qiagen), and for all other samples an RNAeasy mini kit (Qiagen) was used. RNA concentration and purity were determined using a BioAnalyzer to ascertain acceptable RNA integrity number (RIN) values, and mRNA expression was analysed either on an Affymetrix Clariom D Pico gene array (LCM samples; primary cell 3D/2D culture samples; PANC1 NA/AA samples) or on an Illumina HumanHT-12 v4 Expression BeadChip system (U87MG short-term acidosis/hypoxia samples; U87MG NA/AA samples).

## GBM-CM and EV isolation

EVs were isolated from parental U87MG cells grown in serum-free medium, supplemented with 1% bovine serum albumin (BSA; Sigma-Aldrich), to exclude contamination with serum lipoproteins. Conditioned medium (CM) was collected after 48 h and centrifuged twice at 400$g$ and 4 °C to remove cell debris. In some cases, CM from U87MG NA and AA cells (U87MG NA/AA CM) was collected in the same way. EVs were pelleted by ultracentrifugation at 100,000$g$ at 4 °C for 2 h and washed with phosphate-buffered saline (PBS), followed by two additional ultracentrifugation steps at 100,000$g$ for 2 h. The final pellet was resuspended in PBS, and protein concentration was determined by a bicinchoninic acid assay (Pierce). EVs were characterized by immunoblotting for EV markers (see 'Western blot analysis' section) and by an Exoid system (Izon) for high-resolution measurements of particle size and concentration.

## Generation of U87MG ChABC-expressing cell line

A plasmid containing an optimized chondroitinase ABC (ChABC) sequence was generously provided by Dr E. M. Muir[69]. Restriction cloning was used to insert the ChABC sequence into the pLenti-CMV-IRES-puro lentiviral gene expression vector (Addgene). ChABC lentivirus for transduction was produced by PEI transfection with third-generation plasmids and U87MG cells were transduced overnight (multiplicity of infection (MOI) of 10). U87MG ChABC-expressing cells were selected and routinely cultured in puromycin (2 µg ml$^{-1}$, Sigma-Aldrich).

## siRNA transfection

For siRNA-mediated knockdown (KD), U87MG NA and AA cells were transfected with siRNAs targeting *CSGALNACT1* (siRNA#1: Hs_ChGn_8 FlexiTube, cat. no. SI04193273; siRNA#2: Hs_ChGn_1 FlexiTube, cat. no. SI00345793; both Qiagen) or a non-targeting control (siCtrl: negative control siRNA, cat. no. 1022076; Qiagen), at a final concentration of 10 nM, using Lipofectamine RNAiMAX (Thermo Fisher Scientific) according to the manufacturer's instructions in Opti-MEM I reduced serum medium (Gibco). Six hours after the initial transfection, the medium was replaced with fresh culture medium. After 48 h, the transfection procedure was repeated. At 96 h post-initial transfection, cells were collected for downstream experiments and analyses.

## Cell treatments

**Lipid particles.** Exogenous lipid particle treatments in 2D cultures were conducted at either pH 7.4 or pH 6.4 in SF routine culture medium, according to cell line, supplemented with or without EVs (50 or 100 µg ml$^{-1}$), LDL (15, 50 or 100 µg ml$^{-1}$) or 10% FBS. Unless otherwise specified in the figures or figure legends, low-dose LDL was applied at 15 µg ml$^{-1}$ in U3047MG and U3054MG cells, and at 50 µg ml$^{-1}$ in U87MG cells.

**3D treatments.** For 3D spheroid culture treatments, cells were first cultured for three days in PrimeSurface 3D culture spheroid plates (S-Bio) under standard culture conditions appropriate for each cell line, at pH 7.4. Treatments were then applied, with specific compounds and treatment durations detailed in the corresponding figures or figure legends. All treatments in 3D cultures were conducted in pH 7.4 medium.

**CSPG inhibition.** CSPG biosynthesis was inhibited by treatment with 4-nitrophenyl β-D-xylopyranoside[36] (CSi; 0.625, 1.25 or 2.5 mM). Cells were either pre-treated (48 h) before proceeding with further experiments or treated continuously. Treatment durations and culture medium conditions are detailed in the corresponding figures or figure legends. For PG sulfation inhibition experiments, cells were pre-treated (24 h) with sodium chlorate[70] (chlorate; 25 mM) or NaCl (Sigma-Aldrich), to control for osmotic effects of high chlorate concentration, before proceeding with further experiments. For CS enzymatic digestion experiments, cells were cultured in SF routine culture medium (pH 7.4) and treated without or with chondroitinase ABC lyase (60 mU ml$^{-1}$) and chondroitinase AC1 lyase (30 mU ml$^{-1}$) for 3 h at 37 °C. Enzyme addition was repeated, then incubation for another 3 h at 37 °C, followed by extensive washing before proceeding with further experiments.

**Targeting lipid metabolism.** Cells were treated with the FASNi SML1815 (50 µM) or the diacylglycerol *O*-acyltransferase-1 (DGAT1) inhibitor A922500 (DGAT1i; 12.5, 20, 25 or 50 µM). Treatment durations and culture medium conditions are detailed in the corresponding figures or figure legends.

**Treatments inducing and inhibiting ferroptosis.** Where indicated, DGAT1i treatment was combined with CSi (as described above) or applied following siRNA-mediated KD of *CSGALNACT1*. In some experiments, cells were pre-treated for 24 h and subsequently co-treated with alpha-tocopherol (α-Toco; 0.25 or 50 mM), ferrostatin-1 (Fer-1; 1 µM), liproxstatin-1 (Lip-1; 0.25, 0.5 or 1 µM), necrostatin 1S (Nec-1s; 1 or 5 µM), 3-methyladenine (3-MA; 10 or 20 µM) or quinoline-Val-Asp-difluorophenoxymethylketone (QVD; 20 µM). Treatment durations and culture medium conditions are detailed in the corresponding figures or figure legends.

**TGF-β and DMOG treatments.** Cells were treated with exogenous TGF-β1 or TGF-β2 (1 or 4 ng ml$^{-1}$) for 48 h, or with the hypoxia mimetic agent dimethyloxalylglycine (DMOG; 0.5 or 1 mM) for 72 h, at pH 7.4 in

SF culture medium supplemented with LDL (15 or 50 µg ml$^{-1}$). TGF-β1 and TGF-β2 treatments were preceded by 24 h of SF starvation. In some experiments, treatments were combined with the TGF-β receptor inhibitor SB431542 (TGFβRi; 5 or 15 µM). Additionally, in some experiments, TGF-β1 and DMOG were co-administered. All compounds used in cell treatments are listed in the section 'Compounds and antibodies'.

#### Lipid particle surface binding and uptake experiments

EVs were isolated as described above and, after the second centrifugation step, labelled with PKH67 green or PKH26 red fluorescence lipophilic dyes (Sigma-Aldrich), as previously described and recommended by the manufacturer[5,14,15]. For lipid particle uptake experiments, adherent cells were incubated with U87MG-derived PKH-labelled EVs, DiL-labelled LDL or DiL-labelled HDL (15 µg ml$^{-1}$ or as indicated) in SF routine culture medium (pH 7.4) for 1 h at 37 °C. The cells were extensively washed with PBS and 1 M NaCl, and either fixed in 4% paraformaldehyde (PFA; Sigma-Aldrich) and analysed by confocal microscopy or detached by trypsin (Gibco) and analysed by flow cytometry. For confocal co-localization experiments of EVs and endocytosis markers, cells were co-incubated with PKH-labelled EVs (50 µg ml$^{-1}$) and either cholera toxin subunit B-AF488 (CtxB; 25 µg ml$^{-1}$) or dextran–FITC (Dx; 2.5 mg ml$^{-1}$) before fixation and imaging. For confocal co-localization studies of SDC1 with PKH-labelled EVs or DiL-labelled LDL, the cells were pre-incubated with an anti-SDC1 antibody on ice for 30 min, followed by extensive washing with PBS. Lipid particle uptake was then performed as described above, after which cells were fixed, permeabilized, stained and imaged by confocal microscopy. For surface binding experiments, cells were detached using 0.5 mM ethylenediaminetetraacetic acid (EDTA; Sigma-Aldrich), washed, and incubated with U87MG-derived PKH67-labelled EVs or DiL-labelled LDL (15–50 µg ml$^{-1}$) in PBS containing 3% BSA for 1 h at 4 °C. The cells were then extensively washed with PBS and analysed by flow cytometry. All compounds, antibodies and dilution factors are listed in the section 'Compounds and antibodies'.

#### Tissue section and cell imaging

Human tumour and mouse brain cryosections (6 µm) were rehydrated in PBS for 5 min and fixed in 4% PFA. Plated 2D cells and 3D spheroid cultures were fixed in 4% PFA, and spheroids were subsequently incubated in 0.5 M sucrose at 4 °C overnight before being embedded in optimal cutting temperature (OCT) compound and sectioned (6 µm). For staining of cell-surface antigens, samples were blocked for 1 h at room temperature (r.t.) in PBS supplemented with 3% BSA (for plated cells) or 3% normal goat serum (for tissue and spheroid sections). For intracellular antigen staining, samples were permeabilized with 0.5% saponin for 15 min at r.t. Following blocking and/or permeabilization, samples were incubated overnight at 4 °C with primary antibodies diluted in the respective blocking solution. Samples were washed with PBS and fluorescently labelled with secondary antibodies for 1 h at r.t. All antibodies and dilution factors are listed in the section 'Compounds and antibodies'. LDs were stained with HCS LipidTOX (1:1,000) for 30 min at r.t. Terminal deoxynucleotidyl transferase dUTP nick end labelling (TUNEL) staining for dead cells was performed using the Click-iT Plus TUNEL Assay Kit and Alexa Fluor 647 (C10619, Thermo Fisher Scientific) according to the manufacturer's instructions. Nuclei were stained with Hoechst 33342 for 10 min at r.t., and sections were washed and mounted with fluorescent mounting medium (Invitrogen). For imaging of the acidic pH reporter pHLIP peptide in 2D plated cells, live cells were incubated for 30 min on ice with TAMRA-conjugated pHLIP V3 (2 µM) in SF culture medium set to pH 6.0 or 7.4. Cells were washed with PBS, fixed in 4% PFA, and the nuclei were stained with Hoechst 33342 before analyses. For 3D spheroid cultures, four- or nine-day-old spheroids were incubated for 24 h with TAMRA-conjugated pHLIP V3 (2 µM) in SF pH 7.4 medium. Afterwards, the spheroids were collected, fixed in PFA, incubated with sucrose, embedded in OCT, sectioned, and stained as described above. For mitochondrial imaging, live cells were

stained with MitoTracker Red FM (200 nM) or MitoSOX Red (2.5 µM) in SF culture medium for 30 min at 37 °C. After staining, the cells were washed and maintained in SF medium without phenol red (FluoroBrite DMEM, Gibco) and immediately imaged live.

Three imaging platforms were used and all samples from the same experiment were imaged with the same gain and exposure settings. The first is an LSM710 Airyscan confocal platform (Carl Zeiss AG), as follows: an inverted Axio Observer Z.1 LSM 710 confocal laser scanning microscope with an Airyscan detector and a photomultiplier tube (PMT) detector (Zeiss), equipped with a ×63/1.4 Plan-Apochromat oil-immersion, a ×40/1.3 EC Plan-Neofluar oil-immersion objective lens (Zeiss) and a diode laser (405 nm), a Lasos argon laser (488 nm), DPSS 561 nm and HeNe laser 633 nm (Zeiss); this system operates under ZEN 2.1 (black). The second platform is an LSM980 confocal platform (Zeiss) as follows: an inverted Axio Observer 7 LSM980 confocal laser scanning microscope (Zeiss), equipped with a 32-channel GaAsP spectral PMT detector, a ×63/1.40 C Plan-Apochromat oil-immersion lens, a ×40/1.20 C-Apochromat water-immersion objective lens (Zeiss) and diode lasers at 405 nm, 488 nm, 561 nm and 633 nm (Zeiss); this system operates under ZEN 3.8.2 (blue). The third platform is an Axio Scan.Z1 slide scanner (Zeiss) set-up as follows: an Axiocan 506 camera, a ×20/0.8 M27 Plan-Apochromat objective lens and a Colibri 5/7 LED light source (all Zeiss), with illumination performed with 385-nm, 475-nm, 555-nm and 630-nm LEDs; this system operates under ZEN 3.1 (blue).

Images were processed for analysis and visualization using ZEN 3.1 (blue), and the brightness and contrast settings were linearly adjusted and kept identical for images intended for comparison. All image analysis was performed using ImageJ software (v1.54p). For image-based quantifications of CS (Fig. 4d and Extended Data Fig. 8h), MitoTracker (Extended Data Fig. 10e,f), Mito-SOX (Fig. 8e and Extended Data Fig. 10f,g) or TUNEL (Fig. 8g), the signal fluorescence area was quantified on single-channel images after thresholding and, where indicated, normalized to the corresponding cell number within the same field. For CS quantification in LD$^+$ versus LD$^-$ regions of patient GBM sections (Fig. 2c), CD31 was used to identify and exclude areas of vessels, and the CS signal fluorescence area was quantified as described above. For image-based LD quantification (Figs. 2f and 4a,d and Extended Data Fig. 6a,e), LD positive area per cell was quantified by particle analysis after thresholding. To quantify the co-localization of internalized EVs with endocytosis markers (Extended Data Fig. 7f), regions of interest (ROIs) from single-cell outlines were saved in ImageJ software (v1.54p) and then converted into images using a custom-made MATLAB script. Endocytosis marker segmentation masks were created using maximum correlation thresholding in CellProfiler (v4.2.1) and were used to create masked objects from the EV channel. Finally, EV pixel intensities were quantified using MATLAB (v2018a) from the entire cell and from the masked EV images. EV-signal co-localizing with endocytosis marker was normalized against total internalized EV signal before plotting to obtain the proportion of co-localizing signal per cell.

#### Immunohistochemistry

Human tumour and mouse brain cryosections (6 µm) were fixed in 4% PFA in PBS, washed with tap water, and counterstained with haematoxylin and eosin (H&E; Histolab). Slides were then briefly dipped in graded alcohols (70% and 100%) and cleared twice in xylene for 5 min each. Finally, the slides were mounted and imaged using an Axio Scan.Z1 slide scanner (Zeiss).

#### Flow cytometry analysis

For staining of cell-surface antigens, cells were detached using 0.5 mM EDTA (Sigma-Aldrich), washed with PBS containing 3% BSA, and incubated with primary antibodies diluted in 3% BSA-PBS for 1 h at 4 °C. After incubation, the cells were washed, fixed in 2% PFA, and incubated with fluorescently labelled secondary antibodies for 1 h at r.t. Finally,

the cells were extensively washed in PBS before analysis. For antibody uptake experiments, primary and fluorescently labelled secondary antibodies were pre-complexed for 30 min at r.t., then incubated with adherent cells for 1 h at 37 °C. Following incubation, the cells were detached using trypsin and washed in PBS before analysis. All antibodies and dilution factors are listed in the section 'Compounds and antibodies'. Endocytic activity was assessed by incubating adherent cells with endocytic ligands in SF medium for 1 h at 37 °C. The ligands included cholera toxin subunit B-AF488 (CtxB; 5 µg ml$^{-1}$), dextran–FITC (Dx; 0.5 mg ml$^{-1}$) and transferrin-AF488 (Tfn; 10 µg ml$^{-1}$). Following incubation, the cells were washed with PBS, detached by trypsin, and washed again in PBS before analysis. Cell-surface proteins were biotinylated and internalized for 2 h as described in the 'Membrane protein biotinylation and endocytosis' section. The cells were then detached with trypsin, fixed, permeabilized (0.5% saponin, 30 min), blocked with 3% BSA, and stained with streptavidin-AF488 (5 µg ml$^{-1}$) before PBS washes and analysis. All samples were analysed on an Accuri C6 flow cytometer (BD Biosciences). For each sample, at least 10,000 events were recorded and analysed using BD CSampler Plus software v1.0.27.1 (BD Biosciences) and FlowJo (v10).

## IncuCyte live-cell analysis

Cell confluency (2D cultures), 3D spheroid culture growth, cytotoxicity (2D and 3D cultures), spheroid invasion capacity, lipid peroxidation potential, LD accumulation and acidic pH reporter TAMRA-conjugated pHLIP V3 accumulation were monitored using the IncuCyte S3 live-cell analysis system (Sartorius), housed in a humidified 5% $CO_2$ incubator at 37 °C. Cells and 3D spheroid cultures were treated as described in the 'Cell treatments' section. To assess cytotoxicity, treatments were performed in the presence of IncuCyte Cytotox green or red dye (2.5 µM for 2D cultures; 1.25 µM for 3D cultures). For 3D culture invasion assays, spheroids were formed over three days as described above, then embedded in 10% Matrigel (Corning) diluted in SF culture medium for 30 min at 37 °C. Following embedding, treatments were initiated, and the spheroid invasive area was monitored over time. Lipid peroxidation potential was evaluated by adding the fluorescent lipid probe C11-BODIPY$^{581/591}$ (2.5 µM) in SF culture medium 24 h after treatment initiation, and incubated for 12 h before image acquisition. LD accumulation was assessed by adding HCS LipidTOX (1:1,000 dilution) in SF culture medium three days after treatment initiation, followed by incubation for 12 h before image acquisition. The acidic pH reporter pHLIP V3 integration was evaluated in 2D cultures by image acquisition 30 min after the addition of TAMRA-conjugated pHLIP V3 (2 µM) in SF culture medium set to pH 6.0, 6.4 or 7.4. For 3D cultures, four- or nine-day-old spheroids were incubated for 24 h with 2 µM TAMRA-conjugated pHLIP V3 in SF pH 7.4 medium before image acquisition. Unless otherwise stated, phase contrast and fluorescent images were acquired at four distinct locations in each well (for 2D cultures) or in one location per well (for 3D cultures) every third hour for four days or longer, as indicated in the figures or figure legends. IncuCyte S3 integrated software (v2022B Rev2 or v2024B) was used for analysis and visualization of the IncuCyte images, and all settings were adjusted and kept identical across images intended for comparison. For statistical analyses, each well was considered an individual data point. For cytotoxicity analyses in 2D cultures, total area (µm$^2$ per image) of the Cytotox signal (above a set threshold) was normalized to confluency percent per well. Cytotoxicity is expressed as fold of Ctrl for each time point, or as fold of $t = 0$, as indicated in the figures or figure legends. For analyses of 3D cultures, spheroid size (brightfield object total area, µm$^2$ per image) was normalized to $t = 0$ for each spheroid. Alternatively, total area (µm$^2$ per image) of the Cytotox signal (above a set threshold) was normalized to the brightfield object total area per spheroid and expressed as the cytotoxicity percent of the total spheroid. For spheroid invasion capacity, the largest invading brightfield object area (µm$^2$) was quantified. Data were either presented as largest

invading brightfield object area (µm$^2$) over time or expressed as area under curve (AUC) values of invasive capacity over time. For LD accumulation in 3D cultures and pHLIP integration in 2D and 3D cultures, the respective signals are expressed as integrated intensity per cell (for 2D) or per spheroid (for 3D) and, when indicated in the figures or figure legends, normalized to Ctrl samples. Lipid peroxidation potential was calculated based on green integrated intensity (oxidized Bodipy) per well normalized to red integrated intensity (reduced Bodipy) per well and divided by the number of cells per well. The data are presented as fold of Ctrl, as indicated in the figures or figure legends.

## Cell metabolic assay

Cell metabolic activity was assessed using the MTT assay (Sigma-Aldrich) following 24 h of treatment, as described in the 'Cell treatments' section, according to the manufacturer's instructions.

## Quantitative real-time quantitative PCR

Total RNA was extracted using a GenElute Mammalian Total RNA Miniprep Kit (Sigma-Aldrich) according to the manufacturer's protocol, and complementary DNA was synthesized with a SuperScript III First-Strand Synthesis System kit (Thermo Fisher Scientific) with random hexamer primers running on a MasterCycler EpGradient 5341 thermal cycler. Real-time (RT) quantitative polymerase chain reaction (qPCR) was performed on a StepOnePlus real-time qPCR system (Applied Biosystems) using SYBR Green JumpStart Taq Readymix (Sigma-Aldrich). All reactions were run in triplicate with $n \geq 2$ biological replicates. Gene expression was normalized to the *GAPDH* housekeeping gene and the relative expression was calculated using the comparative Ct method ($2^{-\Delta\Delta Ct}$). The primers, previously designed in our laboratory, are as follows (Thermo Fisher Scientific): *BGN* (Biglycan): Fv: CTCAACTACCT-GCGCATCTCAG, Rv: GATGGCCTGGATTTTGTTGTG; *CHSY1* (chondroitin sulfate synthase 1): Fv: 5′-GCCCAGAAATACCTGCAGAC-3′, Rv: 5′-GCA CTACTGGAATTGGTACAGATG-3′; *CSGALNACT1* (chondroitin sulfate *N*-acetylgalactosaminyl transferase 1): Fv: 5′-TCAGGGAGAT GTGCATTGAG-3′, Rv: 5′-AGTTGGCAGCTTTGGAAGTG-3′; *DCN* (Decorin): Fv: 5′-AATGCCATCTTCGAGTGGTC-3′, Rv: 5′-TGCAGGTCTAGCAG AGTTGTGT-3′; *DSE* (dermatan sulfate epimerase): Fv: 5′-GTCCAGA GGCACTTCAACATC-3′, Rv: 5′-AGTCCGCAATAGCCACAGTC-3′; *GAPDH* (glyceraldehyde 3-phosphate dehydrogenase): Fv: 5′-GAAGG TGAAGGTCGGAGTCAAC-3′, Rv: 5′-CAGAGTTAAAAGCAGCCCTGGT-3′.

## Western blot analysis

Cells, EVs or DiL-HDL particles were lysed in radioimmunoprecipitation assay (RIPA) buffer supplemented with cOmplete Mini EDTA-free protease inhibitor cocktail (Roche) and PhosSTOP phosphatase inhibitor (Roche). For PG core protein analyses, cells were lysed in 2% Triton X-100 buffer (Sigma-Aldrich), and total PGs were purified using diethylaminoethyl cellulose (DEAE)-cellulose chromatography, desalted with PD-10 columns, and subsequently freeze-dried, as previously described[71]. GAG chains were digested (or left untreated) with heparinase III (0.6 mIU ml$^{-1}$) and chondroitinase ABC (40 mU ml$^{-1}$) lyases at 37 °C overnight. Proteins were separated on a 4–12% NuPAGE Bis-Tris gel (Thermo Fisher Scientific) and transferred onto polyvinylidene difluoride (PVDF) membranes (Thermo Fisher Scientific). The membranes were blocked for 1 h at r.t. in either 5% skimmed milk or 3% BSA diluted in Tris-buffered saline with 0.1% Tween 20 detergent (TTBS), then incubated overnight at 4 °C with the indicated primary antibodies. After washing, the membranes were incubated with HRP-conjugated secondary antibodies for 1 h at r.t. All antibodies and dilution factors are listed in the section 'Compounds and antibodies'. Target proteins were detected using ECL western blotting substrate (Thermo Fisher Scientific) according to the manufacturer's instructions. Blot images were processed for analysis and visualization using ImageJ software (v1.54p) or Image Studio Lite (v5.3.5), and brightness and contrast were linearly adjusted. All unprocessed images of blots are available in the source data.

## GAG composition analyses

CS and HS disaccharide composition analyses were performed as previously described[72,73]. Briefly, U87MG NA and AA cells were grown to subconfluency, collected by scraping, and freeze-dried. Conditioned medium (48 h) from U87 NA and AA cells was collected in parallel and centrifuged twice at 400$g$ to remove debris. Freeze-dried cell pellets and CM were digested with chondroitinases and heparinases at 37 °C overnight, and the resulting disaccharides were analysed by HPLC.

## Membrane protein biotinylation and endocytosis

Cell-surface biotinylation and proteomic analyses were performed as previously described[74,75]. Briefly, U87MG NA and AA cells were incubated on ice with 1 mg ml$^{-1}$ sulfo-NHS-SS-biotin (Thermo Fisher Scientific). Unbound biotin was quenched with 0.1 M glycine in PBS. For endocytosis assays, cells were incubated in pre-warmed SF medium at 37 °C for 2 h, then placed on ice to stop internalization. Surface biotin was removed by treatment with 300 mM sodium 2-mercaptoethanesulfonate (MesNa; Thermo Scientific), followed by quenching with 5 mg ml$^{-1}$ iodoacetamide (Sigma-Aldrich). For liquid chromatography tandem mass spectrometry (LC-MS/MS) analyses, biotinylated proteins were purified using HiTrap streptavidin HP 1-ml columns (GE Healthcare) and eluted with 150 mM MesNa in PBS containing 0.1% Triton X-100. Proteins were precipitated in 10% trichloroacetic acid, resuspended in 6 M urea, digested with trypsin, desalted, and analysed using a Thermo Easy-nLC 1000 system coupled to a Q-Exactive HF-X mass spectrometer (Thermo Fisher Scientific). Raw data-dependent acquisition (DDA) data were analysed with Proteome Discoverer 2.3 (PD 2.3) software (Thermo Fisher Scientific), in which the peptides were identified with SEQUEST HT paired with the UniProtKB human database (release 2020_05).

## CUT & RUN

Genome-wide binding sites of HIF-1α were determined in U87MG AA and NA cells, alongside DMOG-treated and the corresponding Ctrl parental cells, using the CUT & RUN assay kit (active motif, #53180, version 47) following the manufacturer's instructions. Sample preparation was performed as previously described[76]. Briefly, $5 \times 10^5$ cells per line and per CUT & RUN reaction were collected and mildly fixed in 0.1% formaldehyde (Thermo Fisher Scientific) for 2 min at r.t. on a shaker. Crosslinking was quenched by adding glycine (125 mM final concentration) for 5 min, and the samples were then washed in cold 1× PBS, flash-frozen, and stored until used. For normalization purposes, 5,000 *Drosophila melanogaster* nuclei (Active Motif, #53183) were then added as spike-in before sample nuclei isolation. The isolated nuclei were first incubated with the concavalin beads, followed by overnight incubation with 1 µg of HIF-1α antibody (GeneTex, GTX127309) per CUT & RUN reaction at 4 °C. Thereafter, chromatin-bound beads were mixed with pAG-MNase in cell permeabilization buffer, and the enzyme was activated by adding 1 µl of 0.1 M cold calcium chloride, followed by incubation at 4 °C for 2 h while rotating at 25 r.p.m. Decrosslinking was performed by incubation with Stop Solution containing RNase and glycogen at 37 °C for 10 min. Enriched DNA was purified using the provided DNA purification columns SF and further processed for library preparation using the NEBNext Ultra II DNA library Prep Kit for Illumina (New England Biolabs) and Multiplex Oligos (New England Biolabs), following Active Motif's CUT & RUN library preparation protocol. Library fragment size distribution was assessed using a TapeStation High Sensitivity DNA Analysis assay, and the libraries were sequenced as PE150 on a NovaSeqX Sequencing System (Illumina).

CUT & RUN data were processed following previously described pipelines[77]. Raw sequencing files (FASTQ) were quality-checked using FastQC (https://www.bioinformatics.babraham.ac.uk/projects/fastqc/). Adapter trimming was performed with Trimmomatic (v0.39), and reads were aligned to both the human genome (GRCh38.p14/hg38)

and the *D. melanogaster* genome (FlyBase r6.62) using Bowtie2 (v2.4.5) with the following parameters:--local--very-sensitive-local--no-unal--no-mixed--no-discordant--phred33 -I10 -X 700. Duplicate reads were identified and removed using collate, fixmate and markdup functions in samtools. Genome-wide signal coverage was normalized to reads per genomic content (RPGC) per bin (bin size: 50 bp) and scaled using a spike-in-derived factor based on the ratio of *D. melanogaster* reads per sample to total *D. melanogaster* reads aligned in IgG controls with deepTools (3.5.5). The fraction of reads in peaks (FRiP) was calculated using the featureCounts subtool from SubRead (v2.1.1). Peak calling was performed using SEACR (1.3) with a stringent cutoff of false discovery rate (FDR) < 0.01. Pairwise comparisons of HIF-1α peaks were conducted for AA versus NA and DMOG-treated versus Ctrl cells using ChIPpeakAnno and ChIPseeker (Bioconductor/3.20) in R. Called peaks annotated as sample-specific or common were assigned to the closest genes using EnsDb.Hsapiens.v86 and TxDb, Hsapiens, UCSC, hg38, knownGene. To quantify HIF-1α binding near key genes, genomic bins within 5, 10 and 100 kb of selected gene promoters were analysed. Gene sets related to CS biosynthesis, PGs and GAG metabolism were retrieved from EnsDb.Hsapiens.v86 and compared to genes associated with sample-specific and common HIF-1α binding sites. Genome coverage files and peak sets for NA versus AA and DMOG versus Ctrl were uploaded to Galaxy (usegalaxy.org, 25.0.rc1)[78] and visualized using the UCSC Genome Browser (hg38)[79]. Additional quality control metrics and information for CUT & RUN analyses are provided in Supplementary Table 1.

## GBM xenograft mouse models

Experiments involving mouse orthotopic xenografts were approved by the Ethical Committee for Animal Research in Lund-Malmö (permit nos. 5.8.18-14006/2019 and 5.8.18-01073/2024) and were carried out according to national care regulations of the Swedish Board of Animal and European Union Animal Rights and Ethics Directives. Mice were group-housed in a specific pathogen-free facility with standard food and water, a 12-h light/dark cycle, 20–26 °C temperature and 30–70% humidity. For all in vivo experiments, female NOD SCID gamma (NSG) mice, aged 5–7 weeks (obtained from the Jackson Laboratory (JAX)), were used. GBM models included (1) a patient-derived xenograft model of U3054MG cells or (2) a cell line-derived human xenograft model of U87MG 7.4/NA or 6.4/AA cells. In all cases, $1 \times 10^5$ glioma cells in 4 µl of SF culture medium with 10% Matrigel (Corning) were injected into the brains of mice anaesthetized with isoflurane, then they were placed on a stereotactic frame. A hole was drilled into the skull and cells were inoculated in the right hemisphere, 1 mm anterior and 1.5 mm lateral from the bregma, and 2.5 mm ventral from the dura. In some cases, mice were monitored with T2-weighted MRI scans on a 9.4-T MRI machine (Bruker). For treatment studies, pumps for continuous intratumoral delivery (7- or 14-day mini-osmotic pumps, Alzet model 1007D or 1002) were filled with control sham vehicle (artificial cerebrospinal fluid, aCSF, Biotechne) or active treatments: 4-nitrophenyl β-D-xylopyranoside (CSi; 1.25 or 2.5 mM), DGAT1 inhibitor A922500 (80 µM) or a combination of the two, and implanted subcutaneously into the anaesthetized mice. A catheter delivered the treatment intratumorally into the cerebrum through the original drill hole. The skin incision was closed using metal clips. When treatment duration was ended (after 7 or 14 days), the pumps were removed under general anaesthesia. Tumour burden in orthotopic xenograft models was assessed based on neurological symptoms. The mice were monitored daily and euthanized immediately upon the onset of neurological distress, in accordance with ethical approval. When tumour size was assessed by MRI, only asymptomatic mice were included in the analysis, and ethical permission limits were not exceeded. The primary endpoint was overall survival (OS), with 6–10 mice per group. Mouse brains were dissected and cryopreserved by snap-freezing in isopentane for further immunohistochemistry and IF evaluation.

### Gene array processing

Gene array data were processed using the R statistical language (v4.4.2) within RStudio. In the case of the Affymetrix array experiments, data preprocessing steps were executed using the oligo (v1.70.0)[80] package. First, raw CEL files were loaded into R (oligo::read.celfiles), then transcript abundances were normalized using the Robust Multichip Average (RMA) preprocessing methodology, including background correction and quantile normalization (oligo::rma). Annotation of probe IDs was performed with the affycoretools (v1.78.0) package (affycoretools::annotateEset) with the clariomdhumantranscript-cluster.db (v8.8.0) ChipDb package. Illumina BeadChip data were processed using the limma package (v3.62.1)[81]. Probe profile files were imported with limma::read.ilmn, normalized using limma's background correction method for Illumina BeadChips (limma::neqc) and annotated against the HumanHt12v4 annotation data using the illuminaHumanv4.db (v1.26.0) package. To reduce unannotated probes and update deprecated identifiers, an additional round of annotation was performed using org.Hs.eg.db (v3.20.0). Probes without annotation and, in the case of Illumina data, those lacking confident detection ($P < 0.05$ in at least three arrays), were excluded from downstream analysis. Differential expression analysis was performed as follows. The design matrix was built with no baseline group, using stats::model.matrix (v4.4.2), treating all groups independently. A linear model was fitted to each gene using the design matrix along with the normalized gene expression matrix (limma::lmFit). This was followed by the construction of a contrast matrix (limma::makeContrasts) and the computation of estimated coefficients and standard errors from the fitted linear model (limma:: contrasts.fit). Empirical Bayes statistics moderation was applied (limma::eBayes) to compute moderated $t$- and $F$-statistics and the log-odds of differential expression. Multiple testing correction was performed using the Benjamini–Hochberg method. Significantly differentially expressed genes were extracted using limma::topTable with number set to infinity, to return the full annotated dataset. Visualization of gene expression data was generated with the package ggplot2 (3.5.1)[82].

### Pathway analysis and signature generation

Gene set enrichment analysis (GSEA) was performed with $\log_2$-transformed gene expression change values as input using the clusterProfiler (v4.14.4)[83] R package (clusterProfiler::GSEA). Enriched sets were investigated amongst Hallmarks (H), Gene Ontology Biological Processes (C5, GO:BP), KEGG (C2, CP:KEGG) and REACTOME (C2, CP:REACTOME) pathway annotated gene sets from the Molecular Signatures Database (MSigDB) in R with msigdbr::msigdbr (v7.5.1)[84]. GSEA results were further analysed by clustering and network analysis as follows. Cohen's kappa was calculated between every gene set, and an adjacency matrix was set up with the threshold 0.25, then an undirected network was created from the enriched neighbouring terms, and Louvain community detection was employed to find clusters. For each node within the resulting network, a hub score was computed with igraph::hub_score (2.2.1)[85] to estimate its influence within the topology. For the generation of an LD$^+$/CS$^+$ transcriptional signature, 21 genes were selected based on their consistent upregulation ($\geq 0.5$ $\log_2$(fold change, FC)) in LCM LD$^+$ versus LD$^-$ samples and being significantly upregulated ($\geq 0.5$ $\log_2$FC, (adjusted $P$ value) adj$P_v$ < 0.05) in at least two out of three 3D versus 2D primary cell cultures (see 'Laser microdissection' and 'Sample preparation for gene expression analysis' sections). Scoring of the LD$^+$/CS$^+$ gene signature in the Ivy Glioblastoma Atlas Project (IvyGap) (RRID: SCR_005044)[21] was performed using the hack_sig function from the hacksig (v0.1.2) R package, with 'zscore' as sample-wise signature scoring method. Results were plotted with ggplot2 package combined with the ggpubr package (v0.6.0; RRID:SCR_021139) for Wilcoxon-based unpaired mean comparison between plotted groups (stat_compare_means function) and $P$ value generation. Results were plotted with the ggplot2 package combined

with the ggExtra package (v0.10.0) for visualization of the signature score distribution as boxplots (ggMarginal function).

### Software

The software used for individual analyses is described in the previous sections. R (v4.4.2) with RStudio and GraphPad Prism (v10.5.0) were used to create figures and perform statistical testing. Schematics were created with BioRender.com and figure composition was performed with Adobe Illustrator v.28.6.

### Statistics and reproducibility

Statistical analyses were performed in R with RStudio, or in GraphPad Prism. GSEA statistics for enrichment score (ES), normalized enrichment score (NES), nominal $P$ value and FDR were performed in R using the clusterProfiler (v4.14.4)[83] R package. The significance of pathway overrepresentation terms was calculated according to Fisher's exact test. Significance tests of differentially expressed genes were performed on $\log_2$-transformed expression values (for which normality assumptions are applicable due to the lognormal distribution) using moderated $t$ statistics as per the limma package. Comparisons of tumour region (IvyGap) means were performed with the one-sample Wilcoxon signed-rank test. Statistical analyses of quantitative experimental models were performed using either an unpaired two-tailed Student's $t$-test for between two group comparisons, one-way ANOVA tests with Tukey´s post hoc test for multiple group comparisons and two-way ANOVA with Tukey's post hoc test (multiple groups) or Šidák's post hoc test (between two groups) for repeated measures. For survival curves, $P$ values were obtained by using the log-rank (Mantel–Cox) test. In vitro experiments were carried out with at least three independent biological replicates in a minimum of two independent experiments, unless otherwise indicated in the figure legends. Both technical and biological replicates were reproducible. Data are represented as mean ± s.e.m., with the level of significance defined as $P < 0.05$, unless otherwise specified in figure legends.

**Sample size determination.** No statistical methods were used to predetermine sample sizes, but our sample sizes are similar to those reported in previous publications[5,14–16].

**Data exclusion.** No data were excluded from the analyses.

**Randomization.** For in vitro studies, experiments were not randomized; however, all cell lines/organoids were treated identically without prior designation. For in vivo mouse experiments involving drug treatment, same-aged female mice were randomly assigned into experimental groups.

**Blinding.** Data collection and analysis were not performed blind to the conditions of the experiments.

**Assumptions for statistical test.** Data distribution was assumed to be normal, but this was not formally tested.

### Reporting Summary

Further information on research design is available in the Nature Portfolio Reporting Summary linked to this Article.

## Data availability

All data supporting the graphs in this paper, as well as all unprocessed blot images, are available in the source data files. Additional quality control metrics and information for CUT & RUN analyses are provided in Supplementary Table 1. The mRNA array datasets generated have been deposited in the NCBI Gene Expression Omnibus (GEO) under accession codes GES300758, GSE300765, GSE300768 and GSE300771. The CUT & RUN datasets are available in GEO under accession code

GSE300142. Imaging files and all other raw data files are available from the corresponding author (due to the size of this material). Source data are provided with this paper.

## Code availability

All R code and processed data supporting the findings for Figs. 1b–e, 2e and 3a,f and Extended Data Figs. 3b,h, 4a,e and 6a,f are available from Zenodo at (https://doi.org/10.5281/zenodo.18414879)[86], which provides the full reproducible analysis pipeline.

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

## Acknowledgements

We thank all the patients that contributed to this study. We are grateful to our colleagues at the Neurosurgery and Oncology Departments for their continuous input. We thank B. Baldetorp for support with infrastructure, the BEA core facility, Karolinska Institute, for help with gene expression arrays, F. de Winter and E. Muir for generously providing the ChABC lyase plasmid, and J. Johansson and A. Wittrup for help with confocal microscopy co-localization analysis. The computations were enabled by resources in projects hpc2n2025-057 and hopc2nstor2025-029 provided by the National Academic Infrastructure for Supercomputing in Sweden (NAISS) through HPC2N (High Performance Computing Center North) at Umeå University (Sweden). This work was supported by Swedish Cancer Society grants CAN 23 2655 Pj, 23 2937 Pj and 24 3666 Pj (to M.B., J.B. and S.R.), Swedish Research Council grants VR-MH 2023-02106 and 2024-02736 (to M.B. and S.R.), Swedish Childhood Cancer Foundation grants PR2023-0078 and PR2022-0117 (to M.B. and J.B.), Cancer Research Coordinating Committee and NHLBI grants C23CR5578 and HL131474 (to J.D.E.), the Fru Berta Kamprad Foundation (to M.B. and J.B.), the Sjöberg Foundation (to M.B. and J.B.), Skåne University Hospital donation funds (to M.B.), governmental funding of clinical research within the national health services, ALF (to M.B.), the European Union's Horizon 2020 COFUND Programme CanFaster 754299 (to M.B.), the Knut and Alice Wallenberg Foundation WCMM-Umeå (to S.R.) and a generous donation by Viveca Jeppsson (to M.B. and J.B.).

## Author contributions

A.B.-R., with support from M.C.-M., A.B. and S.B., acquired the flow cytometry, IncuCyte, RT–qPCR, western blot and confocal microscopy data, V.G., with support from M.H., generated laser capture microdissection transcriptomics data. A.B.-R., with support from H.T., E.G., S.J., J.E.P. and M.C.J., performed animal treatment studies. M.H. generated transcriptome datasets with support from K.G.d.O. and H.T., and C.C., I.N. and S.R. generated CUT & RUN datasets. Patient tumour samples and cell cultures were provided by J.B. and K.F.-N., and L.K., E.T. and A.M. performed GAG disaccharide analyses. T.H.v.K. generated and provided anti-CS and anti-HS antibodies. A.B.-R., M.C.-M., J.D.E., V.G. and M.B. wrote the manuscript. M.B. designed the study. All authors edited and approved the manuscript.

## Funding

## Inclusion and ethics

The study involved human tissue specimens and patient-derived cell cultures obtained with informed consent and under protocols approved by institutional review boards. All animal experiments were performed in compliance with institutional and national ethical guidelines, with

approval from the appropriate animal care and use committees (Methods). This work reflects a collaborative effort across multiple institutions and disciplines, and all contributors are appropriately acknowledged. The authors are committed to principles of inclusion, equity and transparency, and will make all relevant data and materials available in accordance with the policies of *Nature Cell Biology*.

## Competing interests

The authors declare no competing interests.

## Additional information

**Extended data** is available for this Paper at https://doi.org/10.1038/s41556-026-01879-y.

**Correspondence and requests for materials** should be addressed to Mattias Belting.

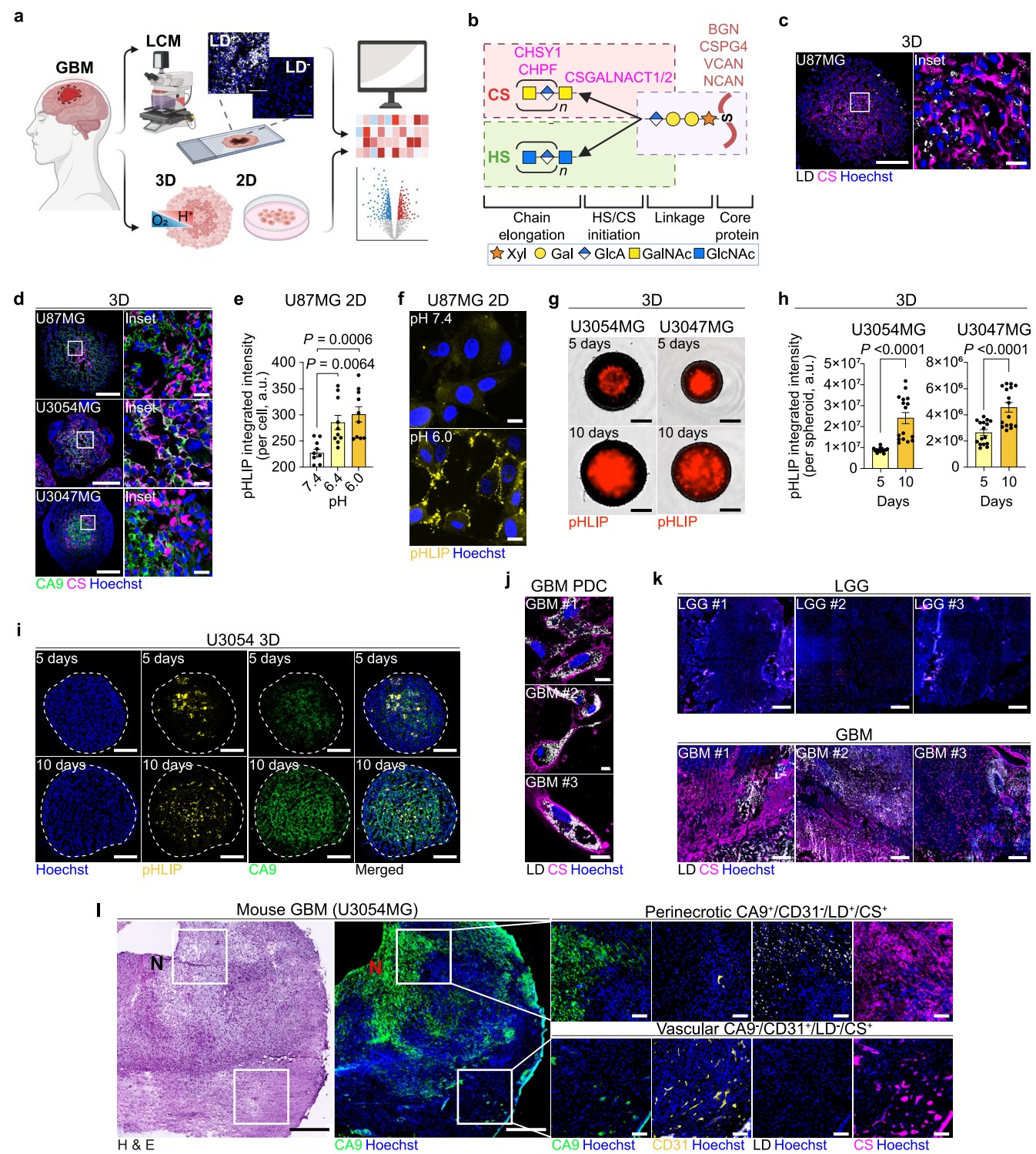

**Extended Data Fig. 1 | See next page for caption.**

**Extended Data Fig. 1 | CS-enriched glycocalyx defines the lipid-rich, stressed tumour niche. a**, Schematic overview of comparative gene expression analyses performed on LD$^+$ versus LD$^-$ GBM tumour areas captured by LCM ($n = 5$ patients), and in primary GBM 3D (LD$^+$) versus 2D (LD$^-$) cultures (established from $n = 3$ patients). **b**, Schematic illustration of key genes involved in CSPG biosynthesis. **c** and **d**, Fluorescence imaging of LDs and CS (**c**), and CA9 and CS (**d**) in the indicated GBM 3D cultures (representative of $n > 10$ spheroids/culture). Scale bars: 200 and 20 μm (zoomed). **e** and **f**, Accumulation of the acidic pH reporter TAMRA-conjugated pHLIP in U87MG cells at pH 6.0 and 6.4 (and pH 7.4 as control) quantified in (**e**) by IncuCyte (mean pHLIP integrated intensity per cell ± s.e.m., $n = 10$, 2 independent experiments) and visualized by confocal imaging (**f**) at pH 6.0 or 7.4 (representative from 2 independent experiments). Scale bars: 10 μm. **g** and **h**, IncuCyte images (**g**) of the acidic compartment in patient-derived U3054MG and U3047MG 3D cultures by TAMRA-conjugated pHLIP (at 5 and 10 days) (representative of $n = 16$ spheroids/condition), and corresponding quantification (**h**) (mean pHLIP integrated intensity/spheroid ± s.e.m., $n = 16$ spheroids/condition, 2 independent experiments). Scale bars: 300 μm. **i**, Confocal imaging shows central accumulation of TAMRA-conjugated pHLIP, overlapping with the acidic marker CA9 in sections from U3054MG 3D cultures (representative of $n > 10$ spheroids). Scale bars: 200 μm. **j**, Fluorescence imaging of LDs and CS in freshly resected GBM PDCs ($n = 3$ individual tumours). Scale bars: 10 μm. **k**, Fluorescence imaging of LDs and CS expression in tumour sections from LGG (top), and GBM (bottom) (representative of $n \geq 3$ patients/group). Scale bars: 500 μm. **l**, H&E and matching fluorescence images of tumour sections from mice xenografted with the patient-derived GBM culture U3054MG, highlighting perinecrotic region (upper row; CA9$^+$/CD31$^-$/LD$^+$/CS$^+$) and vascular region (lower row; CA9$^-$/CD31$^+$/LD$^-$/CS$^+$) (representative of $n = 3$ individual tumours). Scale bars: 500 and 100 μm (zoomed). N, necrosis. CS was visualized via CS-56 antibody (**c**) or scFv clone GD3G7 (**d**, **j**, **k** and **l**). Data in (**e**, **g** and **h**) was acquired by IncuCyte live-cell imaging. Significance was determined one-way ANOVA (**e**) or two-sided t-test (**h**). Illustration (**a** and **b**) was created with Biorender.com.

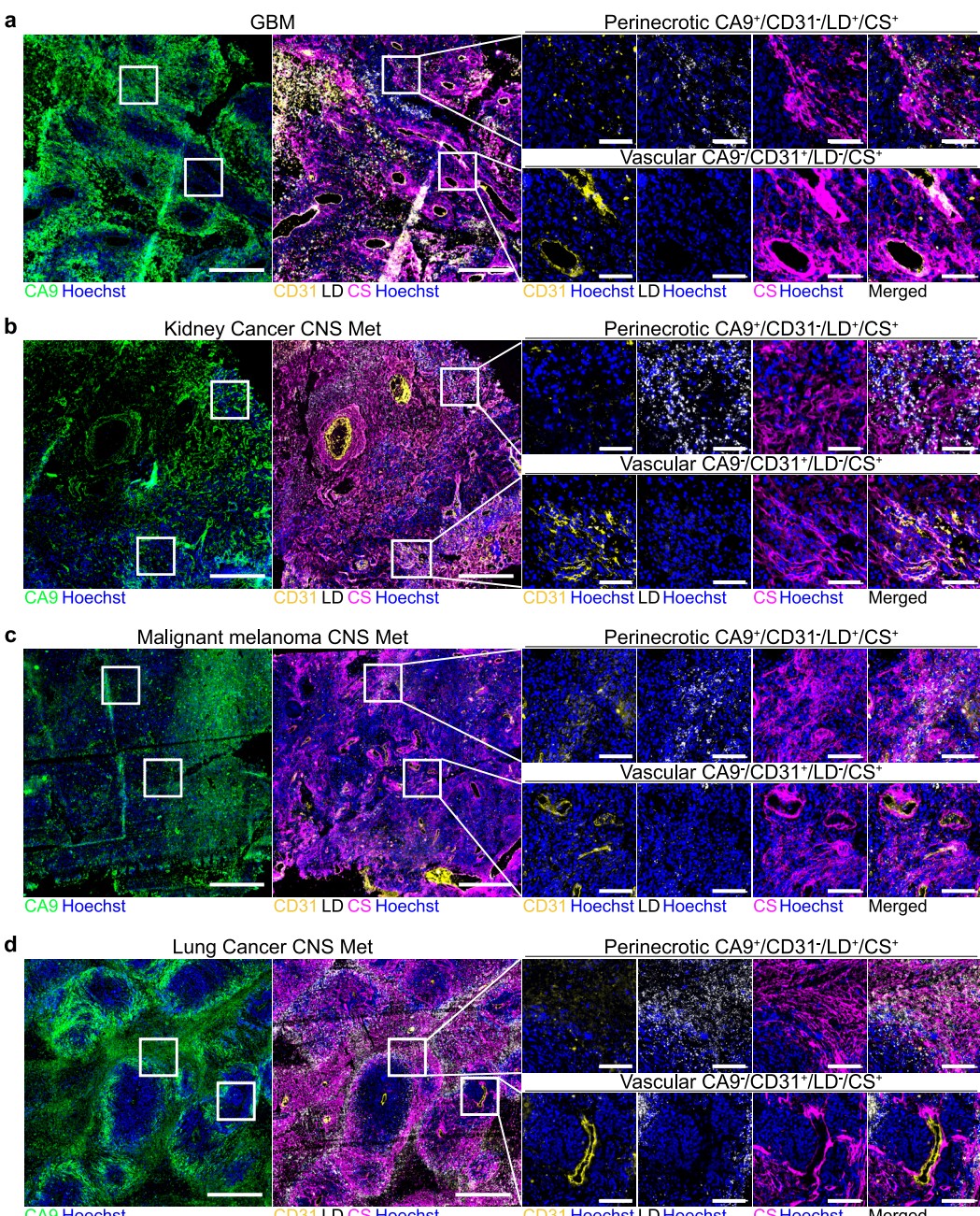

**Extended Data Fig. 2 | CS-enriched glycocalyx defines the lipid-rich, stressed tumour niche in GBM and CNS metastases. a–d**, Fluorescence imaging of tumour sections highlighting perinecrotic region (upper row; CA9+/CD31−/LD+/CS+) and vascular region (lower row; CA9−/CD31+/LD−/CS+) from GBM (**a**) (representative of >5 patients), and CNS metastases originating from kidney cancer (**b**), malignant melanoma (**c**), and lung cancer (**d**) (representative of 1-2 patients/tumour entity). Scale bars: 500 and 100 μm (zoomed). CS was visualized via scFv clone GD3G7 (**a-d**).

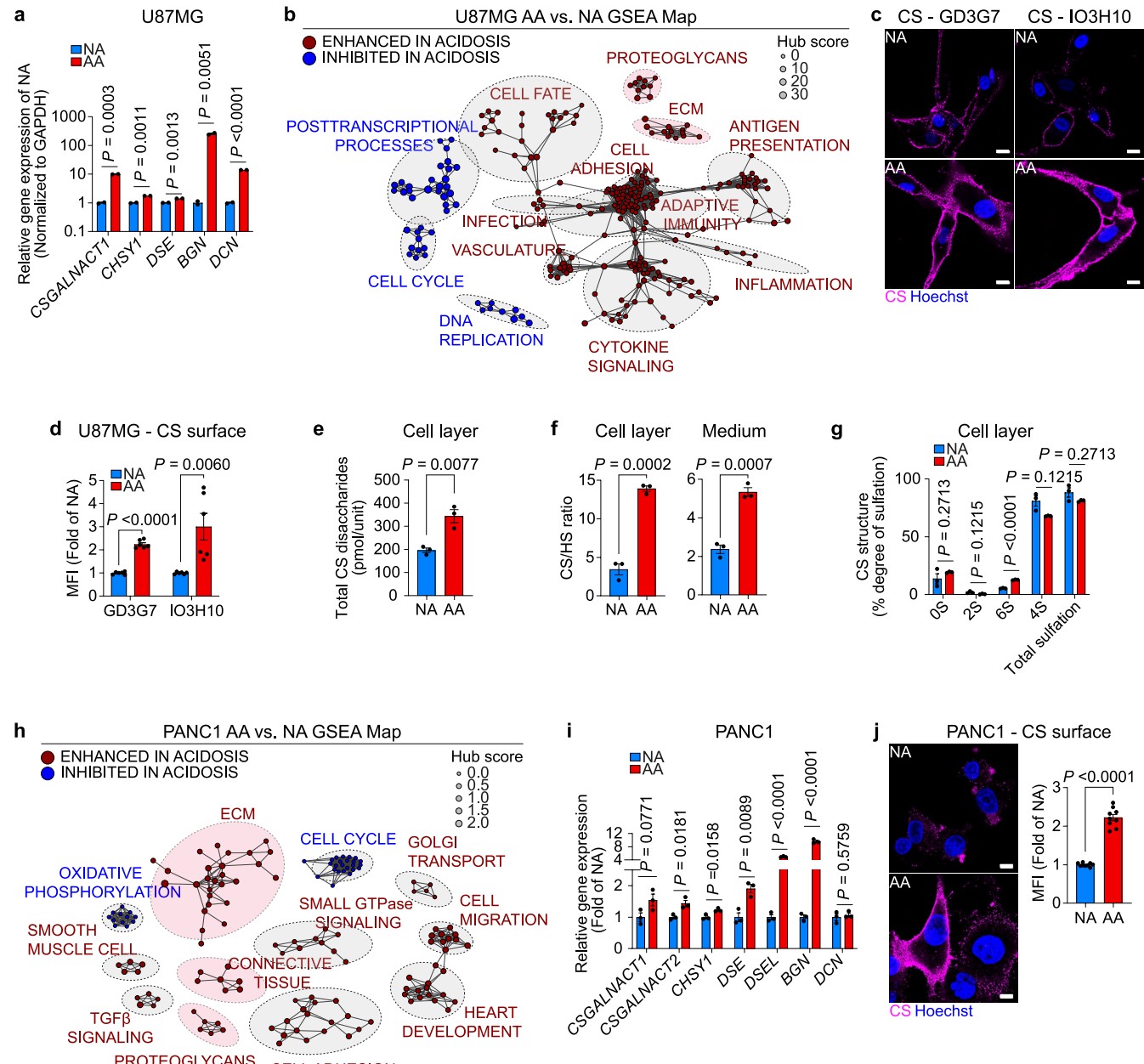

**Extended Data Fig. 3 | CS-glycocalyx encapsulation is an adaptive response to tumour acidosis. a**, Real-time qPCR quantification of key genes involved in CS-glycocalyx formation in U87MG AA versus NA cells (mean fold of NA ± s.e.m., *n* = 2 biological replicates, each with 3 technical replicates). **b**, GSEA enrichment mapping of significantly enhanced gene sets (≥ 5 gene sets/cluster with adj*P*ᵥ < 0.001) in U87MG AA versus NA cells (*n* = 3 biological replicates). **c**, Confocal imaging of CS surface signal in U87MG AA and NA cells detected with GD3G7 and IO3H10 antibodies (representative of 2 independent experiments). Scale bars: 10 μm. **d**, Flow cytometry quantification of CS surface signal as in (**c**) (mean fold of NA ± s.e.m., *n* = 6, 2 independent experiments). **e-g**, Disaccharide composition analysis of U87MG AA and NA cells, indicating total CS (**e**) (mean pmol CS normalized to cell input ± s.e.m.), CS/HS ratio in cell layer (**f**, left) (mean pmol CS disaccharides/pmol HS disaccharides ± s.e.m.) or medium (**f**, right) (mean ng CS disaccharides/ng HS disaccharides ± s.e.m.),

and CS sulphation pattern (**g**) (mean % degree of sulphation ± s.e.m.); (**e-g**) *n* = 3 biological replicates. **h**, GSEA enrichment mapping of significantly enhanced gene sets (≥ 5 gene sets/cluster with adj*P*ᵥ < 0.0001) in PANC1 AA versus NA cells (*n* = 3 biological replicates). **i**, Relative expression by mRNA array of key genes involved in CS-glycocalyx formation in PANC1 AA versus NA cells (mean fold of NA ± s.e.m., *n* = 3 biological replicates). **j**, Confocal imaging of CS surface signal in PANC1 AA and NA cells (left; representative of 2 independent experiments), and corresponding quantification by flow cytometry (right; mean fold of NA ± s.e.m., *n* = 9, 3 independent experiments). Scale bars: 10 μm. GSEA employed Hallmark, Reactome, KEGG, and GO databases (**b** and **h**), node size represents influence within the topology. CS surface signal was visualized via scFv clones GD3G7 and IO3H10 (**c**) or CS-56 antibody (**j**) and quantified via GD3G7-AF488 and IO3H10-AF488 (**d**) or CS-56-AF488 (**j**). Significance was determined by two-sided t-test (**a**, **d-g**, **i** and **j**).

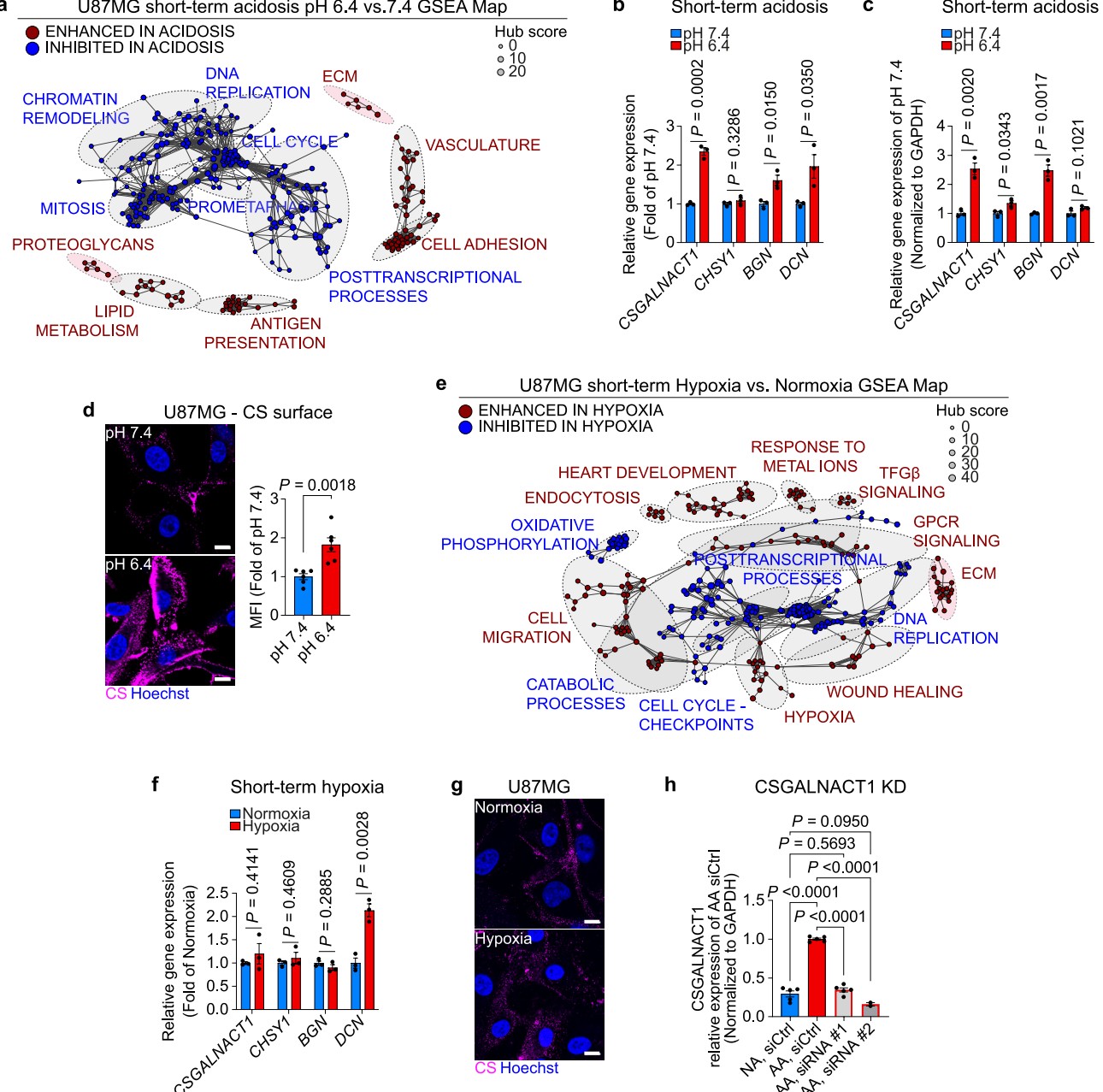

**Extended Data Fig. 4 | CS-glycocalyx encapsulation in short-term acidosis.**
**a**, GSEA enrichment mapping of significantly enhanced gene sets (≥ 5 gene sets/cluster with adj$P_v$ < 0.001) in U87MG cells after short-term acidosis (pH 6.4, 48 h) versus Ctrl (pH 7.4) (n = 3 biological replicates). **b** and **c**, Relative expression of key genes involved in CS-glycocalyx formation by mRNA array (**b**), and real-time qPCR analyses (**c**), from cells treated as in (**a**) (mean fold of pH 7.4 ± s.e.m., n = 3 biological replicates). **d**, Confocal imaging of CS surface signal in cells treated as in (**a**) (left; representative of ≥3 independent experiments), and corresponding quantification by flow cytometry (right; mean fold of pH 7.4 ± s.e.m., n = 6, 2 independent experiments). Scale bars: 10 µm. **e**, GSEA enrichment mapping of significantly enhanced gene sets (≥ 5 gene sets/cluster with adj$P_v$ < 0.001) in U87MG cells after short-term hypoxia (1% O2, 48 h) versus normoxia (21%

O2) (n = 3 biological replicates). **f**, Relative expression of key genes involved in CS-glycocalyx formation by mRNA array analyses in cells treated as in (**e**) (mean fold of normoxia ± s.e.m., n = 3 biological replicates). **g**, Confocal imaging of CS surface signal in cells treated as in (**e**) (representative of 2 independent experiments). Scale bars: 10 µm. **h**, Real-time qPCR analysis of CSGALNACT1 mRNA expression in U87MG AA and NA cells after siRNA CSGALNACT1 KD for 96 h (mean fold of AA siCtrl ± s.e.m., n = 2 (siRNA#2) and n = 5 (all other groups) biological replicates, each with 3 technical replicates). GSEA employed Hallmark, Reactome, KEGG, and GO databases (**a** and **e**), node size represents influence within the topology. CS surface signal was visualized via CS-56 antibody (**d** and **g**) and quantified via CS-56-AF488 (**d**). Significance was determined by two-sided t-test (**b-d** and **f**) or one-way ANOVA (**h**).

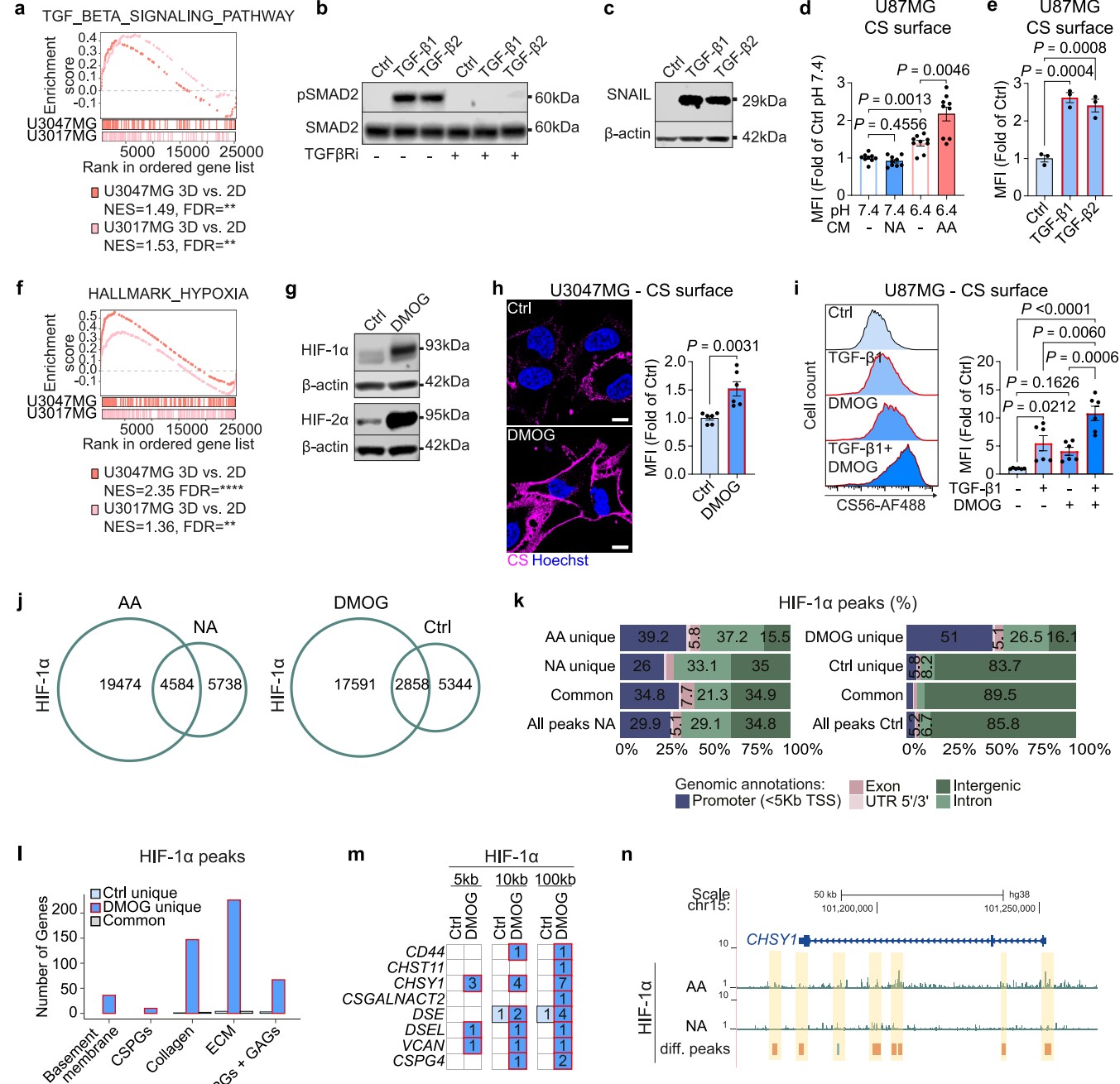

Extended Data Fig. 5 | See next page for caption.

**Extended Data Fig. 5 | CS-glycocalyx encapsulation during acidosis adaptation involves TGF-β and HIF signalling. a**, Significant enrichment of "TGF-β signalling pathway" genes in GBM 3D versus 2D cultures from U3047MG and U3017MG (*n* = 3 biological replicates). **b**, Immunoblotting for phosphorylated (Ser465/467) and total SMAD2 in U87MG cells after treatment with/without exogenous TGF-β1 or TGF-β2 (4 ng ml$^{-1}$, 6 h, pH 7.4), with/without TGFβRi (5 μM), as indicated (representative of 2 independent experiments). **c**, Immunoblotting for SNAIL in U87MG cells after treatment with/without exogenous TGF-β1 or TGF-β2 (4 ng ml$^{-1}$, 6 h, at pH 7.4) (representative of 2 independent experiments). β-actin was used as a loading control. **d**, Flow cytometry quantification of CS surface signal in U87MG cells after 48 h incubation in serum-free (SF, Ctrl) or conditioned medium (CM) from NA or AA cells (mean fold of Ctrl ± s.e.m., *n* = 9, 3 independent experiments). **e**, Flow cytometry quantification of CS surface signal in U87MG cells after treatment with/without TGF-β1 or TGF-β2 (4 ng ml$^{-1}$, 48 h, at pH 7.4) (mean fold of Ctrl ± s.e.m., *n* = 3 biological replicates). **f**, Significant enrichment of "hallmark hypoxia" genes in GBM 3D versus 2D cultures from U3047MG and U3017MG (*n* = 3 biological replicates). **g**, Immunoblotting for HIF-1α and HIF-2α in U87MG cells after treatment with/without DMOG (0.5 mM, 24 h, at pH 7.4) (representative of 2 independent experiments). β-actin was used as a loading control. **h**, Confocal imaging of CS surface signal in U3047MG cells after treatment with/without DMOG (1 mM, 72 h, at pH 7.4) (left; representative of 2 independent experiments), and corresponding flow cytometry quantification

(right; mean fold of Ctrl ± s.e.m., *n* = 6, 2 independent experiments). Scale bars: 10 μm. **i**, Flow cytometry representative histograms (left), and corresponding quantification (right), of CS surface signal in U87MG after 72 h treatment with TGF-β1 (1 ng ml$^{-1}$) and/or DMOG (0.5 mM) at pH 7.4, as indicated (mean fold of Ctrl ± s.e.m., *n* = 6, 2 independent experiments). **j**, Overlap of HIF-1α binding sites detected by CUT & RUN in U87MG AA and NA cells (left), or U87MG cells with/without DMOG treatment (0.5 mM, 72 h, at pH 7.4) (right). **k**, Genomic annotations of HIF-1α peaks as promoter ( < 5 kb from transcription start site, TSS), UTR 5'/3', exon, intron or intergenic regions across the indicated peak subsets in cells treated as in (**j**). **l**, Number of genes related to glycocalyx remodelling with HIF-1α peaks at promoter regions ( < 5 kb from TSS) in the indicated peak subsets (Ctrl unique, DMOG unique, common) in cells treated as in (**j**, right). **m**, HIF-1α binding sites in the proximity of genes of interest ( < 5 kb, <10 kb and <100 kb from TSS) in cells treated as in (**j**, right). **n**, Visualization of HIF-1α binding sites at the loci of CHSY1 in U87MG AA and NA cells. Yellow-shaded regions indicate promoters annotated by the European Promoter Database or regulatory elements defined by ENCODE. Differential peaks: gained (red) or lost (blue) in DMOG-treated vs. Ctrl cells, and invariable (grey). CS surface signal was quantified via CS-56-AF488 (**d**, **e**, **h** and **i**) and visualized via CS-56 antibody (**h**). Significance was determined by BH-adjusted nominal p-value (**a** and **f**), one-way ANOVA (**d**, **e** and **i**) or two-sided t-test (**h**). ** FDR < 0.05 and ****< 0.001 (**a** and **f**).

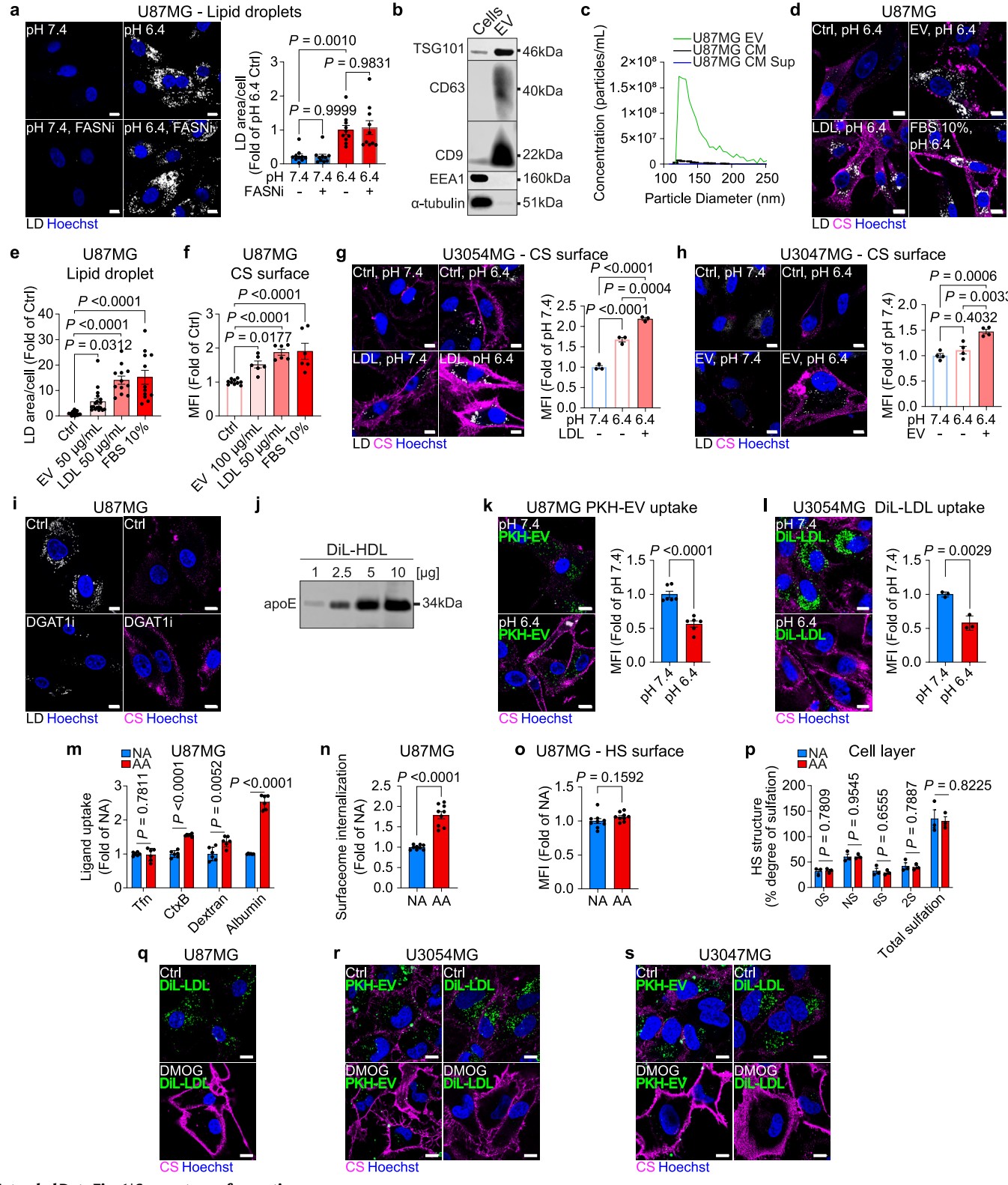

**Extended Data Fig. 6 | See next page for caption.**

**Extended Data Fig. 6 | CS-Glycocalyx is induced in response to exogenous lipid particles and prevents their uptake under acidic conditions. a**, Confocal imagining of LDs in U87MG cells after treatment with/without FASNi (50 μM, 72 h, at pH 6.4 or 7.4) (left; representative of 2 independent experiments), and corresponding quantification (right; mean fold of pH 6.4 Ctrl ± s.e.m., $n$ = 10 images/condition, representative of 2 independent experiments). Scale bars: 10 μm. **b**, Immunoblotting of EV markers (TSG101, CD63 and CD9), and cellular proteins (EEA1 and α-tubulin) in isolated EVs and corresponding cell lysates from U87MG cells (from 1 experiment). **c**, Nanoparticle analyses by Exoid-IZON of isolated EVs, conditioned medium (CM), and supernatant from EV-depleted CM (CM Sup) isolated from U87MG cells, show typical EV size distribution (50-200 nm). **d**, Confocal imaging of LDs and CS surface signal in U87MG cells grown in serum-free medium (Ctrl) or with exogenous lipids (48 h, at pH 6.4), as indicated (representative of ≥2 independent experiments). Scale bars: 10 μm. **e**, Corresponding quantification of LDs from (**d**) (mean fold of Ctrl ± s.e.m., $n$ = 24 (Ctrl), $n$ = 18 (EVs) and $n$ = 12 (LDL and FBS 10 %) images/condition, 2-4 independent experiments. **f**, Flow cytometry quantification of CS surface signal in U87MG cells treated as in (**d**) (mean fold of Ctrl ± s.e.m., $n$ = 10 (Ctrl) and $n$ = 6 (EVs, LDL and FBS 10 %), 3 or 2 independent experiments, respectively). **g** and **h**, Confocal imaging of LDs and CS surface signal (left; representative of ≥2 independent experiments), and corresponding flow cytometry quantification of CS surface signal (right), in U3054MG (**g**) and U3047MG (**h**) cells after short-term treatment with/without exogenous lipids (48-72 h, at pH 6.4 or pH 7.4) (mean fold of pH 7.4 ± s.e.m., $n$ = 3 (U3054MG) and $n$ = 4 (U3047MG) biological replicates). Scale bars: 10 μm. **i**, Confocal imaging of LDs and CS surface signal in U87MG cells treated with LDL (50 μg ml⁻¹, 48 h, at pH 6.4) with/without DGAT1i (10 μM) (representative of 2 independent experiments). Scale bars: 10 μm. **j**, Immunoblotting of DiL-HDL particle lysates (1-10 μg) confirms the presence of apoE (from 1 experiment). **k**, Confocal imaging of CS surface signal and PKH67-EV uptake (50 μg ml⁻¹, 1 h) in U87MG cells after short-term treatment at pH 6.4 or pH 7.4 (left; representative of 2 independent experiments), and corresponding flow cytometry quantification of PKH67-EV uptake (15 μg ml⁻¹, 1 h) (right; mean fold of pH 7.4 ± s.e.m., $n$ = 6, 2 independent experiments). Scale bars: 10 μm. **l**, Confocal imaging of CS surface signal and DiL-LDL uptake (20 μg ml⁻¹, 1 h) in U3054MG cells after 1 week treatment at pH 6.4 or pH 7.4 (left; representative of 2 independent experiments), and corresponding flow cytometry quantification of DiL-LDL uptake (20 μg ml⁻¹, 1 h) (right; mean fold of pH 7.4 ± s.e.m., $n$ = 3 biological replicates). Scale bars: 10 μm. **m**, Flow cytometry quantification of endocytosis marker uptake (Tfn 10 μg ml⁻¹, CtxB 5 μg ml⁻¹, Dextran 0.5 mg ml⁻¹, and Albumin 0.5 mg ml⁻¹; 2 h) in U87MG AA and NA cells (mean fold of NA ± s.e.m., $n$ = 6, 2 independent experiments). Tfn, Transferrin; CtxB, Cholera toxin-B. **n**, Flow cytometry quantification of biotinylated surfaceome internalization (2 h) in U87MG AA and NA cells (mean fold of NA ± s.e.m., $n$ = 9, 3 independent experiments). **o**, Flow cytometry quantification of HS surface signal in U87MG AA and NA cells (mean fold of NA ± s.e.m., $n$ = 9, 3 independent experiments). **p**, HS disaccharide analysis of cell lysates from U87MG AA and NA cells (mean % degree of sulphation ± s.e.m., $n$ = 3 biological replicates). **q**, Confocal imaging of CS surface signal and DiL-LDL uptake (50 μg ml⁻¹, 1 h) in U87MG pre-treated with/without DMOG (0.5 mM, 72 h, at pH 7.4) (representative of 2 independent experiments). Scale bars: 10 μm. **r** and **s**, Confocal imaging of CS surface signal and PKH67-EV or DiL-LDL uptake (50 μg ml⁻¹, 1 h) in U3054MG (**r**) or U3047MG (**s**) cells pre-treated with/without DMOG (1 mM, 72 h, at pH 7.4) (representative of 2 independent experiments). Scale bars: 10 μm. CS surface signal was visualized via CS-56 antibody (**d**, **g-i**, **k**, **l**, **q-s**) and quantified via CS-56-AF488 (**f** and **g**) or scFv GD3G7-AF488 (**h**). Surface HS was quantified via scFv AO4BO8-AF488 (**o**). Significance was determined by one-way ANOVA (**a**, **e-h**) or two-sided t-test (**k-p**).

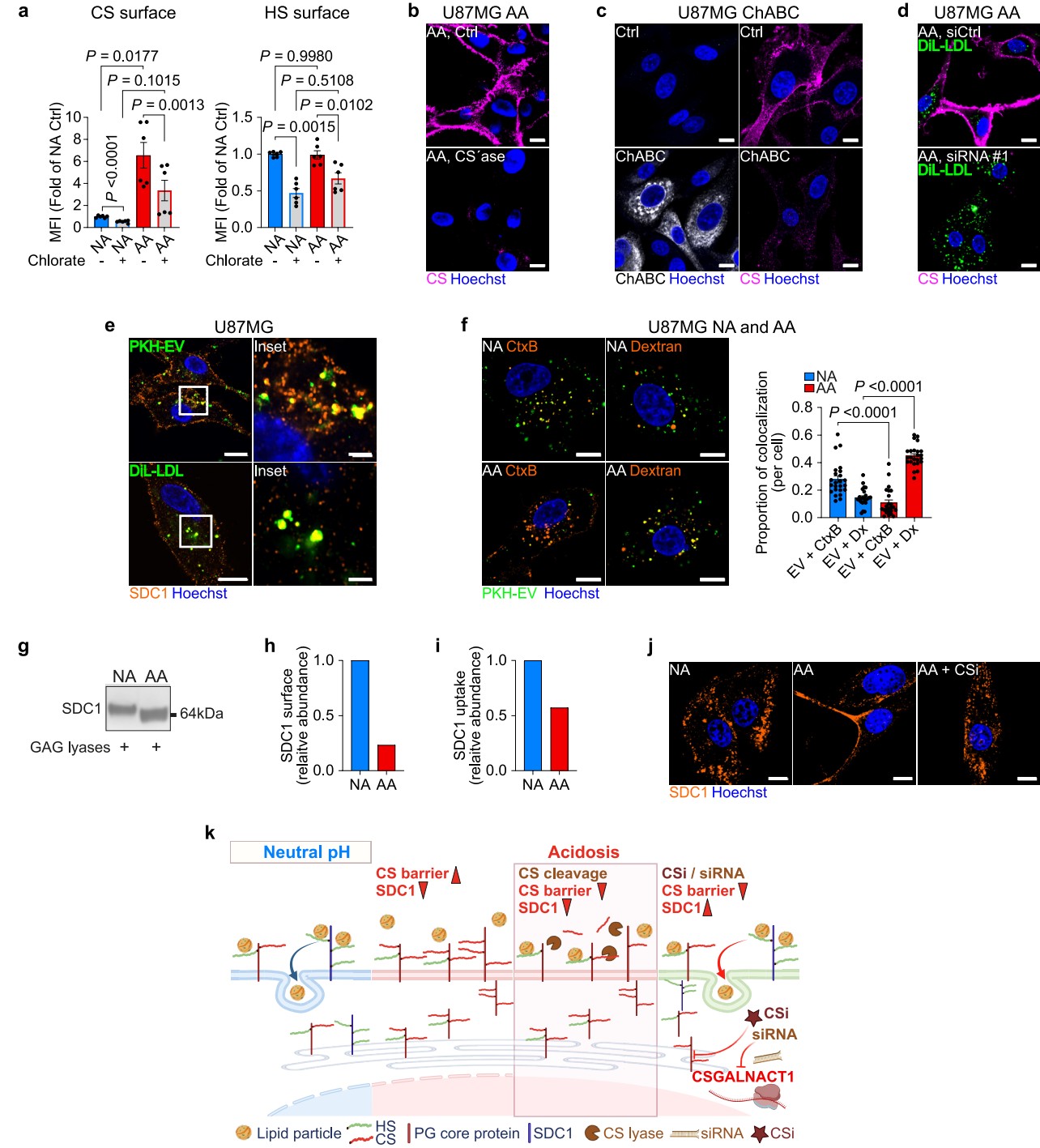

**Extended Data Fig. 7 | See next page for caption.**

**Extended Data Fig. 7 | Acidosis-induced CS-glycocalyx restricts lipid uptake through encapsulation and SDC1 glycan remodelling. a**, Flow cytometry quantification of CS surface signal (left), and HS surface signal (right) in U87MG AA and NA cells pre-treated with/without sodium chlorate (Chlorate, 25 mM, 24 h) (mean fold of NA Ctrl ± s.e.m., $n = 6$, 2 independent experiments). **b**, Confocal imaging of CS surface signal in U87MG AA cells treated with/without ABC/AC1 lyases (CS' ase; 6 h) (representative of 2 independent experiments). Scale bars: 10 μm. **c**, Confocal imaging of endogenous ChABC (left), or CS surface signal (right), in U87MG cells (48 h, at pH 6.4) (representative of 2 independent experiments). Scale bars: 10 μm. **d**, Confocal imaging of CS surface signal and DiL-LDL uptake (20 μg ml$^{-1}$, 1 h) in U87MG AA cells after siRNA KD of CSGALNACT1 (or Ctrl siRNA, siCtrl) for 96 h (representative of 2 independent experiments). Scale bars: 10 μm. **e**, Confocal imaging of PKH67-EV (50 μg ml$^{-1}$) or DiL-LDL (40 μg ml$^{-1}$) co-internalization (30 min) with anti-SDC1 antibody in U87MG cells (representative of 2 independent experiments). Scale bars: 10 and 2 μm (zoomed). **f**, Confocal imaging showing co-internalization of PKH67-EV (50 μg ml$^{-1}$) with markers of raft-mediated endocytosis (Cholera toxin-B, CtxB

25 μg ml$^{-1}$) or macropinocytosis (Dextran, Dx 1 mg ml$^{-1}$) in U87MG AA and NA cells (left), and corresponding quantification of relative co-localization (right; data are presented as mean proportion co-localization per cell ± s.e.m., $n = 24$ cells/condition, 2 independent experiments). Scale bars: 10 μm. **g**, Immunoblotting of SDC1 core protein isolated from U87MG AA and NA cell lysates and digested with GAG lyases (HS lyase III and ABC lyase) (from 1 experiment). **h** and **i**, Relative abundance ratio of SDC1, at the surface (**h**) or internalized for 2 h (**i**), as determined by LC-MS/MS proteomic analyses of biotinylated U87MG AA and NA cells ($n = 1$ biological replicate, from 3 technical replicates). **j**, Confocal imaging of SDC1 distribution in U87MG AA and NA cells pre-treated or not with CSi (2.5 mM, 48 h) (representative of 2 independent experiments). Scale bars: 10 μm. **k**, Schematic overview of strategies used to alleviate the CS-glycocalyx barrier to lipid particle binding and uptake. CS surface was quantified via CS-56-AF488 (**a**, left) and visualized by CS-56 antibody (**b-d**). Surface HS was quantified via scFv AO4BO8-AF488 (**a**, right). Squares indicate zoomed area (**e**). Significance was determined by one-way ANOVA (**a** and **f**). Illustration (**k**) was created with Biorender.com.

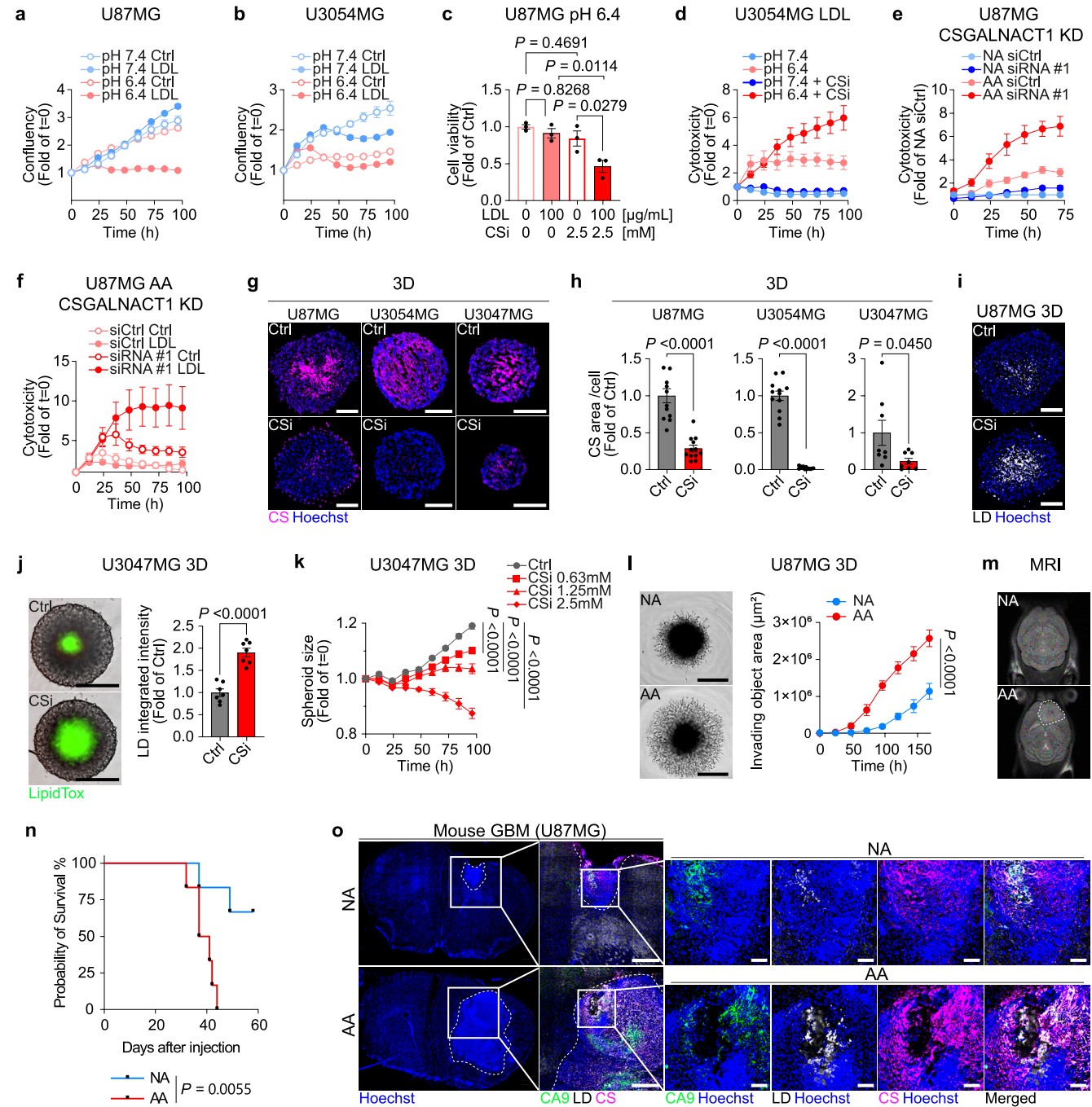

**Extended Data Fig. 8 | See next page for caption.**

**Extended Data Fig. 8 | CS-glycocalyx functions as a protective shield preventing lipid overload and cytotoxicity during acidosis adaptation. a** and **b**, Proliferation over time in U87MG (**a**) and U3054MG (**b**) cells challenged with/without high-dose LDL (100 or 50 µg ml$^{-1}$, for U87MG and U3054MG respectively; at pH 6.4 or 7.4) (mean fold of t = 0 ± s.e.m., $n$ = 3 biological replicates). **c**, MTT assay in U87MG cells pre-treated with CSi (48 h) prior to high-dose LDL challenge (24 h, at pH 6.4), as indicated (mean fold of Ctrl ± s.e.m., $n$ = 3 biological replicates). **d**, Cytotoxicity over time in U3054MG cells treated with CSi and low-dose LDL (at pH 6.4 or 7.4) (mean fold of t = 0 ± s.e.m., $n$ = 4 biological replicates). **e**, Cytotoxicity over time in U87MG AA and NA cells (10% FBS) after siRNA-mediated CSGALNACT1 KD (mean fold of NA siCtrl ± s.e.m., $n$ = 12, 2 independent experiments). **f**, Cytotoxicity over time in U87MG AA cells treated with/without low-dose LDL after siRNA-mediated CSGALNACT1 KD (mean fold of t = 0 ± s.e.m., $n$ = 6, 2 independent experiments). **g**, Fluorescence imaging of CS in U87MG, U3054MG and U3047MG 3D cultures after treatment with/without CSi (1.25 mM, 72 h) (representative of $n$ = 12 spheroids/condition). Scale bars: 200 µm. **h**, Quantification of CS area in (**g**) (mean fold of Ctrl ± s.e.m., $n$ = 12 (U87MG and U3054MG) and $n$ = 8 (U3047MG) spheroids/condition). **i**, Fluorescence imaging of LDs in U87MG 3D cultures treated as in (**g**) (representative of $n$ = 10 spheroids/condition). Scale bars: 200 µm. **j**, Incucyte images (left) of LipidTox accumulation in U3047MG spheroids treated as in (**g**), and corresponding quantification (right; mean fold of Ctrl ± s.e.m., $n$ = 7 spheroids/condition). Scale bars: 300 µm. **k**, Spheroid size over time in U3047MG 3D cultures treated with/without CSi (mean fold of t = 0 ± s.e.m., $n$ = 4 spheroids/condition, representative of 2 independent experiments). **l**, Incucyte images of spheroid invasion (at 5 days) (left), and corresponding quantification over time (right), in U87MG AA and NA 3D cultures (mean of invasive area ± s.e.m., $n$ = 8 spheroids/condition, representative of 3 independent experiments). Scale bars: 800 µm. **m**, MRI of U87MG NA (top) and AA (bottom) mouse xenograft tumours, 3 weeks after cell injections (representative of $n$ = 3 mice/group). Dashed lines delineate tumour border. **n**, Kaplan-Meier survival curves of mouse xenograft tumours from U87MG AA and NA cells ($n$ = 6 mice/group). **o**, Fluorescence imaging of tumour sections from U87MG NA (top) or AA (bottom) mouse xenografts, 3 weeks after cell injections, highlighting CA9$^+$/LD$^+$/CS$^+$ regions (representative of $n$ = 3 tumours/cell type). Dashed lines outline tumour border. Scale bars: 500 and 100 µm (zoomed). Data in (**a**, **b**, **d-f**, **j-l**) was acquired by IncuCyte live-cell imaging. CS was visualized via scFv clone GD3G7 (**g**, U87MG; and **o**) or CS-56 antibody (**g**, U3054MG and U3047MG). Significance was determined by one-way ANOVA (**c**), two-sided t-test (**h** and **j**), two-way ANOVA (**k** (at 96 h) and **l** (at 168 h)) or log-rank (Mantel-Cox) test (**n**).

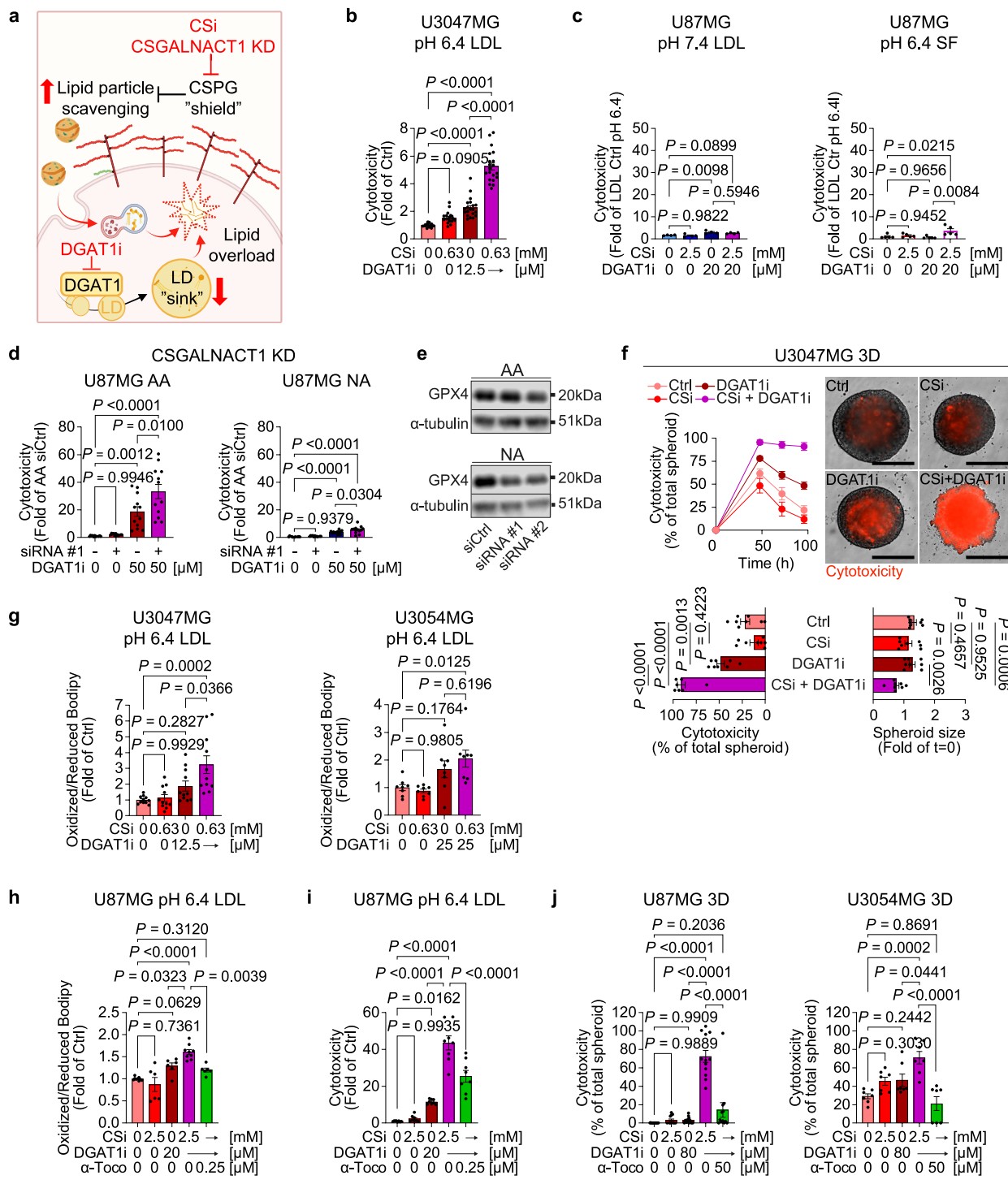

**Extended Data Fig. 9 | See next page for caption.**

**Extended Data Fig. 9 | Dual targeting of CS-glycocalyx and LD formation synergistically triggers lipid peroxidation and cell-death in acidic cancer cells. a**, Schematic illustration of the CS "shield" and LD "sink" dual targeting strategy. **b**, Cytotoxicity quantification at 120 h in U3047MG cells treated with CSi and/or DGAT1i at pH 6.4 in the presence of low-dose LDL, as indicated (mean fold of Ctrl ± s.e.m., $n = 20$, 4 independent experiments). **c**, Cytotoxicity quantification at 120 h in U87MG cells treated with CSi and/or DGAT1i, as indicated, at pH 7.4 in the presence of low-dose LDL (left), or at pH 6.4 in serum-free (SF) conditions (right) (mean fold of LDL Ctrl pH 6.4 ± s.e.m., $n = 4$ (pH 7.4) and $n = 5$ (SF, pH 6.4), 1 and 2 independent experiments, respectively). **d**, Cytotoxicity quantification at 120 h of combined effect of siRNA-mediated CSGALNACT1 KD and DGAT1i treatment in U87MG AA (left) and NA (right) cells, cultured in the presence of exogenous lipids (10 % FBS) (mean fold of AA siCtrl ± s.e.m., $n = 12$, 2 independent experiments). **e**, Immunoblotting of GPX4 in U87MG AA and NA after siRNA-mediated KD of CSGALNACT1 (by siRNA#1 and #2) or control siRNA (siCtrl) for 96 h (representative of 2 independent experiments). **f**, Cytotoxic effect of CSi (2.5 mM) and/or DGAT1i (80 μM) treatment in U3047MG 3D cultures. Cytotoxicity over time (top left), Incucyte images at 120 h (top right), and corresponding quantification of cytotoxicity and spheroid size at 120 h (bottom) (mean ± s.e.m., $n = 8$ spheroids/condition, 2 independent experiments). Scale bars: 400 μm. **g**, Quantification of cellular lipid peroxidation, measured as the ratio of oxidized to reduced Bodipy signal per cell, in U3047MG (left) and U3054MG (right) cells treated as in (**b**) (mean fold of Ctrl ± s.e.m., $n = 11$ (U3047MG) or $n = 8$ (U3054MG), 3 and 2 independent experiments, respectively). **h**, Quantification of cellular lipid peroxidation as in (**g**) of U87MG cells treated with CSi and/or DGAT1i at pH 6.4 in the presence of low-dose LDL, with/without addition of alpha-tocopherol (α-Toco), as indicated (mean fold of Ctrl ± s.e.m., $n = 9$ (Ctrl and CSi + DGAT1i) and $n = 6$ (CSi, DGAT1i and CSi + DGAT1i + α-Toco), 3 and 2 independent experiments, respectively). **i**, Cytotoxicity quantification at 120 h in U87MG cells treated as in (**h**) (mean fold of Ctrl ± s.e.m., $n = 8$, 2 independent experiments). **j**, Cytotoxicity quantification at 120 h in U87MG (left) and U3054MG (right) 3D cultures treated with CSi and/or DGAT1i with/without addition of alpha-tocopherol (α-Toco), as indicated (mean % of total spheroid area ± s.e.m., $n = 12$ (U87MG) and $n = 7$ (U3054MG) spheroids, 3 or 2 independent experiments respectively). Data in (**b-d**, **f-j**) was acquired by IncuCyte live-cell imaging. Significance was determined by one-way ANOVA (**b-d**, **f-j**). Illustration (**a**) was created with Biorender.com.

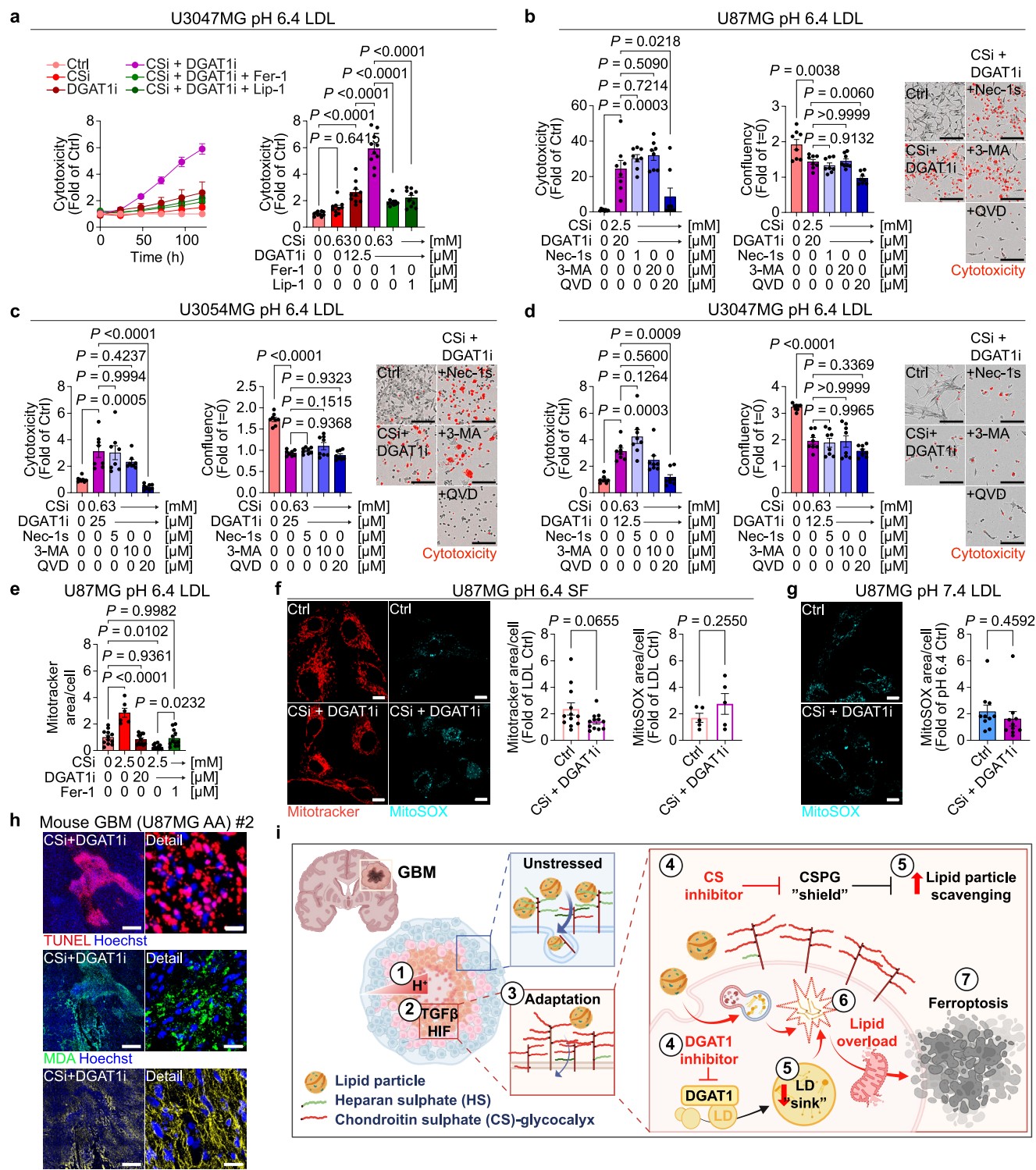

**Extended Data Fig. 10 | See next page for caption.**

**Extended Data Fig. 10 | Combined inhibition of CS-glycocalyx and LD formation triggers ferroptotic cell-death in acidic cancer cells. a**, Cytotoxicity over time (left), and corresponding quantification at 120 h (right), in U3047MG cells treated with CSi and/or DGAT1i at pH 6.4 in the presence of low-dose LDL, with/without addition of ferrostatin-1 (Fer-1) or liproxstatin-1 (Lip-1), as indicated (mean fold of Ctrl ± s.e.m., $n = 10$, 2 independent experiments). **b**–**d**, Cytotoxicity quantification (left), proliferation quantification (middle) and representative IncuCyte images (right), all at 120 h, in U87MG (**b**), U3054MG (**c**) and U3047MG (**d**) cells, treated with/without CSi and DGAT1i at pH 6.4 in the presence of low-dose LDL, with/without the addition of Necrostatin 1S (Nec-1s), 3-Methyladenine (3-MA) or Q-V-Oph (QVD), as indicated (mean fold of Ctrl ± s.e.m., $n = 8$, 2 independent experiments). Scale bars: 200 µm. **e**, Quantification of Mitotracker signal in U87MG cells after 30 h treatment with/without CSi and DGAT1i at pH 6.4 in the presence of low-dose LDL, with/without the addition ferrostatin-1 (Fer-1) (mean fold of Ctrl ± s.e.m., $n = 6$ (CSi) and $n = 12$ (all other groups) images/ group). **f**, Confocal imaging of U87MG cells treated as in (**e**) but with serum-free (SF) conditions visualizing mitochondria integrity by Mitotracker Red after 30 h of treatment or mitochondrial peroxidized lipids by MitoSOX after

26 h of treatment (left), and corresponding quantifications (middle and right; mean fold of LDL Ctrl ± s.e.m., $n = 12$ (Mitotracker) and $n = 5$ (MitoSOX) images/ group). Scale bars: 10 µm. **g**, Confocal imaging of MitoSOX signal (left), and corresponding quantification (right) of U87MG cells treated as in (**e**) but at neutral pH (7.4) (mean fold of pH 6.4 Ctrl ± s.e.m., $n = 10$ images/group). Scale bars: 10 µm. **h**, Fluorescence imaging of TUNEL, MDA and SLC7A11 staining in consecutive sections of U87MG AA xenograft mouse (#2) tumour treated with the combination of CSi (1.25 mM) and DGAT1i (80 µM) (representative of $n = 3$ mice). Scale bars: 500 and 20 µm (zoomed). **i**, Graphical abstract of sequence of events, illustrating how (1) tumour acidosis remodels the glycocalyx through a (2) HIF/TGFβ-driven switch, resulting in (3) a CS-glycocalyx barrier that limits lipid uptake and protects cells from ferroptosis. (4) Combined inhibition of CS biosynthesis (top) and lipid droplet (LD) formation (bottom), (5) restores lipid scavenging capacity (top) and disrupts the protective LD sink (bottom), triggering (6) lipid peroxidation and (7) ferroptotic death in tumour cells. Data in (**a**-**d**) was acquired by IncuCyte live-cell imaging. Significance was determined by one-way ANOVA (**a**-**e**) or two-sided t-test (**f** and **g**). Illustration (**i**) was created with Biorender.com.

# Reporting Summary

## Statistics

For all statistical analyses, confirm that the following items are present in the figure legend, table legend, main text, or Methods section.

| n/a | Confirmed | |
|---|---|---|
| ☐ | ☒ | The exact sample size (*n*) for each experimental group/condition, given as a discrete number and unit of measurement |
| ☐ | ☒ | A statement on whether measurements were taken from distinct samples or whether the same sample was measured repeatedly |
| ☐ | ☒ | The statistical test(s) used AND whether they are one- or two-sided<br>*Only common tests should be described solely by name; describe more complex techniques in the Methods section.* |
| ☐ | ☒ | A description of all covariates tested |
| ☐ | ☒ | A description of any assumptions or corrections, such as tests of normality and adjustment for multiple comparisons |
| ☐ | ☒ | A full description of the statistical parameters including central tendency (e.g. means) or other basic estimates (e.g. regression coefficient) AND variation (e.g. standard deviation) or associated estimates of uncertainty (e.g. confidence intervals) |
| ☐ | ☒ | For null hypothesis testing, the test statistic (e.g. *F*, *t*, *r*) with confidence intervals, effect sizes, degrees of freedom and *P* value noted<br>*Give P values as exact values whenever suitable.* |
| ☒ | ☐ | For Bayesian analysis, information on the choice of priors and Markov chain Monte Carlo settings |
| ☒ | ☐ | For hierarchical and complex designs, identification of the appropriate level for tests and full reporting of outcomes |
| ☐ | ☒ | Estimates of effect sizes (e.g. Cohen's *d*, Pearson's *r*), indicating how they were calculated |

*Our web collection on statistics for biologists contains articles on many of the points above.*

## Software and code

Policy information about availability of computer code

| | |
|---|---|
| Data collection | Four imaging platforms were used: LSM710 Airyscan confocal platform (Carl Zeiss AG, Oberkochen, Germany), operating under ZEN 2.1 (black); LSM980 confocal platform (Zeiss), operating under ZEN 3.8.2 (blue); Axio Scan.Z1 slide scanner (Zeiss), operating under ZEN 3.1 (blue); and Incucyte® S3 Live-Cell Analysis System (Sartorius), operating under the 2024B controller version (Sartorius).<br>Flow cytometry analyses were performed on Accuri C6 Flow Cytometer (BD Biosciences).<br>PCR was performed on StepOnePlus Real-Time PCR System (Applied Biosystems).<br>Extracellular vesicles (EVs) were characterized using the Exoid platform (Izon Science Ltd).<br>Laser Capture Microdissection (LCM) was performed on the Zeiss PALM system (Zeiss).<br>For CUT&RUN analyses, libraries were sequenced as PE150 on a NovaSeqX Sequencing System (Illumina).<br>Gene expression profiling was performed using either the Affymetrix Clariom D Pico Gene Array or the Illumina Human HT-12 v4 Expression BeadChip. |
| Data analysis | Confocal/scanner images were processed and quantified using the following softwares: ZEN 3.1 (blue), ImageJ software (v1.54p), CellProfiler (v4.2.1) and MATLAB (v2018a).<br>Incucyte S3 integrated software (v2022B Rev2 or v2024B) was used for analysis and visualization of Incucyte images.<br>Flow cytometry data were analyzed using BD CSampler™ Plus Software v1.0.27.1 (BD Biosciences) and FlowJo (v10).<br>Western blot images were processed by ImageJ software (v1.54p) or Image Studio Lite (v5.3.5).<br>For CUT&RUN analyses: Data were processed using FastQC (https://www.bioinformatics.babraham.ac.uk/projects/fastqc/), Trimmomatic (v0.39), Bowtie2 (v2.4.5), Subread (v2.1.1), SEACR (v1.3), and R (v4.x) with Bioconductor (v3.20). Gene annotation was performed using EnsDb.Hsapiens.v86. Data visualization was carried out using Galaxy (usegalaxy.org; v25.0.rc1) and the UCSC Genome Browser (hg38). |

For gene expression analysis: Gene array analyses were performed using R (v4.4.2) with RStudio, and the oligo (v1.70.0), affycoretools (v1.78.0), clariomdhumantranscriptcluster.db (v8.8.0), illuminaHumanv4.db (v1.26.0), and org.Hs.eg.db (v3.20.0) annotation packages. Differential gene expression analysis was performed using limma (v3.62.1) and the stats package (v4.4.2).
Gene set enrichment analysis (GSEA) was conducted using clusterProfiler (v4.14.4), msigdbr (v7.5.1), and igraph (v2.1.2).
Signature scoring was performed using hacksig (v0.1.2) and ggExtra (v0.10.0).
Gene expression data were visualized using ggplot2 (v3.5.1).
All R code and processed data scripts for mRNA array data are available from Zenodo at https://doi.org/10.5281/zenodo.17581666, which provides the full reproducible analysis pipeline.
R (v4.4.2) with RStudio using clusterProfiler (v4.14.4) and GraphPad Prism (v10.5.0) were used to create figures and perform statistical testing.
Schematics were created with BioRender.com.
Figure composition was performed with Adobe Illustrator v.28.6.

For manuscripts utilizing custom algorithms or software that are central to the research but not yet described in published literature, software must be made available to editors and reviewers. We strongly encourage code deposition in a community repository (e.g. GitHub). See the Nature Portfolio guidelines for submitting code & software for further information.

## Data

Policy information about availability of data

All manuscripts must include a data availability statement. This statement should provide the following information, where applicable:
- Accession codes, unique identifiers, or web links for publicly available datasets
- A description of any restrictions on data availability
- For clinical datasets or third party data, please ensure that the statement adheres to our policy

All data supporting the graphs in this paper, as well as all unprocessed blot images, are available in the Source Data files. Source data is available for Figs. 2c, f-h, 3b, d, g, i-k, 4a, b, d-h, j, k, 5a-c, e, g-j, 6a-e, g, h, j, 7a-f, 8a-c, e-g and Extended Data Figs. 1e, h, 3a, d-g, i, j, 4b-d, f, h, 5b-e, g-m, 6a-c, e-h, j-p, 7a, f-i, 8a-f, h, j-l, n, 9b-j, 10a-g. Additional quality control metrics and information for CUT & RUN analyses are provided in Supplementary Table 1. The mRNA array datasets generated have been deposited in the NCBI Gene Expression Omnibus (GEO) under accession codes GES300758, GSE300765, GSE300768 and GSE300771. The CUT & RUN datasets are available in GEO under accession code GSE300142. Imaging files and all other raw data files are available from the corresponding author (due to the size of this material).

## Research involving human participants, their data, or biological material

Policy information about studies with human participants or human data. See also policy information about sex, gender (identity/presentation), and sexual orientation and race, ethnicity and racism.

| Reporting on sex and gender | No exclusion criteria related to sex and gender were present for the study. |
|---|---|
| Reporting on race, ethnicity, or other socially relevant groupings | Not applicable. The experiments in this study did not involve reporting on race, ethnicity, or other socially relevant groupings. All human participant data used were analyzed in a manner that did not require stratification by these characteristics. |
| Population characteristics | Patients with glioma (WHO grade 2 to 4) or CNS metastasis (from kidney cancer, malignant melanoma or lung cancer) were collected from both male and female participants, age ≥ 18 years. |
| Recruitment | Clinical samples were collected from patients referred to the Neurosurgery Department at Lund University Hospital. Inclusion criteria were age 18 y or above, WHO performance status 0 to 4, and ability to give written informed consent. There was no self-selection or other bias in recruitment. Participation was voluntary, and no financial or other incentives were provided. Patients were diagnosed by routine MRI of the brain and surgical and pathological procedures, received standard oncological treatment, and were followed up according to local and national recommendations. |
| Ethics oversight | The study was carried out according to the ICH/GCP guidelines, in agreement with the Helsinki declaration, and approved by the local ethics committee, Lund University (Dnr. 454 2018/37). |

Note that full information on the approval of the study protocol must also be provided in the manuscript.

## Field-specific reporting

Please select the one below that is the best fit for your research. If you are not sure, read the appropriate sections before making your selection.

☒ Life sciences          ☐ Behavioural & social sciences          ☐ Ecological, evolutionary & environmental sciences

For a reference copy of the document with all sections, see nature.com/documents/nr-reporting-summary-flat.pdf

## Life sciences study design

All studies must disclose on these points even when the disclosure is negative.

| Sample size | No statistical methods were used to pre-determine sample sizes, but our sample sizes are similar to those reported in previous publications, |
|---|---|

| | |
|---|---|
| Sample size | as referenced in "Statistics and reproducibility" in the Method section.<br>Sample size and independent biological replicates are detailed in figure legends. A minimum of n = 2 replicates was applied for all assays. Briefly, all in vitro assays included a minimum of n = 3 biological replicates, defined as independent culture of cells, to ensure statistical robustness. For flow cytometry, at least 10,000 cells were analyzed per biological replicate.<br>Patient-derived sample sizes were determined based on feasibility and previous experience, with 2 to 10 independent samples used per experiment, depending on the assay. Key experiments were conducted using tissue from at least 10 glioblastoma (GBM) patients and RNA analysis was performed on pooled tissue from 5 GBM patients. |
| Data exclusions | No animals or other data points were excluded from the analyses. |
| Replication | All experiments involving in vitro cell culture were reliably reproduced and validated with at least three biological replicates in a minimum of two independent experiments, unless otherwise indicated in the figure legends. All replications were successful. When feasible, each independent experiment also included technical replicates. All results are presented as the mean ± either the standard error of the mean (SEM) or standard deviation (SD), as specified. For in vivo analyses, a minimum of n = 6 mice per group was used. |
| Randomization | For in vitro studies, randomization was not applicable; however, all cell lines/organoids were treated identically without prior designation. For in vivo mouse experiments involving drug treatment, same aged female mice were randomly assigned into experimental groups. |
| Blinding | Data collection and analysis were not performed blind to the conditions of the experiments.<br>However, data in all experiments were quantified in automated ways using various software. All processing conditions were applied uniformly across sample sets to avoid bias. In vitro treatment studies were analyzed and quantified in automated ways using Live-Cell Analysis System and Incucyte® S3 software. Microscopic images were captured randomly and analyzed in a blinded manner. In vivo data collection were not performed blinded as all cages were required to clearly label mouse treatment details. |

# Reporting for specific materials, systems and methods

We require information from authors about some types of materials, experimental systems and methods used in many studies. Here, indicate whether each material, system or method listed is relevant to your study. If you are not sure if a list item applies to your research, read the appropriate section before selecting a response.

## Materials & experimental systems

| n/a | Involved in the study |
|---|---|
| ☐ | ☒ Antibodies |
| ☐ | ☒ Eukaryotic cell lines |
| ☒ | ☐ Palaeontology and archaeology |
| ☐ | ☒ Animals and other organisms |
| ☒ | ☐ Clinical data |
| ☒ | ☐ Dual use research of concern |
| ☒ | ☐ Plants |

## Methods

| n/a | Involved in the study |
|---|---|
| ☐ | ☒ ChIP-seq |
| ☐ | ☒ Flow cytometry |
| ☐ | ☒ MRI-based neuroimaging |

## Antibodies

| | |
|---|---|
| Antibodies used | The following antibodies were used for: α-Tubulin (clone DM1A, ab7291, WB: 1:10,000), CD63 (clone MEM-259, ab8219, WB: 1:1,000), Syndecan-1 (clone EPR6454, ab128936, IF/Flow Cyt: 1:500, WB: 1:3,000), EEA1 (ab2900, WB: 1:1,000), Flotillin1 (ab41927, WB: 1:1,000), TSG101 (ab30871, WB: 1:1,000), β-actin (ab8227, WB: 1:10,000), CD9 (clone EPR2949, ab92726, WB: 1:1,000), GPX4 (clone EPNCIR144, ab125066, WB: 1:1,000); all from Abcam. Mouse CD31 (clone MEC 13.3, 553371 IF 1:100) from BD Biosciences. CA9 (clone M75, AB1001, IF: 1:200) from Bioscience Slovakia. CD68 (clone D4B9C, 76437, IF 1:800), HIF-2α (clone D6T8V, 59973, WB: 1:1,000), SNAIL (clone C15D3, 3879, WB: 1:2,000), total-SMAD2 (clone D43B4, 5339, WB: 1:2,000), Phospho-SMAD2 (Ser465/467) (clone 138D4, 3108, WB:1:2,000), TGF-β (3711, WB: 1:2,000); all from Cell Signaling. Human CD31 (Clone JC70A, M0823, IF: 1:50) from Dako. HIF-1α (GTX127309, WB: 1:1,000) from GeneTex. Malondialdehyde (clone 6H6, MA5-27559, IF: 1:50), SLC7A11 (clone A7C6-R, MA5-44922, IF: 1:200); both from Invitrogen. Chondroitinase ABC (ChABC) (clone 1E10, NBP1-96141, IF:100), apoE (clone WUE-4, NB110-60531, WB: 1:500); both from Novus Biologicals. CS (clone CS-56, C8035, IF/Flow Cyt: 1:200) from Sigma-Aldrich. Single chain fragment variable (scFv) HS (clone, AO4B08, IF/Flow Cyt: 1:50), CS (clone GD3G7, IF/Flow Cyt: 1:50), CS (clone IO3H10, IF/Flow Cyt: 1:50) (kindly provided by Dr. Toin H. van Kuppevelt) and used together with mouse anti-VSV (clone P5D4, V5507, IF/Flow Cyt: 1:500) or rabbit anti-VSV (V4888, IF/Flow Cyt: 1:500); all from Sigma-Aldrich.<br>The following secondary antibodies were used: Horseradish-peroxidase-conjugated anti-rabbit (7074, WB 1:10,000) from Cell Signaling or anti-mouse (a9044, WB: 1:10,000) from Sigma-Aldrich. Goat anti-mouse Alexa Fluor 488 (A1100, 1:500), Alexa Fluor 546 (A11030, 1:500), Alexa Fluor 647 (A21235, 1:500) or Goat anti-rabbit Alexa Fluor 488 (A11008, 1:500), Alexa Fluor 546 (A11010, 1:500), Alexa Fluor 647 (A21244, 1:500); Streptavidin Alexa Fluor 488 (S32354, 1:500), Streptavidin Alexa Fluor 546 (S11225, 1:500) or Streptavidin Alexa Fluor 647 (S21374, 1:500); all from Invitrogen. |
| Validation | All commercial antibodies have been validated by the manufacturer and used according to the applications recommended by their respective manufacturers.<br>Single chain fragment variable (scFv) anti-CS clone GD3G7, clone IO3H10 and anti-HS clone AO4B08, were validated in previous |

studies as indicated by the respective references under the Methods section "Compounds and antibodies".

# Eukaryotic cell lines

Policy information about cell lines and Sex and Gender in Research

| | |
|---|---|
| Cell line source(s) | Human Glioblastoma (GBM) cell line: U87MG was purchased from the American Type Culture Collection (ATCC, Cat# HTB-14). Human pancreatic cancer cell line: PANC-1 was purchased from ATCC (Cat# CRL-1469).<br>Human GBM primary cell cultures U3054MG, U3047MG, U3017MG were provided by HGCC, Uppsala. |
| Authentication | None of these cell lines were authentication in this study. |
| Mycoplasma contamination | All cell lines used in this study tested negative for mycoplasma contamination. |
| Commonly misidentified lines (See ICLAC register) | No commonly misidentified cell lines were used. |

# Animals and other research organisms

Policy information about studies involving animals; ARRIVE guidelines recommended for reporting animal research, and Sex and Gender in Research

| | |
|---|---|
| Laboratory animals | For all in vivo experiments of the study female NOD SCID gamma (NSG) mice, aged 5-7 weeks, obtained from the Jackson Laboratory (JAX), were used. GBM models included: (1) patient-derived xenograft (PDX) model using U3054MG cells, and (2) a cell line derived xenograft model using U87MG pH 7.4/NA or pH 6.4/AA cells.<br>All mice were housed in a specific pathogen-free facility with standard access to water and laboratory diet. Animals were group-housed under a 12-hour light/dark cycle, with an ambient temperature of 68–79°F and relative humidity of 30–70%.<br>Mice were monitored daily, and individuals were euthanized immediately when displaying symptoms of neurological distress. In selected experiments, mice were monitored using T2-weighted MRI scans on a 9.4T MRI system (Bruker). |
| Wild animals | No wild animals were used. |
| Reporting on sex | All experiments were performed using female mice only. |
| Field-collected samples | No field-collected samples were used in this study. |
| Ethics oversight | Experiments involving mouse orthotopic xenografts were approved by the Ethical Committee for Animal Research in Lund-Malmö (permit numbers 5.8.18-14006/2019 and 5.8.18-01073/2024) and were carried out according to national care regulations of the Swedish Board of Animal and European Union Animal Rights and Ethics Directives. |

Note that full information on the approval of the study protocol must also be provided in the manuscript.

# Plants

| | |
|---|---|
| Seed stocks | n/a |
| Novel plant genotypes | n/a |
| Authentication | n/a |

# ChIP-seq

## Data deposition

☒ Confirm that both raw and final processed data have been deposited in a public database such as GEO.

☐ Confirm that you have deposited or provided access to graph files (e.g. BED files) for the called peaks.

| | |
|---|---|
| Data access links<br>*May remain private before publication.* | The CUT&RUN datasets generated in this study have been deposited in GEO (Gene Expression Omnibus) under the accession number GSE300142 (https://www.ncbi.nlm.nih.gov/geo/query/acc.cgi?&acc=GSE300142). |

| Files in database submission | GSE300142 |
|---|---|
| Genome browser session (e.g. UCSC) | n/a |

## Methodology

| Replicates | Biological duplicates. |
|---|---|
| Sequencing depth | Library fragment size distribution was assessed via TapeStation High Sensitivity DNA Analysis assay, and libraries were sequenced as paired-end 150 bp (PE150) on a NovaSeqX Sequencing System (Illumina). |
| Antibodies | HIF-1α (GTX127309, GeneTex). |
| Peak calling parameters | Peak calling was performed using SEACR (1.3) with a stringent cutoff of FDR < 0.01. |
| Data quality | Raw sequencing files (FASTQ) were quality-checked using FastQC. |
| Software | FastQC (quality control), Trimmomatic v0.39 (adapter trimming), Bowtie2 v2.4.5 (alignment), samtools (duplicate removal), deepTools v3.5.5 (signal normalization and scaling), SEACR v1.3 (peak calling), featureCounts / SubRead v2.1.1 (FRiP calculation), ChIPpeakAnno, ChIPseeker (annotation and comparison), EnsDb.Hsapiens.v86, TxDb.Hsapiens.UCSC.hg38.knownGene (gene annotation), Galaxy / UCSC Genome Browser (visualization). |

# Flow Cytometry

## Plots

Confirm that:

☒ The axis labels state the marker and fluorochrome used (e.g. CD4-FITC).

☒ The axis scales are clearly visible. Include numbers along axes only for bottom left plot of group (a 'group' is an analysis of identical markers).

☐ All plots are contour plots with outliers or pseudocolor plots.

☒ A numerical value for number of cells or percentage (with statistics) is provided.

## Methodology

| Sample preparation | The samples used in this study were derived from cell cultures treated as described in the Methods section. Cells were detached using either 0.5 mM EDTA (for surface antigen analysis) or trypsin (for lipid particle uptake experiments). Further details are provided in the Methods section "Flow cytometry analysis". |
|---|---|
| Instrument | All analyses were performed with Accuri C6 Flow Cytometer (BD Biosciences). |
| Software | All flow cytometry data were analyzed by using BD CSampler™ Plus Software v1.0.27.1 (BD Biosciences) and FlowJo (v10). |
| Cell population abundance | At least 10,000 cells were analyzed per biological replicate. Depending on the experiment, PE, FITC or AlexaFluor488 signal were considered for analysis. |
| Gating strategy | Cells were gated to exclude dead cells/debris based on FSC-H/SSC-H. |

☐ Tick this box to confirm that a figure exemplifying the gating strategy is provided in the Supplementary Information.

# Magnetic resonance imaging

## Experimental design

| Design type | Animal brain imaging. |
|---|---|
| Design specifications | T2 weighted 9.4T imaging |
| Behavioral performance measures | n/a |

## Acquisition

| Imaging type(s) | Structural. |
|---|---|
| Field strength | 9.4 T |
| Sequence & imaging parameters | Imaging was performed on a 9.4 T Agilent magnet (Agilent, Santa Clara, USA) equipped with Bruker BioSpec AVIII |

| Sequence & imaging parameters | electronics operating with ParaVision (PV) 7.0.0 and a BGA 12S HP gradient system (Bruker, Ettlingen, Germany) and a mouse brain cryo coil was used. |
| | Scanning sequence, RARE, RARE factor 8; TR, 2200ms; TE, 35ms; FOV, 18x18 mm; Resolution, 53 x 53um2; Matrix size, 340 x 340; Slice thickness, 0,7mm; Number of slices, 8; Number of average, 4. |

Area of acquisition Brain.

Diffusion MRI ☐ Used ☒ Not used

## Preprocessing

| Preprocessing software | n/a |
| Normalization | n/a |
| Normalization template | n/a |
| Noise and artifact removal | n/a |
| Volume censoring | n/a |

## Statistical modeling & inference

| Model type and settings | n/a |
| Effect(s) tested | n/a |

Specify type of analysis: ☒ Whole brain ☐ ROI-based ☐ Both

| Statistic type for inference | n/a |

(See Eklund et al. 2016)

| Correction | n/a |

## Models & analysis

| n/a | Involved in the study |
|---|---|
| ☒ ☐ | Functional and/or effective connectivity |
| ☒ ☐ | Graph analysis |
| ☒ ☐ | Multivariate modeling or predictive analysis |

