## [Peer Review File · Nature Cell Biology]

Tumor Acidosis Remodels the Glycocalyx to Control Lipid Scavenging and Ferroptosis

Corresponding Author: Professor Mattias Belting

This manuscript has been previously reviewed at another journal. This document only contains information relating to versions considered at Nature Cell Biology.

Version 0:

Decision Letter:

*Please delete the link to your author homepage if you wish to forward this email to co-authors.

Dear Professor Belting,

Your manuscript, "Tumor Acidosis Remodels the Glycocalyx to Control Lipid Scavenging and Ferroptosis", has now been seen by 3 referees, who are experts in tumor acidity, glycobiology, cancer (referee 1); lipid biology, ferroptosis (referee 2); and cancer, lipid biology, ferroptosis (referee 3). As you will see from their comments (attached below) they find this work of potential interest, but have raised substantial concerns, which in our view would need to be addressed with considerable revisions before we can consider publication in Nature Cell Biology.

Nature Cell Biology editors discuss the referee reports in detail within the editorial team, including the chief editor, to identify key referee points that should be addressed with priority, and requests that are overruled as being beyond the scope of the current study. To guide the scope of the revisions, I have listed these points below. We are committed to providing a fair and constructive peer-review process, so please feel free to contact me if you would like to discuss any of the referee comments further.

In particular, it would be essential to:

A) Strengthen the in vivo relevance of the findings:

Reviewer 1

"...In vitro experiments have confirmed that an acidic environment is the main trigger for tumor cells to generate a CS-enriched glycocalyx, but this needs further validation in the tumor microenvironment. Can the author measure the acidity level of the tumor microenvironment to further confirm whether regions with CS-rich glycocalyx accumulation have lower pH values. Does the central region of in vitro 3D-cultured tumor cells exhibit a lower pH?"

Reviewer 2

"...While the contribution of EV to fatty acid transport in brains remains to be established, LDL definitely cannot cross BBB to reach brains. Thus, the choice of these fatty acid delivery vehicles makes it uncertain whether the mechanism they uncovered in vitro is physiologically relevant..."

B) Provide necessary controls as requested by Reviewer 3:

"...Because siRNA-mediated knockdown of GSGALNACT1 is used in this work, the authors should acknowledge this limitation and provide the controls demonstrating the extent to which this phenomenon could be impacting interpretation of the ferroptosis-pertinent results."

"In Fig. 6i and Fig. 6j, in order to demonstrate that the cytotoxicity is caused by ferroptosis in addition to the ferrostatin and liproxstatin controls, inhibitors of other forms of cell death (QVD, 3-MA, Nec-1s) should be included in order to demonstrate specificity of the cell death to ferroptosis (see PMID: 40204928 for reagents and dosages to include)"

C) Clarify potential data discrepancies:

Reviewer 1

"The study shows that tumor cells increase chondroitin sulfate (CS) expression in acidic environments, which limits fatty acid uptake and protects against ferroptosis caused by lipid peroxidation. They also showed that Short-term acidosis was sufficient to increase cell-surface CS levels. However, Figure 5 indicates that combining acidic conditions (pH 6.4) with LDL results in enhanced tumor cell death. Given that acidity-induced CS production should reduce cytotoxicity by limiting lipid uptake, why does this combination exacerbate cell

death?"

Reviewer 2

"The manuscript contains parts that are in conflict with each other. For example, in figure 1, the authors left an impression of the existence of LD⁺/CS⁺ glioblastoma, and used the increased expression of FASN as the gene expression signature that marked for the tumor cells accumulating LDs. However, in figure 3, they showed images that the accumulation of LDs and CS was mutually exclusive, and in extended data Fig. 6 they concluded that FASN was not required for LD accumulation. These inconsistencies make the manuscript difficult to understand."

D) All other referee concerns pertaining to strengthening existing data, providing controls, methodological details, clarifications and textual changes, should also be addressed.

E) Finally please pay close attention to our guidelines on statistical and methodological reporting (listed below) as failure to do so may delay the reconsideration of the revised manuscript. In particular please provide:

We would be happy to consider a revised manuscript that would satisfactorily address these points, unless a similar paper is published elsewhere, or is accepted for publication in Nature Cell Biology in the meantime.

- ensure that it conforms to our format instructions and publication policies (see below and <https://www.nature.com/nature/for-authors>).

- provide a point-by-point rebuttal to the full referee reports verbatim, as provided at the end of this letter.

- for any revision that includes light microscopy data, we ask our authors to please include a completed light microscopy reporting table https://www.nature.com/documents/Light_microscopy_reporting_table.xlsx to ensure the methods are described thoroughly. The table will be available to reviewers and ultimately published should the manuscript be accepted at the journal.

- provide the completed Reporting Summary (found here <https://www.nature.com/documents/nr-reporting-summary.pdf>) <https://www.nature.com/documents/nr-reporting-summary.pdf>). This is essential for reconsideration of the manuscript will be available to editors and referees in the event of peer review. For more information see <http://www.nature.com/authors/policies/availability.html> or contact me.

When submitting the revised version of your manuscript, please pay close attention to our [href="https://www.nature.com/nature-portfolio/editorial-policies/image-integrity">Digital Image Integrity Guidelines.](https://www.nature.com/nature-portfolio/editorial-policies/image-integrity) and to the following points below:

Nature Cell Biology is committed to improving transparency in authorship. As part of our efforts in this direction, we are now requesting that all authors identified as 'corresponding author' on published papers create and link their Open Researcher and Contributor Identifier (ORCID) with their account on the Manuscript Tracking System (MTS), prior to acceptance. ORCID helps the scientific community achieve unambiguous attribution of all scholarly contributions. You can create and link your ORCID from the home page of the MTS by clicking on 'Modify my Springer Nature account'. For more information please visit <http://www.springernature.com/orcid>.

This journal strongly supports public availability of data. Please place the data used in your paper into a public data repository, or alternatively, present the data as Supplementary Information. If data can only be shared on request, please explain why in your Data Availability Statement, and also in the correspondence with your editor. Please note that for some data types, deposition in a public repository is mandatory - more information on our data deposition policies and available repositories appears below.

Link Redacted

We would like to receive a revised submission within six months.

We hope that you will find our referees' comments, and editorial guidance helpful. Please do not hesitate to contact me if there is anything you would like to discuss.

Best wishes,

Zhe Wang

Zhe Wang, PhD
Senior Editor
Nature Cell Biology

Tel: +44 (0) 207 843 4924
email: zhe.wang@nature.com

Reviewers' Comments:

Reviewer #1 (Remarks to the Author):

The manuscript "Tumour Acidosis Remodels the Glycocalyx to Control Lipid Scavenging and Ferroptosis" investigates how tumor cells adapt to acidosis and hypoxia microenvironmental stress reprogram cellular metabolism to regulate lipid scavenging and ferroptosis in glioblastoma (GBM) and CNS metastases. They identified a chondroitin sulphate (CS)-enriched glycocalyx as a key feature of the acidic tumor microenvironment, which surrounds tumor cells, restricts extracellular lipid uptake, and shields against lipid-induced ferroptosis. This CS-rich glycocalyx arises from a glycan switch on syndecan-1, driven by HIF and TGF β signaling, replacing heparan sulphate with CS and impairing lipid scavenging. They further show that dual inhibition of CS biosynthesis and diacylglycerol-O-acyltransferase 1 triggers lipid peroxidation and ferroptotic cell death. Overall, the compelling findings highlight the CS-glycocalyx's role in metabolic adaptation and suggest novel therapeutic vulnerabilities in aggressive cancers.

Comments:

1. This study primarily found that tumor cells in an acidic environment can produce a chondroitin sulphate (CS)-enriched glycocalyx to block the uptake of additional lipids by cells. In vitro experiments have confirmed that an acidic environment is the main trigger for tumor cells to generate a CS-enriched glycocalyx, but this needs further validation in the tumor microenvironment. Can the author measure the acidity level of the tumor microenvironment to further confirm whether regions with CS-rich glycocalyx accumulation have lower pH values. Does the central region of in vitro 3D-cultured tumor cells exhibit a lower pH?
2. This study demonstrates that the co-localization of CS-glycocalyx accumulation and lipid stress serves as a hallmark of metabolically challenged regions in aggressive tumors. However, is there a correlation between CS-glycocalyx enrichment and LD accumulation? In Extended Data Fig. 1e, which depicts GBM patient specimens, many regions with high CS-glycocalyx expression lack LDs.
3. Similarly, TGF- β is recognized as a key factor in inducing CS production. Is there a correlation between TGF- β expression and CS levels in the tumor microenvironment?
4. In Figure 1H, the authors demonstrated that in hypoxic regions, tumor cells (CA9+/CD31-) are enriched with lipid droplets (LD) and chondroitin sulfate (CS). Additionally, vascular regions (CA9-/CD31+) also exhibit high CS expression despite lacking LD. How do these vascular endothelial cells highly express CS? Is high CS expression specific to tumor vascular endothelial cells? What is the significance of this for the vasculature itself and tumor growth? A discussion of these aspects would be valuable.
5. In Extended Data Fig. 2, the authors showed that CNS metastases from kidney, melanoma and lung primaries harboured CS-rich cells in perinecrotic and perivascular compartments. Do these non-glioma primary tumors also harbor CS-rich cells?
6. Extended Data Fig. 5f showed that inhibiting TGF- β receptors reduces acidosis-driven CS-glycocalyx formation, a finding that strongly supports TGF- β 's role in inducing CS production. I suggest moving this result to a main Figure to highlight its significance.
7. The study shows that tumor cells increase chondroitin sulfate (CS) expression in acidic environments, which limits fatty acid uptake and protects against ferroptosis caused by lipid peroxidation. They also showed that Short-term acidosis was sufficient to increase cell-surface CS levels. However, Figure 5 indicates that combining acidic conditions (pH 6.4) with LDL results in enhanced tumor cell death. Given that acidity-induced CS production should reduce cytotoxicity by limiting lipid uptake, why does this combination exacerbate cell death?
8. In addition to impeding fatty acid uptake, does the CS-glycocalyx also block the uptake of other substances by cells?

Reviewer #2 (Remarks to the Author):

In this manuscript, the authors showed that some glioblastoma exhibited higher levels of CS and LDs. They recapitulated this finding in vitro through culturing glioblastoma cancer cell lines under 3-D or acidic pH. Using EV and LDL as the fatty acid carriers, the authors concluded that these metabolic adaptations prevent lipid peroxidation thereby protecting cancer cells from ferroptosis. While the individual piece of the data looks impressive, the story as a whole lacks cohesiveness. More importantly, the physiological relevance of their findings is questionable owing to their choice of vehicles for the fatty acid delivery.

1. The authors relied on exogenous fatty acids delivered through LDL or EV to determine the mechanism behind their findings. While the contribution of EV to fatty acid transport in brains remains to be established, LDL definitely cannot cross BBB to reach brains. Thus, the choice of these fatty acid delivery vehicles makes it uncertain whether the mechanism they uncovered in vitro is physiologically relevant. According to the current knowledge, brain-derived lipoproteins, which contain apoE but not apoB, the primary component of LDL, are the most important fatty acid transporters in brains. The authors may want to reexamine their findings by using these particles to deliver exogenous fatty acids.
2. The manuscript contains parts that are in conflict with each other. For example, in figure 1, the authors left an impression of the existence of LD+/CS+ glioblastoma, and used the increased expression of FASN as the gene expression signature that marked for the tumor cells accumulating LDs. However, in figure 3, they showed images that the accumulation of LDs and CS was mutually exclusive, and in extended data Fig. 6 they concluded that FASN was not required for LD accumulation. These inconsistencies make the manuscript difficult to understand.
3. The authors employed multiple conditions to analyze the accumulation of CS and LDs in the tumor cells such as 3D culture, lower pH,

hypoxia-mimicking treatment, and TGF β without giving a clear picture of the relationship among these events in tumor microenvironment. A diagram illustrating the sequence of these events will markedly help readers to understand these studies.

Reviewer #3 (Remarks to the Author):

Bang-Rudenstam et al. use analyses of patient tumors, patient-derived 3D models, and in vivo systems to identify chondroitin sulphate (CS)-enriched glycocalyx as a hallmark of the acidic tumor microenvironment in glioblastoma and CNS metastases. They go on to show that this glycocalyx can protect cancer cells from ferroptosis, and that targeting of CS and DGAT-1 is capable of inducing cancer cell death by ferroptosis. Although lipid droplet accumulation has been demonstrated before in glioblastoma models, these findings are novel because they demonstrate a targetable dependency of glioblastoma and CNS metastases via glycan remodelling. The implications of this stress-induced glycosylation program in the patient and human PDC-derived models demonstrating potential translatability of these findings will be of great interest to the field. Prior to publication, a few technical concerns should be considered:

Major comments

1. Recent work has shown that siRNA treatment can lead to GPX4 upregulation and ferroptosis sensitization (PMID: 38489367). Because siRNA-mediated knockdown of GSGALNACT1 is used in this work, the authors should acknowledge this limitation and provide the controls demonstrating the extent to which this phenomenon could be impacting interpretation of the ferroptosis-pertinent results.
2. In Fig. 6i and Fig. 6j, in order to demonstrate that the cytotoxicity is caused by ferroptosis in addition to the ferrostatin and liproxstatin controls, inhibitors of other forms of cell death (QVD, 3-MA, Nec-1s) should be included in order to demonstrate specificity of the cell death to ferroptosis (see PMID: 40204928 for reagents and dosages to include)
3. BODIPY stains shown in Fig. 6g, h should be represented a ratio of oxidized to reduced dye and representative flow cytometry plots of these plots should be included
4. In Fig. 9l, it is demonstrated that these effects required the presence of extracellular lipids. Since the types of extracellular lipids (MUFA versus PUFA) make a difference in terms of ferroptosis sensitivities and it would be important to add back either MUFA or PUFA in the serum-free conditions to determine if exogenous MUFA might be able to protect from ferroptosis in this context.
5. For the pilot trial that is references in lines 340-342, this data should be shown (and repeated with greater n) or this sentence should be removed from the manuscript text. The ability to include this data would greatly strengthen the impact of this work.

Minor comments

1. Line 347 - MDA and SLC7A11 would be more appropriately referred to as "markers associated with ferroptosis" as they are not considered ferroptotic markers
2. In Fig. 3e and ED Fig. 8h (U3047MG), presumably the control cohort would be set to 1.0, but this does not appear to be the case. Please check.
3. In the discussion, these findings should be further contextualized and discussed in relation to PMID: 34118189, which has shown an effect of the acidic tumor microenvironment on ferroptosis.

ABSTRACT AND MAIN TEXT – please follow the guidelines that are specific to the format of your manuscript, as listed in our Guide to Authors (http://www.nature.com/ncb/pdf/ncb_gta.pdf) Briefly, Nature Cell Biology Articles, Resources and Technical Reports have 3500 words, including a 150 word abstract, and the main text is subdivided in Introduction, Results, and Discussion sections. Nature Cell

Biology Letters have up to 2500 words, including a 180 word introductory paragraph (abstract), and the text is not subdivided in sections.

Methods should be written concisely, but should contain all elements necessary to allow interpretation and replication of the results. As a guideline, Methods sections typically do not exceed 3,000 words. The Methods should be divided into subsections listing reagents and techniques. When citing previous methods, accurate references should be provided and any alterations should be noted. Information must be provided about: antibody dilutions, company names, catalogue numbers and clone numbers for monoclonal antibodies; sequences of RNAi and cDNA probes/primers or company names and catalogue numbers if reagents are commercial; cell line names, sources and information on cell line identity and authentication. Animal studies and experiments involving human subjects must be reported in detail, identifying the committees approving the protocols. For studies involving human subjects/samples, a statement must be included confirming that informed consent was obtained. Statistical analyses and information on the reproducibility of experimental results should be provided in a section titled "Statistics and Reproducibility".

All Nature Cell Biology manuscripts submitted on or after March 21 2016 must include a Data availability statement as a separate section after Methods but before references, under the heading "Data Availability". For Springer Nature policies on data availability see <http://www.nature.com/authors/policies/availability.html>; for more information on this particular policy see <http://www.nature.com/authors/policies/data/data-availability-statements-data-citations.pdf>. The Data availability statement should include:

- Accession codes for primary datasets (generated during the study under consideration and designated as "primary accessions") and secondary datasets (published datasets reanalysed during the study under consideration, designated as "referenced accessions"). For primary accessions data should be made public to coincide with publication of the manuscript. A list of data types for which submission to community-endorsed public repositories is mandated (including sequence, structure, microarray, deep sequencing data) can be found here <http://www.nature.com/authors/policies/availability.html#data>.
- Unique identifiers (accession codes, DOIs or other unique persistent identifier) and hyperlinks for datasets deposited in an approved repository, but for which data deposition is not mandated (see here for details <http://www.nature.com/sdata/data-policies/repositories>).
- At a minimum, please include a statement confirming that all relevant data are available from the authors, and/or are included with the manuscript (e.g. as source data or supplementary information), listing which data are included (e.g. by figure panels and data types) and mentioning any restrictions on availability.
- If a dataset has a Digital Object Identifier (DOI) as its unique identifier, we strongly encourage including this in the Reference list and citing the dataset in the Methods.

We recommend that you upload the step-by-step protocols used in this manuscript to [protocols.io](https://www.protocols.io). More details can be found at <https://www.protocols.io/help/publish-articles>.

All imaging data should be accompanied by scale bars, which should be defined in the legend.

Cropped images of gels/blots are acceptable, but need to be accompanied by size markers, and to retain visible background signal within the linear range (i.e. should not be saturated). The boundaries of panels with low background have to be demarked with black lines. Splicing of panels should only be considered if unavoidable, and must be clearly marked on the figure, and noted in the legend with a statement on whether the samples were obtained and processed simultaneously. Quantitative comparisons between samples on different gels/blots are discouraged; if this is unavoidable, it should only be performed for samples derived from the same experiment with gels/blots were processed in parallel, which needs to be stated in the legend.

- For line art, graphs, charts and schematics we prefer Adobe Illustrator (.AI), Encapsulated PostScript (.EPS) or Portable Document Format (.PDF). Files should be saved or exported as such directly from the application in which they were made, to allow us to restyle them according to our journal house style.
- We accept PowerPoint (.PPT) files if they are fully editable. However, please refrain from adding PowerPoint graphical effects to objects, as this results in them outputting poor quality raster art. Text used for PowerPoint figures should be Helvetica (preferred) or Arial.
- We do not recommend using Adobe Photoshop for designing figures, but we can accept Photoshop generated (.PSD or .TIFF) files only if each element included in the figure (text, labels, pictures, graphs, arrows and scale bars) are on separate layers. All text should be editable in 'type layers' and line-art such as graphs and other simple schematics should be preserved and embedded within 'vector smart objects' - not flattened raster/bitmap graphics.
- Some programs can generate Postscript by 'printing to file' (found in the Print dialogue). If using an application not listed above, save the file in PostScript format or email our Art Editor, Allen Beattie for advice (a.beattie@nature.com).

EXTENDED DATA FIGURES - When re-submitting your manuscript, please ensure that any supplementary figures and tables that are crucial to the manuscript's conclusions are converted into Extended Data figures and tables to increase visibility of these data. Extended Data figures and tables are online-only (present in the online PDF and full-text HTML versions of the paper), peer-reviewed display items that provide essential background to the article but are not included in the main article due to space constraints. A maximum of ten Extended Data display items (figures and tables) is permitted.

The total number of Supplementary Figures (not including the "unprocessed scans" Supplementary Figure) should not exceed the number of main display items (figures and/or tables (see our Guide to Authors and March 2012 editorial <http://www.nature.com/ncb/authors/submit/index.html#suppinfo>; <http://www.nature.com/ncb/journal/v14/n3/index.html#ed>). No restrictions apply to Supplementary Tables or Videos, but we advise authors to be selective in including supplemental data.

GUIDELINES FOR EXPERIMENTAL AND STATISTICAL REPORTING

REPORTING REQUIREMENTS – We are trying to improve the quality of methods and statistics reporting in our papers. To that end, we are now asking authors to complete a reporting summary that collects information on experimental design and reagents. The Reporting Summary can be found here <https://www.nature.com/documents/nr-reporting-summary.pdf> <https://www.nature.com/documents/nr-reporting-summary.pdf> If you would like to reference the guidance text as you complete the template, please access these flattened versions at <http://www.nature.com/authors/policies/availability.html>.

Version 1:

Decision Letter:

*Please delete the link to your author homepage if you wish to forward this email to co-authors.

Dear Professor Belting,

Your manuscript, "Tumor Acidosis Remodels the Glycocalyx to Control Lipid Scavenging and Ferroptosis", has now been seen by the original referees. As you will see from their comments (attached below) they find this work of interest, but have raised some important points. Although we are also very interested in this study, we believe that their concerns should be addressed before we can consider publication in Nature Cell Biology.

Nature Cell Biology editors discuss the referee reports in detail within the editorial team, including the chief editor, to identify key referee points that should be addressed with priority, and requests that are overruled as being beyond the scope of the current study. To guide the scope of the revisions, I have listed these points below. We are committed to providing a fair and constructive peer-review process, so please feel free to contact me if you would like to discuss any of the referee comments further.

In particular, it would be essential to:

A) Address the remaining concerns from Reviewer 2.

B) Finally please pay close attention to our guidelines on statistical and methodological reporting (listed below) as failure to do so may delay the reconsideration of the revised manuscript. In particular please provide:

We therefore invite you to take these points into account when revising the manuscript. In addition, when preparing the revision please:

- ensure that it conforms to our format instructions and publication policies (see below and www.nature.com/nature/authors/).

- provide a point-by-point rebuttal to the full referee reports verbatim, as provided at the end of this letter.

Nature Cell Biology is committed to improving transparency in authorship. As part of our efforts in this direction, we are now requesting that all authors identified as 'corresponding author' on published papers create and link their Open Researcher and Contributor Identifier (ORCID) with their account on the Manuscript Tracking System (MTS), prior to acceptance. ORCID helps the scientific community achieve unambiguous attribution of all scholarly contributions. You can create and link your ORCID from the home page of the MTS by clicking on 'Modify my Springer Nature account'. For more information please visit <http://www.springernature.com/orcid>.

Link Redacted

We would like to receive the revision within four weeks. If submitted within this time period, reconsideration of the revised manuscript will not be affected by related studies published elsewhere, or accepted for publication in Nature Cell Biology in the meantime. We would be happy to consider a revision even after this timeframe, but in that case we will consider the published literature at the time of resubmission when assessing the file.

We hope that you will find our referees' comments, and editorial guidance helpful. Please do not hesitate to contact me if there is anything you would like to discuss.

Best wishes,

Zhe Wang

Zhe Wang, PhD
Senior Editor
Nature Cell Biology

Tel: +44 (0) 207 843 4924
email: zhe.wang@nature.com

Reviewers' Comments:

Reviewer #1 (Remarks to the Author):

The authors have successfully addressed my previous concerns.

Reviewer #2 (Remarks to the Author):

The revised manuscript addressed some of my previous concerns but the key point, that is, the physiological relevancy of the lipoprotein particles used as the vehicle for fatty acid delivery remains unresolved. The authors argued that GBM may affect integrity of BBB thereby allowing LDL to reach brains. However, the paper they cited in the rebuttal letter only said that GBM altered transportation across BBB but did not say that BBB was so disintegrated that particles as large as LDL could reach brains in patients with GBM. The new experiment, which used HDL, also suffered from the same problem, as HDL cannot cross BBB either. The brain-derived lipoprotein particles, which are assembled by apoE, are NOT HDL, which is assembled by apoA1, even though the brain-derived lipoprotein particles are often referred to HDL-like. The authors may want to use in vitro lipidated apoE (PMID: 39532095) or lipoproteins secreted by astrocytes (PMID: 35235798) to test their hypothesis.

Reviewer #3 (Remarks to the Author):

The authors have sufficiently addressed my prior review comments, and the added experiments and data greatly strengthen the manuscript.

GUIDELINES FOR SUBMISSION OF NATURE CELL BIOLOGY ARTICLES

ARTICLE FORMAT

ABSTRACT – should not exceed 150 words and should be unreferenced. This paragraph is the most visible part of the paper and should briefly outline the background and rationale for the work, and accurately summarize the main results and conclusions. Key genes, proteins and organisms should be specified to ensure discoverability of the paper in online searches.

TEXT – the main text consists of the Introduction, Results, and Discussion sections and must not exceed 3500 words including the abstract. The Introduction should expand on the background relating to the work. The Results should be divided in subsections with subheadings, and should provide a concise and accurate description of the experimental findings. The Discussion should expand on the findings and their implications. All relevant primary literature should be cited, in particular when discussing the background and specific findings.

REFERENCES – are limited to a total of 70 in the main text and Methods combined. They must be numbered sequentially as they appear in the main text, tables and figure legends and Methods and must follow the precise style of Nature Cell Biology references. References only cited in the Methods should be numbered consecutively following the last reference cited in the main text. References only associated with Supplementary Information (e.g. in supplementary legends) do not count toward the total reference limit and do not need to be cited in numerical continuity with references in the main text. Only published papers can be cited, and each publication cited should be included in the numbered reference list, which should include the manuscript titles. Footnotes are not permitted.

Methods should be written concisely, but should contain all elements necessary to allow interpretation and replication of the results. As a guideline, Methods sections typically do not exceed 3,000 words. The Methods should be divided into subsections listing reagents and techniques. When citing previous methods, accurate references should be provided and any alterations should be noted. Information must be provided about: antibody dilutions, company names, catalogue numbers and clone numbers for monoclonal antibodies; sequences of RNAi and cDNA probes/primers or company names and catalogue numbers if reagents are commercial; cell line names, sources and information on cell line identity and authentication. Animal studies and experiments involving human subjects must be reported in detail, identifying the committees approving the protocols. For studies involving human subjects/samples, a statement must be included confirming that informed consent was obtained. Statistical analyses and information on the reproducibility of experimental results should be provided in a section titled "Statistics and Reproducibility".

All Nature Cell Biology manuscripts submitted on or after March 21 2016, must include a Data availability statement as a separate section after Methods but before references, under the heading "Data Availability". For Springer Nature policies on data availability see <http://www.nature.com/authors/policies/availability.html>; for more information on this particular policy see <http://www.nature.com/authors/policies/data/data-availability-statements-data-citations.pdf>. The Data availability statement should include:

- Accession codes for primary datasets (generated during the study under consideration and designated as "primary accessions") and secondary datasets (published datasets reanalysed during the study under consideration, designated as "referenced accessions"). For primary accessions data should be made public to coincide with publication of the manuscript. A list of data types for which submission to community-endorsed public repositories is mandated (including sequence, structure, microarray, deep sequencing data) can be found here <http://www.nature.com/authors/policies/availability.html#data>.
- Unique identifiers (accession codes, DOIs or other unique persistent identifier) and hyperlinks for datasets deposited in an approved repository, but for which data deposition is not mandated (see here for details <http://www.nature.com/sdata/data-policies/repositories>).
- At a minimum, please include a statement confirming that all relevant data are available from the authors, and/or are included with the manuscript (e.g. as source data or supplementary information), listing which data are included (e.g. by figure panels and data types) and mentioning any restrictions on availability.
- If a dataset has a Digital Object Identifier (DOI) as its unique identifier, we strongly encourage including this in the Reference list and citing the dataset in the Methods.

We recommend that you upload the step-by-step protocols used in this manuscript to [protocols.io](http://www.protocols.io). More details can be found at <https://www.protocols.io/help/publish-articles>.

DISPLAY ITEMS – main display items are limited to 6-8 main figures and/or main tables. For Supplementary Information see below.

FIGURES – Colour figure publication costs \$395 per colour figure. All panels of a multi-panel figure must be logically connected and arranged as they would appear in the final version. Unnecessary figures and figure panels should be avoided (e.g. data presented in small tables could be stated briefly in the text instead).

All imaging data should be accompanied by scale bars, which should be defined in the legend.

Cropped images of gels/blots are acceptable, but need to be accompanied by size markers, and to retain visible background signal within the linear range (i.e. should not be saturated). The boundaries of panels with low background have to be demarked with black lines. Splicing of panels should only be considered if unavoidable, and must be clearly marked on the figure, and noted in the legend with a statement on whether the samples were obtained and processed simultaneously. Quantitative comparisons between samples on different gels/blots are discouraged; if this is unavoidable, it has to be performed for samples derived from the same experiment with gels/blots were processed in parallel, which needs to be stated in the legend.

- For line art, graphs, charts and schematics we prefer Adobe Illustrator (.AI), Encapsulated PostScript (.EPS) or Portable Document Format (.PDF). Files should be saved or exported as such directly from the application in which they were made, to allow us to restyle them according to our journal house style.
- We accept PowerPoint (.PPT) files if they are fully editable. However, please refrain from adding PowerPoint graphical effects to objects, as this results in them outputting poor quality raster art. Text used for PowerPoint figures should be Helvetica (preferred) or Arial.
- We do not recommend using Adobe Photoshop for designing figures, but we can accept Photoshop generated (.PSD or .TIFF) files only if each element included in the figure (text, labels, pictures, graphs, arrows and scale bars) are on separate layers. All text should be editable in 'type layers' and line-art such as graphs and other simple schematics should be preserved and embedded within 'vector smart objects' - not flattened raster/bitmap graphics.
- Some programs can generate Postscript by 'printing to file' (found in the Print dialogue). If using an application not listed above, save the file in PostScript format or email our Art Editor, Allen Beattie for advice (a.beattie@nature.com).

Regardless of format, all figures must be vector graphic compatible files, not supplied in a flattened raster/bitmap graphics format, but should be fully editable, allowing us to highlight/copy/paste all text and move individual parts of the figures (i.e. arrows, lines, x and y axes, graphs, tick marks, scale bars etc). The only parts of the figure that should be in pixel raster/bitmap format are photographic images or 3D rendered graphics/complex technical illustrations.

Unprocessed scans of all key data generated through electrophoretic separation techniques need to be presented in a supplementary figure that should be labeled and numbered as the final supplementary figure, and should be mentioned in every relevant figure legend. This figure does not count towards the total number of figures and is the only figure that can be displayed over multiple pages, but should be provided as a single file, in PDF or TIFF format. Data in this figure can be displayed in a relatively informal style, but size markers and the figures panels corresponding to the presented data must be indicated.

The total number of Supplementary Figures (not including the "unprocessed scans" Supplementary Figure) should not exceed the number of main display items (figures and/or tables (see our Guide to Authors and March 2012 editorial <http://www.nature.com/ncb/authors/submit/index.html#suppinfo>; <http://www.nature.com/ncb/journal/v14/n3/index.html#ed>). No restrictions

apply to Supplementary Tables or Videos, but we advise authors to be selective in including supplemental data.

GUIDELINES FOR EXPERIMENTAL AND STATISTICAL REPORTING

REPORTING REQUIREMENTS – To improve the quality of methods and statistics reporting in our papers we have recently revised the reporting checklist we introduced in 2013. We are now asking all life sciences authors to complete two items: an Editorial Policy Checklist (found here https://www.nature.com/authors/policies/Policy.pdf) that verifies compliance with all required editorial policies and a Reporting Summary (found here https://www.nature.com/authors/policies/ReportingSummary.pdf) that collects information on experimental design and reagents. These documents are available to referees to aid the evaluation of the manuscript. Please note that these forms are dynamic 'smart pdfs' and must therefore be downloaded and completed in Adobe Reader. We will then flatten them for ease of use by the reviewers. If you would like to reference the guidance text as you complete the template, please access these flattened versions at http://www.nature.com/authors/policies/availability.html.

Version 2:

Decision Letter:

Our ref: NCB-A58922B

12th December 2025

Dear Dr. Belting,

Thank you for submitting your revised manuscript "Tumor Acidosis Remodels the Glycocalyx to Control Lipid Scavenging and Ferroptosis" (NCB-A58922B). I'm happy to inform you that we'll be happy in principle to publish it in Nature Cell Biology, pending minor revisions to comply with our editorial and formatting guidelines.

Meanwhile, please revise the manuscript files following the guidelines below:

1. Please convert 'Supplementary Data File 1' as a Supplementary Table item in Excel;
2. Please provide 'Supplementary Figure 1-3' as source data files for gels and blots;
3. Please restructure the current figures into 8 main and up to 10 extended data figures, using main figures to present data, and moving schematics into the extended data figures. No peer-reviewed data should be removed.
4. Please present all figures in a portrait format;
5. For better legibility, please ensure that all figures adhere to a maximum page size of roughly 180mm wide x 200mm high to fit standard page format and use a font size of no smaller than 7pt Arial or Helvetica throughout the figures.

Thank you again for your interest in Nature Cell Biology Please do not hesitate to contact me if you have any questions.

Sincerely,

Zhe Wang, PhD
Senior Editor
Nature Cell Biology

Tel: +44 (0) 207 843 4924
email: zhe.wang@nature.com

Version 3:

Decision Letter:

Dear Dr Belting,

I am pleased to inform you that your manuscript, "Tumor Acidosis Remodels the Glycocalyx to Control Lipid Scavenging and Ferroptosis", has now been accepted for publication in Nature Cell Biology.

Please note that *Nature Cell Biology* is a Transformative Journal (TJ). Authors may publish their research with us through the traditional subscription access route or make their paper immediately open access through payment of an article-processing charge (APC). Authors will not be required to make a final decision about access to their article until it has been accepted. <https://www.springernature.com/gp/open-research/transformative-journals> Find out more about Transformative Journals

Authors may need to take specific actions to achieve compliance with funder and institutional open access mandates. If your research is supported by a funder that requires immediate open access (e.g. according to <https://www.springernature.com/gp/open-science/plan-s-compliance> Plan S principles or the <https://www.springernature.com/gp/open-science/us-federal-agency-compliance> NIH public access policy) then you should select the gold OA route, and we will direct you to the compliant route where possible. Because authors warrant under our subscription licensing terms that they haven't committed to licensing any version of their article under a licence inconsistent with the terms of our agreement – including the applicable embargo period – publication under the subscription model isn't suitable for authors whose funders require no embargo.

If your paper includes color figures, please be aware that in order to help cover some of the additional cost of four-color reproduction,

Nature Portfolio charges our authors a fee for the printing of their color figures. Please contact our offices for exact pricing and details.

If you have not already done so, we strongly recommend that you upload the step-by-step protocols used in this manuscript to protocols.io (<https://protocols.io>), an open online resource that allows researchers to share their detailed experimental know-how. All uploaded protocols are made freely available and are assigned DOIs for ease of citation. Protocols and Nature Portfolio journal papers in which they are used can be linked to one another, and this link is clearly and prominently visible in the online versions of both. Authors who performed the specific experiments can act as primary authors for the Protocol as they will be best placed to share the methodology details, but the Corresponding Author of the present research paper should be included as one of the authors. By uploading your Protocols onto protocols.io, you are enabling researchers to more readily reproduce or adapt the methodology you use, as well as increasing the visibility of your protocols and papers. You can also establish a dedicated workspace to collect your lab Protocols. Further information can be found at <https://www.protocols.io/help/publish-articles>.

Nature Cell Biology encourages authors presenting evidence for cell, biological, molecular, and genetic interactions to consider communicating these findings using Biofactoid (<https://biofactoid.org/>). This tool helps users share a searchable representation of interactions (e.g. binding, gene expression, post-translational modification) between genes, gene products, or chemicals. Information added to Biofactoid, with author attribution, is shared on social media and public databases, such as Pathway Commons, where it can be discovered and analyzed in the context of a large and growing corpus of knowledge.

With kind regards,

Zhe Wang, PhD
Senior Editor
Nature Cell Biology

Tel: +44 (0) 207 843 4924
email: zhe.wang@nature.com

** Visit the Springer Nature Editorial and Publishing website at http://editorial-jobs.springernature.com?utm_source=ejP_NCB_email&utm_medium=ejP_NCB_email&utm_campaign=ejP_NCB for more information about our career opportunities. If you have any questions please click [here](mailto:editorial.publishing.jobs@springernature.com).**

Open Access This Peer Review File is licensed under a Creative Commons Attribution 4.0 International License, which permits use, sharing, adaptation, distribution and reproduction in any medium or format, as long as you give appropriate credit to the original author(s) and the source, provide a link to the Creative Commons license, and indicate if changes were made. In cases where reviewers are anonymous, credit should be given to 'Anonymous Referee' and the source. The images or other third party material in this Peer Review File are included in the article's Creative Commons license, unless

indicated otherwise in a credit line to the material. If material is not included in the article's Creative Commons license and your intended use is not permitted by statutory regulation or exceeds the permitted use, you will need to obtain permission directly from the copyright holder.

Reviewers' Comments:

Reviewer #1 (Remarks to the Author):

“The manuscript "Tumour Acidosis Remodels the Glycocalyx to Control Lipid Scavenging and Ferroptosis" investigates how tumor cells adapt to acidosis and hypoxia microenvironmental stress reprogram cellular metabolism to regulate lipid scavenging and ferroptosis in glioblastoma (GBM) and CNS metastases. They identified a chondroitin sulphate (CS)-enriched glycocalyx as a key feature of the acidic tumor microenvironment, which surrounds tumor cells, restricts extracellular lipid uptake, and shields against lipid-induced ferroptosis. This CS-rich glycocalyx arises from a glycan switch on syndecan-1, driven by HIF and TGF β signaling, replacing heparan sulphate with CS and impairing lipid scavenging. They further show that dual inhibition of CS biosynthesis and diacylglycerol-O-acyltransferase 1 triggers lipid peroxidation and ferroptotic cell death. Overall, the compelling findings highlight the CS-glycocalyx's role in metabolic adaptation and suggest novel therapeutic vulnerabilities in aggressive cancers.”

Our response: We thank the reviewer for acknowledging the novelty and therapeutic implications of our work.

Comments:

“1. This study primarily found that tumor cells in an acidic environment can produce a chondroitin sulphate (CS)-enriched glycocalyx to block the uptake of additional lipids by cells. In vitro experiments have confirmed that an acidic environment is the main trigger for tumor cells to generate a CS-enriched glycocalyx, but this needs further validation in the tumor microenvironment. Can the author measure the acidity level of the tumor microenvironment to further confirm whether regions with CS-rich glycocalyx accumulation have lower pH values. Does the central region of in vitro 3D-cultured tumor cells exhibit a lower pH?”

Our response: While we believe there is consensus from the literature that the perinecrotic region of GBM tumours is acidic, we agree with the reviewer that CAIX, the most widely used marker of acidosis, is not entirely specific of acidosis vs. hypoxia. Accordingly, in Fig. 2 and Extended Data Fig. 3 and 4 it was critical to directly show that acidosis, and not hypoxia, could induce the CS-glycocalyx phenotype. In line with the reviewer's suggestion, we have performed additional experiments employing a pH-sensitive folding and transmembrane insertion of pH (low) insertion peptide (pHLIP), indeed showing that the central, LD and CS-rich region of 3D cultures exhibits an acidic environment. These new data are presented in Extended Data Fig. 1d to h, and changes

were made to the Results (p. 6 lines 109-115) and Methods (p.1, lines 20-24; p.10, lines 233-240; p. 13, lines 310-311; p.14, lines 323-327 and p. 15 lines 342-345) sections.

“2. This study demonstrates that the co-localization of CS-glycocalyx accumulation and lipid stress serves as a hallmark of metabolically challenged regions in aggressive tumors. However, is there a correlation between CS-glycocalyx enrichment and LD accumulation? In Extended Data Fig. 1e, which depicts GBM patient specimens, many regions with high CS-glycocalyx expression lack LDs.”

Our response: We thank the reviewer for this insightful comment and agree that CS is not restricted to LD-rich regions. We have performed additional analyses of GBM patient specimens, scoring CS high expression in LD-positive (LD⁺) vs. LD-negative (LD⁻) regions. The new data, presented in Fig. 1i, showing CS enrichment in LD-positive regions, now provide more direct support for our conclusion. We have updated the Methods accordingly (pp. 11-12, lines 266-269).

“3. Similarly, TGF- β is recognized as a key factor in inducing CS production. Is there a correlation between TGF- β expression and CS levels in the tumor microenvironment?”

Our response: We thank the reviewer for this suggestion and agree that TGF β is a key regulator of CS induction. However, given that TGF β is a soluble factor with activity determined by extracellular interactions and proteolytic processing, rather than steady-state abundance of the cytokine itself, we believe it is technically challenging to accurately capture local TGF β expression in the tumour microenvironment. In our view, the mechanistic experiments linking TGF β activation to CS induction in acidosis (Fig. 2d-h and Extended Data Fig. 5a-e), provide stronger evidence than correlative staining for TGF β in archival specimens. As suggested by the reviewer (point 6), we have highlighted functional evidence that inhibiting TGF- β receptors reduces acidosis-driven CS-glycocalyx formation by moving Extended Data Fig. 5f to Fig. 2h.

“4. In Figure 1H, the authors demonstrated that in hypoxic regions, tumor cells (CA9+/CD31-) are enriched with lipid droplets (LD) and chondroitin sulfate (CS). Additionally, vascular regions (CA9-/CD31+) also exhibit high CS expression despite lacking LD. How do these vascular endothelial cells highly express CS? Is high CS expression specific to tumor vascular endothelial cells? What is the significance of this for the vasculature itself and tumor growth? A discussion of these aspects would be valuable.”

Our response: We appreciate the reviewer's comment and agree that vascular CS enrichment is biologically relevant. Notably, recent work (Shi, et al., Nature, 2025) mapped the brain endothelial glycocalyx in mice, showing ageing-associated shifts in GAGs (including CS/HS). Such vascular glycocalyx alterations may affect barrier leakiness, immune cell infiltration, and the perivascular invasion routes of GBM cells. Future studies should determine whether CS accumulates in the endothelial glycocalyx or is associated with perivascular pericytes, potentially under the influence of TGF β , and whether its abundance distinguishes GBM from healthy brain and low-grade glioma vasculature. We have added a paragraph to the Discussion section on these aspects (p. 19, lines 424-432).

“5. In Extended Data Fig. 2, the authors showed that CNS metastases from kidney, melanoma and lung primaries harboured CS-rich cells in perinecrotic and perivascular compartments. Do these non-glioma primary tumors also harbor CS-rich cells?”

Our response: We agree with the reviewer that this is an interesting and important question. While our data demonstrate the presence of CS-enriched regions in brain metastases from several primaries, we believe that systematically analysing CS remodelling in the respective extracranial primary tumours would require a dedicated effort beyond the scope of the current study.

“6. Extended Data Fig. 5f showed that inhibiting TGF- β receptors reduces acidosis-driven CS-glycocalyx formation, a finding that strongly supports TGF- β 's role in inducing CS production. I suggest moving this result to a main Figure to highlight its significance.”

Our response: We thank the reviewer for this suggestion and have now moved this result to Fig. 2h.

“7. The study shows that tumor cells increase chondroitin sulfate (CS) expression in acidic environments, which limits fatty acid uptake and protects against ferroptosis caused by lipid peroxidation. They also showed that Short-term acidosis was sufficient to increase cell-surface CS levels. However, Figure 5 indicates that combining acidic conditions (pH 6.4) with LDL results in enhanced tumor cell death. Given that acidity-induced CS production should reduce cytotoxicity by limiting lipid uptake, why does this combination exacerbate cell death?”

Our response: We thank the reviewer and agree that clarification was needed. In Fig. 5a, b and Extended Data Fig. 8a, b, parental GBM cells were exposed to high-dose LDL simultaneously with the introduction of acidosis, i.e. prior to metabolic adaptation and before a fully established CS-glycocalyx could form. Under these conditions, the combined stress of acidosis and excess lipid uptake leads to enhanced cell death. Inhibition of CS-glycocalyx formation sensitized cells, with lower lipid doses being sufficient to induce sustained cytotoxicity (Fig. 5c and Extended Data Fig. 8c, d). Moreover, in fully acidosis adapted cells, where a CS-rich glycocalyx had been

established, lipid-induced cytotoxicity required CS inhibition (Fig. 5d, e and Extended Data Fig. 8e, f). We have revised the Results section to clarify the experimental setup and interpretation of these findings (p. 12, lines 266-270).

8. In addition to impeding fatty acid uptake, does the CS-glycocalyx also block the uptake of other substances by cells?

Our response: We thank the reviewer for this interesting point. Our data indicate that CS-glycocalyx enrichment does not cause a general blockade of cellular uptake. Overall endocytic activity and global cell-surface protein internalization were in fact increased in AA compared with NA cells (Extended Data Fig. 6m, n). This suggests that the CS-glycocalyx is selective rather than more general. However, it is likely that other structures known to utilize the SDC1-HSPG-dependent entry route, including viruses, nanoparticles, peptide–nucleic acid complexes, and cell penetrating peptides, may also be hindered by CS-glycocalyx enrichment in acidic tumour regions. We have added this important implication of our findings to the Discussion section of the revised manuscript (p. 18, lines 402-405).

Reviewer #2 (Remarks to the Author):

“In this manuscript, the authors showed that some glioblastoma exhibited higher levels of CS and LDs. They recapitulated this finding in vitro through culturing glioblastoma cancer cell lines under 3-D or acidic pH. Using EV and LDL as the fatty acid carriers, the authors concluded that these metabolic adaptations prevent lipid peroxidation thereby protecting cancer cells from ferroptosis. While the individual piece of the data looks impressive, the story as a whole lacks cohesiveness. More importantly, the physiological relevance of their findings is questionable owing to their choice of vehicles for the fatty acid delivery.”

Our response: We thank the reviewer for a thoughtful overall assessment and suggestions of how to improve the clarity of our study.

“1. The authors relied on exogenous fatty acids delivered through LDL or EV to determine the mechanism behind their findings. While the contribution of EV to fatty acid transport in brains remains to be established, LDL definitely cannot cross BBB to reach brains. Thus, the choice of these fatty acid delivery vehicles makes it uncertain whether the mechanism they uncovered in vitro is physiologically relevant. According to the current knowledge, brain-derived lipoproteins, which contain apoE but not apoB, the primary component of LDL, are the most important fatty acid transporters in brains. The authors may want to reexamine their findings by using these particles to deliver exogenous fatty acids.”

Our response: We appreciate the reviewer’s concern that LDL is unlikely to cross an intact BBB. However, GBM tumours are well known to disrupt BBB integrity, creating regions of vascular permeability and abnormal transcytosis, particularly in hypoxic/acidic tumour areas (see e.g., Arvanitis *et al.*, *Nat Rev Cancer*, 2023). Moreover, our central conclusion, that a CS-rich glycocalyx inhibits extracellular lipid particle scavenging, is not dependent on LDL itself or on a specific apolipoprotein receptor. Rather, we demonstrate disruption of the SDC1–HSPG uptake pathway, which also mediates scavenging of apoE-containing lipoproteins (Wilsie, *et al.*, *Lipids Health Dis.*, 2006; MacArthur, *et al.*, *J Clin Inv.*, 2007; Gonzales *et al.*, *J Clin Inv.*, 2013). Nevertheless, we have performed additional experiments with HDL (that contains apoE, see new Extended Data Fig. 6j), indeed showing reduced uptake in AA vs. NA cells that was comparable to our previous data with LDL and EVs (see new Fig. 3h). We have updated the Result section (p.10, lines 211-212) and Methods sections (p. 1, line 16-17; p. 9, line 202; p. 16, line 376) accordingly.

We have clarified these important points in the revised Discussion section and explicitly note that LDL and EVs were employed in parallel to probe the broader principle of lipid particle uptake in this context (p. 18, lines 396-402).

“2. The manuscript contains parts that are in conflict with each other. For example, in figure 1, the authors left an impression of the existence of LD⁺/CS⁺ glioblastoma, and used the increased expression of FASN as the gene expression signature that marked for the tumor cells accumulating LDs. However, in figure 3, they showed images that the accumulation of LDs and CS was mutually exclusive, and in extended data Fig. 6 they concluded that FASN was not required for LD accumulation. These inconsistencies make the manuscript difficult to understand.”

Our response: We thank the reviewer for raising this point and appreciate the opportunity to clarify. FASN was included in the gene expression signature for regulators of lipid metabolism and LD formation in Fig. 1f. However, this was not conclusive of a requirement for FASN in LD formation under acidosis, which motivated functional experiments with FASN inhibition. Our data show that acidosis-induced LD accumulation is independent of FASN (Extended Data Fig. 6a) and instead relies on extracellular lipids (Extended Data Fig. 6b-e). Likewise, CS-glycocalyx induction under acidosis also depended on extracellular lipid availability (Extended Data Fig. 6d, f-h), indicating that the CS⁺/LD⁺ phenotype is independent of FASN and instead requires extracellular lipids. We also wish to clarify a possible misunderstanding: Our data do not show that CS and LD accumulation are mutually exclusive. Rather, they are functionally connected. To dissect how the **co-existence** of increased lipid storage and CS-glycocalyx induction may be functionally linked, we pharmacologically blocked LD formation using the DGAT1i. LD disruption resulted in a further, compensatory increase of CS-glycocalyx expression in both acidic 2D cultures (Fig. 3b and Extended Data Fig. 6i) and 3D spheroids (Fig. 3c, d). Conversely, inhibition of CS results in compensatory upregulation of LD accumulation (Fig. 5f, g and Extended Data Fig. 8i, j). This reciprocal regulation is central to our model, in which the CS-glycocalyx shield and the LD intracellular sink cooperatively mediate lipid homeostasis during cellular adaptation to acidosis, preventing lipid particle overload and lipotoxicity. We have revised the Results section to make this interplay between LDs and CS clearer (p. 9, lines 195-208).

“3. The authors employed multiple conditions to analyze the accumulation of CS and LDs in the tumor cells such as 3D culture, lower pH, hypoxia-mimicking treatment, and TGFβ without giving a clear picture of the relationship among these events in tumor microenvironment. A diagram illustrating the sequence of these events will markedly help readers to understand these studies.”

Our response: We agree and have now refined our schematic figure (new Extended data Fig. 10g) that summarizes the sequential interplay between acidosis, HIF/TGFβ signalling, LDs, CS switch, lipid uptake, and ferroptosis.

Reviewer #3 (Remarks to the Author):

“Bang-Rudenstam et al. use analyses of patient tumors, patient-derived 3D models, and in vivo systems to identify chondroitin sulphate (CS)-enriched glycocalyx as a hallmark of the acidic tumor microenvironment in glioblastoma and CNS metastases. They go on to show that this glycocalyx can protect cancer cells from ferroptosis, and that targeting of CS and DGAT-1 is capable of inducing cancer cell death by ferroptosis. Although lipid droplet accumulation has been demonstrated before in glioblastoma models, these findings are novel because they demonstrate a targetable dependency of glioblastoma and CNS metastases via glycan remodelling. The implications of this stress-induced glycosylation program in the patient and human PDC-derived models demonstrating potential translatability of these findings will be of great interest to the field. Prior to publication, a few technical concerns should be considered:”

Our response: We are grateful to the reviewer for recognizing the novelty and translational relevance of our findings.

“Major comments

1. Recent work has shown that siRNA treatment can lead to GPX4 upregulation and ferroptosis sensitization (PMID: 38489367). Because siRNA-mediated knockdown of GSGALNACT1 is used in this work, the authors should acknowledge this limitation and provide the controls demonstrating the extent to which this phenomenon could be impacting interpretation of the ferroptosis-pertinent results.”

Our response: We thank the reviewer for raising this important point. We have now acknowledged this limitation and performed additional controls demonstrating that siRNA-mediated knockdown of CSGALNACT1 does not alter GPX4 expression, confirming that the observed effects are specific to CSGALNACT1 knockdown and in line with CSi treatment studies. The new data are presented in Extended Data Fig. 9d and we have updated the Results section accordingly (p. 14, lines 316-319).

“2. In Fig. 6i and Fig. 6j, in order to demonstrate that the cytotoxicity is caused by ferroptosis in addition to the ferrostatin and liproxstatin controls, inhibitors of other forms of cell death (QVD, 3-MA, Nec-1s) should be included in order to demonstrate specificity of the cell death to ferroptosis (see PMID: 40204928 for reagents and dosages to include)”

Our response: We thank the reviewer for this important suggestion. In line with the recommendation, we included QVD, 3-MA, and Nec-1s in the treatment assays. Neither 3-MA nor Nec-1s mitigated the cytotoxic effects of CSi/DGAT1i treatment, confirming

that these cell death pathways are not involved. While QVD reduced cytotoxicity as measured by cell membrane disruption (Cytotox), none of the inhibitors, including QVD, rescued overall cell density, indicating that the ferroptosis-driven loss of viability persisted. The QVD effect on membrane disruption is consistent with recent studies reporting that caspase inhibition can modulate ferroptotic cytotoxicity downstream of lipid peroxidation (PMID: 40301648), reflecting an emerging cross-talk between ferroptosis and apoptosis (PMID: 39929794). These new results, presented in Extended Data Fig. 9l-n, support that ferroptotic and apoptotic pathways may intersect mechanistically without contradicting our conclusion that the CSi/DGAT1i treatment effect is ferroptosis dependent. Text changes were made to the Results (p.15, lines 341-346) and Methods (p.1, lines 9, 10 and 18; p. 8, lines 186-188) sections.

“3. BODIPY stains shown in Fig. 6g, h should be represented a ratio of oxidized to reduced dye and representative flow cytometry plots of these plots should be included”

Our response: We thank the reviewer for raising this important point and fully agree that the data should be represented as a ratio of oxidized to reduced BODIPY signal. The experiments evaluating lipid peroxidation potential in this study were performed using the IncuCyte S3 Live-Cell Analysis System, which quantifies the signals of the oxidized (green) and reduced (red) forms of the BODIPY probe. We have now reanalyzed the data and included additional experiments in which lipid peroxidation potential was calculated as the ratio of green integrated intensity (oxidized BODIPY) to red integrated intensity (reduced BODIPY) per well, normalized to cell number. The reanalyzed results are consistent with our original findings and have been updated in Fig. 6g (including representative IncuCyte images) and in Extended Data Fig. 9g, h. We have also revised the Methods section (p. 14, lines 318–321; pp. 14-15, lines 341–343) and clarified the analysis workflow for IncuCyte data (p. 15, lines 345–348).

“4. In Fig. 9l, it is demonstrated that these effects required the presence of extracellular lipids. Since the types of extracellular lipids (MUFA versus PUFA) make a difference in terms of ferroptosis sensitivities and it would be important to add back either MUFA or PUFA in the serum-free conditions to determine if exogenous MUFA might be able to protect from ferroptosis in this context.”

Our response: We thank the reviewer for this valuable comment. While our current study does not explicitly distinguish MUFA and PUFA contributions, our data demonstrating that extracellular lipoprotein and EV availability is required for ferroptosis induction are fully consistent with the concept that natural lipid sources shape ferroptosis sensitivity. Strategies to modulate PUFA supply represent an important mechanistic extension, and we have highlighted this as a future direction in the revised Discussion (see our response to minor comment 3).

“5. For the pilot trial that is references in lines 340-342, this data should be shown (and repeated with greater n) or this sentence should be removed from the manuscript text. The ability to include this data would greatly strengthen the impact of this work.”

Our response: We appreciate the reviewer’s comment. Because this was conducted as a pilot feasibility trial, we agree and have removed the sentence referring to it in the revised manuscript.

“Minor comments

1. Line 347 - MDA and SLC7A11 would be more appropriately referred to as “markers associated with ferroptosis” as they are not considered ferroptotic markers”

Our response: We agree and have made suggested changes (p.16, lines 359-360).

“2. In Fig. 3e and ED Fig. 8h (U3047MG), presumably the control cohort would be set to 1.0, but this does not appear to be the case. Please check.”

Our response: We thank the reviewer for noting this. The control values in Fig. 3e and Extended Data Fig. 8h have now been corrected to 1.0, and we carefully checked all the figure panels.

“3. In the discussion, these findings should be further contextualized and discussed in relation to PMID: 34118189, which has shown an effect of the acidic tumor microenvironment on ferroptosis.”

Our response: We agree and now discuss PMID: 34118189 in relation to our findings in greater detail (pp. 19-20, lines 440-449). We believe the mechanistic complementarity highlights a therapeutic opportunity, whereby concurrent disruption of LDs and the CS-glycocalyx barrier could be particularly effective when combined with interventions that increase the dietary supply and peroxidation of PUFAs.

Reviewers' Comments:

Reviewer #1 (Remarks to the Author):

“The authors have successfully addressed my previous concerns.”

Our response: We thank the reviewer for their constructive feedback during the initial review, which has helped substantially strengthen the manuscript.

Reviewer #2 (Remarks to the Author):

“The revised manuscript addressed some of my previous concerns but the key point, that is, the physiological relevancy of the lipoprotein particles used as the vehicle for fatty acid delivery remains unresolved. The authors argued that GBM may affect integrity of BBB thereby allowing LDL to reach brains. However, the paper they cited in the rebuttal letter only said that GBM altered transportation across BBB but did not say that BBB was so disintegrated that particles as large as LDL could reach brains in patients with GBM. The new experiment, which used HDL, also suffered from the same problem, as HDL cannot cross BBB either. The brain-derived lipoprotein particles, which are assembled by apoE, are NOT HDL, which is assembled by apoA1, even though the brain-derived lipoprotein particles are often referred to HDL-like. The authors may want to use in vitro lipidated apoE (PMID: 39532095) or lipoproteins secreted by astrocytes (PMID: 35235798) to test their hypothesis.”

Our response: We thank the reviewer for revisiting this important point and for the opportunity to clarify our rationale more explicitly. In the revised manuscript, we now clearly state that 1) HDL-like, apoE-containing lipid particles constitute the predominant physiological lipid carriers in the healthy brain, and 2) neither LDL nor canonical plasma HDL traverse the intact BBB. We also appreciate the reviewer’s point that, although GBM exhibits profound BBB disruption, the extent to which circulating lipoproteins contribute to the tumor ecosystem remains incompletely defined. Of note, however, GBM-derived EVs themselves readily leak into the circulation (e.g., PMID: 38567448), illustrating that lipid-particle exchange is feasible under pathological conditions.

That said, we respectfully disagree that the physiological relevance of our lipid particle models remains unresolved. We do not claim that plasma LDL is a physiological lipid carrier in the CNS. Rather, our central focus is on EVs, which represent endogenous lipid carriers in the brain (reviewed in *Nat Rev Neurosci.*; PMID: 39972160). Because EVs and HDL-like particles are technically challenging to isolate in large quantities, we employ LDL, HDL, and EVs as established model particles to systematically interrogate the SDC1-HSPG uptake pathway.

Most importantly, apoE is known to engage the SDC1-HSPG route (Wilsie et al., *Lipids Health Dis.*, 2006; MacArthur et al., *J Clin Invest.*, 2007; Gonzales et al., *J Clin Invest.*, 2013), indicating that HDL-like, apoE-rich particles produced by astrocytes and microglia would rely on the same uptake machinery. Our data therefore demonstrate that CS-glycocalyx induction suppresses uptake of multiple structurally distinct lipid particles that share dependence on SDC1–HSPG-mediated internalization, establishing

a broader principle: the CS-glycocalyx restricts lipid particle scavenging and controls ferroptosis sensitivity.

Finally, we agree that the reviewer's suggested approaches using reconstituted lipidated apoE or astrocyte-derived lipoproteins would provide valuable complementary insight. However, these systems require substantial technical development and scale optimization that extend beyond the scope of the current study. In response to the reviewer's feedback, we have now 1) limited statements regarding plasma lipoprotein entry across the leaky BBB, 2) explicitly framed LDL/HDL/EVs as experimental tools used to probe SDC1-HSPG-dependent lipid scavenging, and 3) highlighted targeted studies using astrocyte-derived apoE particles as a future research direction in the Discussion (p. 18, lines 392–402).

We hope this expanded clarification fully addresses the reviewer's concerns.

Reviewer #3 (Remarks to the Author):

“The authors have sufficiently addressed my prior review comments, and the added experiments and data greatly strengthen the manuscript.”

Our response: We appreciate the reviewer’s constructive input, which has further improved the impact of our study.

Point-by-point response to the reviewers' comments: All reviewer comments were fully addressed in the previous revision, and no additional comments were raised at this stage.